# Industrial and agricultural chemicals exhibit antimicrobial activity against human gut bacteria in vitro

Indra Roux [1,11], Anna E. Lindell[1,7,11], Anne Grießhammer [2], Tom Smith[1,8], Shagun Krishna [3,9], Rui Guan[1,10], Deniz Rad [1], Luisa Faria[1], Sonja Blasche[1], Kaustubh R. Patil[4,5], Nicole C. Kleinstreuer [3], Lisa Maier [2], Stephan Kamrad [1] ✉ & Kiran R. Patil [1,6] ✉

Industrial and agricultural chemicals such as pesticides are often considered to have restricted biological activity. Yet, there are concerns regarding their broader toxicity range and impact on human gut microbiota. Here we report a systematic in vitro screening to assess the impact of 1,076 pollutants, spanning diverse chemistries and indicated applications, on 22 prevalent gut bacteria. Our investigation uncovered 588 inhibitory interactions involving 168 chemicals, the majority of which were not previously reported to have antibacterial properties. Fungicides and industrial chemicals showed the largest impact, with around 30% exhibiting anti-gut-bacterial properties. We demonstrate that the scale of our dataset enables a machine learning approach for predicting the antibacterial activity of pesticides. Mechanistically, chemical–genetic screens using transposon mutant libraries of *Parabacteroides merdae* and *Bacteroides thetaiotaomicron* implicated genes involved in conserved efflux pathways, including the *acrR* locus, as mediators of pollutant resistance. We also found that loss-of-function mutations in genes coding for metabolic enzymes were selected under pollutant exposure, including those for branched short-chain fatty acid biosynthesis under tetrabromobisphenol A, a flame retardant. Taken together, our results suggest that the antibacterial activity of chemical pollutants should be considered in future studies on the microbiome and the emergence of antimicrobial resistance, as well as in toxicological assessments.

Synthetic chemicals have become indispensable for agriculture and industry. Their pervasive use and environmental persistence have led to pollution levels exceeding the planetary boundary for stable and resilient Earth systems[1–3]. Pesticide pollutants are of particular concern, with the Food and Drug Administration and European Food Safety Authority documenting hundreds of different pesticide residues in food[4,5], as well as broader water contamination[6,7]. Industrial chemicals, such as per- and polyfluoroalkyl substances (PFASs), plasticizers, surface coatings and dyes, also routinely enter food and water via environmental pollution, agricultural application or industrial processing[8].

The contamination of food and water by chemical pollutants potentially exposes the human gastrointestinal tract, through passage and enterohepatic circulation, to thousands of xenobiotic compounds[9]. Over 95% of Americans and nearly the entire Dutch population are estimated to have detectable levels of PFASs in their blood[10,11]. In a UK cohort, all tested urine samples showed detectable levels of

pesticides, such as cypermethrin and permethrin (>96% samples), diethylphosphate (75% samples) and glyphosate (53% samples)[12]. Similarly, a German mother cohort study found hundreds of chemicals derived from consumer goods, cosmetics and food in plasma, reaching up to micromolar concentrations[13]. These studies indicate that the gut microbiota—a community of hundreds of microbial species—is probably exposed to numerous chemicals. Yet, little is known about the impact of agricultural and industrial chemical contaminants on gut bacteria.

Chemicals such as pesticides are usually marketed with a narrow definition of target organisms (for example, as insecticides, herbicides or fungicides). Yet, many chemicals have been reported to show off-purpose activity on non-target organisms, including soil bacteria[14] and pathogens[15]. A small number of pesticides and industrial chemicals have also been shown to impact human and animal gut microbiota[12,16–24]. More broadly, human-targeted drugs have also been shown to impact gut bacterial growth and metabolism[25,26]. Collectively, these studies indicate that many pesticides and other chemical pollutants might exhibit inhibitory activity against gut bacteria. However, safety assessments for these chemicals currently do not consider the human gut microbiome.

In vivo assessment of the impact of chemicals on gut bacteria is currently not possible due to the lack of systematic and quantitative data on chemical exposure. Furthermore, disentangling the effects of chemicals from other factors such as diet and medications is not possible without mechanistic knowledge of direct chemical–bacteria interactions[27,28]. We therefore conducted a large-scale in vitro screen assessing the anti-gut-bacterial effect of 1,076 chemical pollutants, including pesticides, pesticide metabolites and industrial chemicals. The screen uncovered numerous growth inhibitory interactions, particularly with fungicides and industrial chemicals. We further used chemical–genetic screens to identify mechanisms of bacterial susceptibility to chemicals that overlap with antibiotic resistance. Our study expands the knowledgebase on xenobiotic toxicity in bacteria and demonstrates the feasibility of computational prediction using machine learning on chemical structures.

## Results

### Comprehensive library of chemical contaminants

To systematically assess the impact of chemical contaminants, we used a large library of 1,076 compounds that are likely to enter food and water (Extended Data Fig. 1a). The library includes 829 pesticides (mainly herbicides, insecticides and fungicides), along with compounds targeting other organisms, including spiders, nematodes, bacteria and rodents. It also contains 119 known pesticide metabolites formed by biotransformation[29] and 75 pesticide-related compounds (precursors, breakdown products, chemical isomers or formulation products). In addition, the library contains 48 industrial chemicals, such as the widespread contaminants bisphenols and nitrosamines, and five mycotoxins found in mould-contaminated foods. Several compounds are considered to be persistent organic pollutants, such as PFASs and the insecticides dichlorodiphenyltrichloroethane (DDT, also known as clofenotane) and chlordecone. Overall, the library spans 291 out of 352 pesticide classes in the British Crop Production Council Pesticide Compendium and 87% of compounds reported in a study of pesticide discharge from rivers into oceans[6]. The library, together with extensive metadata (Supplementary Data 1), constitutes a resource for systematic investigation of the impact of chemical pollutants on biological systems.

### Chemical contaminants from diverse classes inhibit the growth of gut bacteria

We performed the chemical–bacteria screen at 20 µM concentration for all compounds. This dose was chosen for consistency with previous xenobiotic screens[25,30,31], thus enabling direct comparison. To contextualize this dose, we analysed xenobiotic plasma concentrations

from a recent study[13]. In 305 of 592 samples, at least one xenobiotic was observed at a concentration of 10 µM or higher. Among the measured xenobiotics, spinosyn A, melamine, mepiquat, acrylamide and amitrole, all of which are present in our compound library, most frequently surpassed 2 µM in plasma (45, 40, 39, 34 and 33 samples, respectively). Pharmacokinetic studies suggest that colonic concentrations tend to be in the same range or higher than in the plasma[25,32]. The relevance of micromolar-range doses is further supported by the food regulatory limits of some chemicals[33]. Furthermore, chemicals such as the antiparasitic closantel have been found in dairy products at micromolar levels[34,35].

Next, we assessed the impact of all 1,076 compounds in the library on the growth of 22 commensal gut bacterial strains belonging to 21 species (Fig. 1a,b). These strains were selected based on their abundance and prevalence in the healthy human gut microbiota, as well as phylogenetic and metabolic representation (Supplementary Data 2)[36,37]. All strains were grown anaerobically at 37 °C in mGAM broth, a rich medium designed to support the growth of a maximum number of strains. Bacterial growth was monitored for 24 h and quantified as the area under the growth curve (AUC). We defined growth inhibition hits as chemical–bacteria interactions featuring a growth decrease of >20% and a false-discovery-rate adjusted $P$ value ($P_{adj}$) < 0.05. Only hits identified in at least two out of three biological replicates are considered (Supplementary Data 3 and Extended Data Fig. 1b).

Around one-sixth of the tested chemicals (168) were inhibitory against at least one bacterial strain (Fig. 1c). The most sensitive taxa were Bacteroidales, especially *Parabacteroides distasonis*, whereas *Escherichia coli*, *Fusobacterium nucleatum* subspecies *animalis* and *Akkermansia muciniphila* were the least sensitive species (Fig. 1b,d). The chemical categories featuring prevalent anti-gut-bacterial activity (at least one in five compounds) were fungicides, industrial chemicals and acaricides (Fig. 1b). Although most compounds were inhibitory against a few strains, indicating a narrow spectrum of activity, 24 chemicals showed broad toxicity, inhibiting more than one-third of the species tested (Extended Data Fig. 2b). Broad-spectrum inhibitory compounds include closantel (a livestock antiparasitic; 19 strains), tetrabromobisphenol A (TBBPA; a brominated flame retardant; 19 strains), chlordecone (an insecticide; 17 strains), bisphenol AF (BPAF; used in plastics; 12 strains), fluazinam (a fungicide; nine strains) and emamectin benzoate (an insecticide; 13 strains).

Around 150 chemical–bacteria interactions featured >90% growth reduction, uncovering strong anti-gut-bacterial activity for 33 compounds (Fig. 1e). To test whether the inhibitory activity was retained at lower concentrations, we tested 11 pesticides against eight bacteria across multiple concentrations (Fig. 1f and Supplementary Data 4). Independently purchased compounds were used to account for any errors associated with batch variations and large-scale library preparations. Most interactions from the main screen were also observed in this independent experiment (72%; 23 out of 32; Extended Data Fig. 1c,d), which is the expected range considering the physiological idiosyncrasies of non-model bacteria[38]. Many compounds show strong inhibitory effects at substantially lower concentrations than those tested in the main screen. For example, imazalil sulfate, closantel and prochloraz inhibit *Eubacterium rectale*, a prevalent species that produces the health-associated metabolite butyrate[39], at a concentration as low as 2.5 µM. Together, our systematic assessment revealed hundreds of chemical–bacteria interactions, including contaminants with broad and potent anti-gut-bacterial activity.

### Commonality of susceptibility to therapeutic drugs and chemical contaminants

The off-target activity of chemicals against gut bacteria has been reported for >1,000 human-targeted drugs and antibiotics by Maier et al.[25,30]. As in the case of therapeutic drugs, we observed a positive correlation between the number of compounds affecting a species

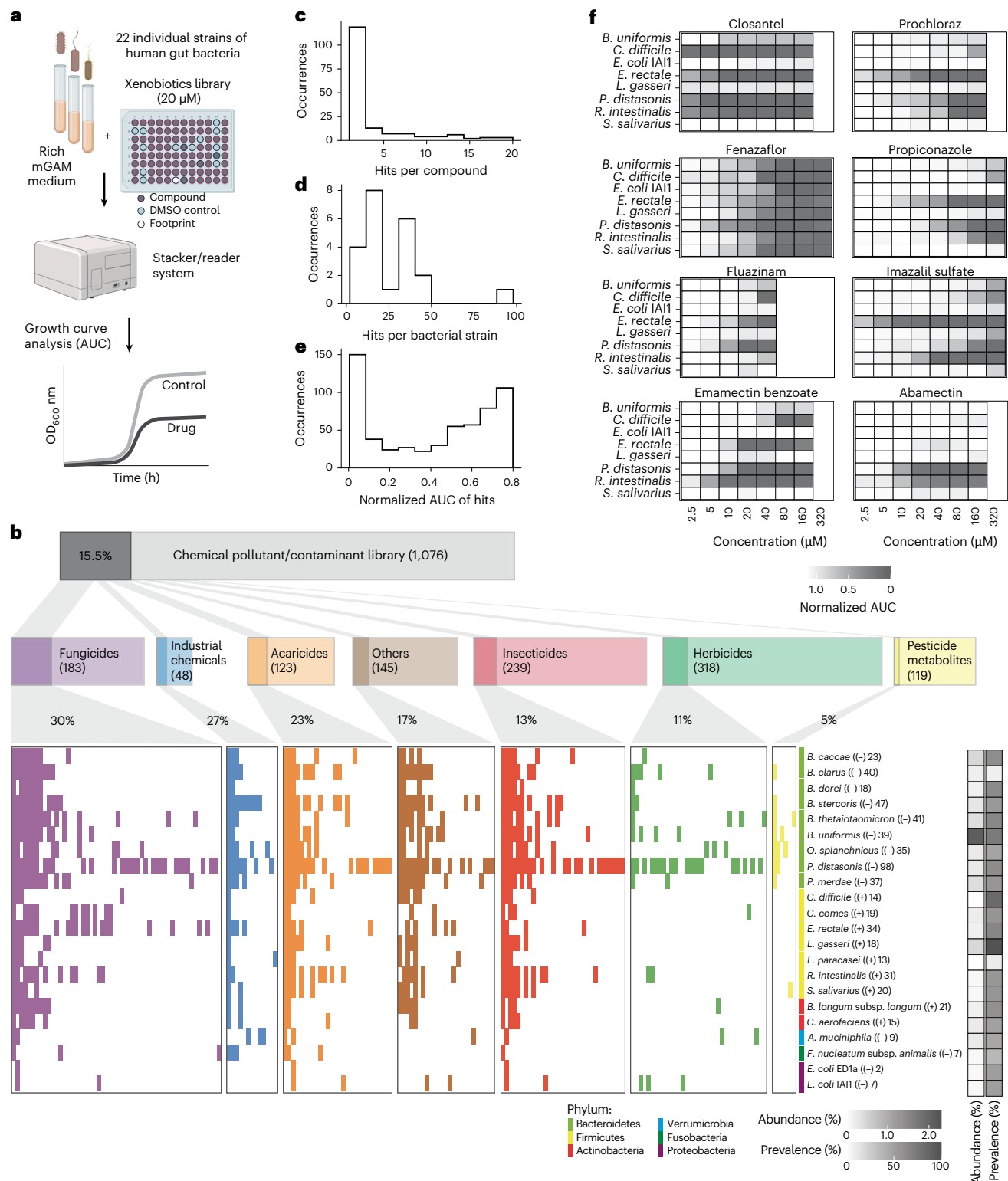

**Fig. 1 | Extensive impact of food contaminants on prevalent human gut bacteria. a**, Workflow of high-throughput screening to test the susceptibility of gut bacteria against industrial and agricultural chemicals. Monocultures were grown in mGAM medium with 20 µM of each compound (*n* = 3 experiments on independent days). Growth curves were recorded and the AUC relative to DMSO controls was used as a fitness proxy. **b**, High-throughput growth screening results for 22 gut bacterial strains. Significant pollutant chemical–strain interactions, where the growth of a specific strain was inhibited versus the DMSO control, are highlighted with a coloured bar ($P_{adj}$ < 0.05; >20% reduction in AUC; two out of three independent replicates significant). The data are grouped and coloured by compound class. 'Others' refers to compounds not assigned to any of the

classes shown. Note that in rare cases a single compound can belong to more than one compound class. The phylum, abundance and prevalence of strains are displayed in coloured and greyscale bars, and the Gram stain and number of compounds that inhibit growth are shown in parentheses after each strain name. **c–e**, Distributions of the number of inhibitory bacterial interactions (hits) per compound (**c**) and bacterial strain (**d**), and the strength of growth inhibition, shown as mean normalized AUC values of hits (**e**). **f**, Independent validation screen. Eight bacteria were tested at up to eight concentrations (*n* = 3 independent experiments). The colours indicate mean rAUC values normalized to DMSO controls. Panel **a** created with BioRender.com.

and its relative abundance in human microbiomes (Spearman's rank correlation ($\rho$) = 0.57; $P$ = 0.005) (Fig. 2a), but none with prevalence (Fig. 2b). Thus, chemicals with broad- as well as narrow-spectrum activity are likely to influence microbiome composition due to their impact on abundant taxa. Strains sensitive to many chemical pollutants were also sensitive to many therapeutic drugs tested by Maier et al. ($\rho$ = 0.63; $P$ = 0.009) (Fig. 2c). Pollutant sensitivity correlated more strongly with human-targeted drugs ($\rho$ = 0.66; $P$ = 0.004) than with antibiotics ($\rho$ = 0.33; $P$ = 0.19) (Fig. 2d), which is consistent with the broader-spectrum activity of antibiotics. The correlation between chemical pollutants in our screen and human-targeted drugs supports that gut bacteria interact with diverse chemicals via common and non-specific mechanisms[25,28,40].

Chemical susceptibility profiles of bacterial species grouped clearly by evolutionary relatedness (that is, by phylum; Fig. 2e), with most compounds showing phylum-level differences in toxicity (Extended Data Fig. 2a). For example, some Firmicutes were uniquely sensitive to conazole fungicides, including the widely used imazalil and prochloraz. This indicates a shared genetic basis for toxicity and suggests that chemical susceptibility data could help to predict effects on related bacterial species for in silico toxicological assessments.

## Chemicals alter the composition of synthetic gut bacterial communities

We next investigated how species-level chemical effects translate in bacterial communities using a synthetic community of 20 gut bacteria (Com20)[38] (Methods and Supplementary Data 5). Com20 is a phylogenetically and functionally diverse community, spanning six phyla, 11 families and 17 genera, collectively encoding >60% of the metabolic pathways present in healthy human gut microbiomes. Members of the community grow together stably and reproducibly in mGAM[38]—the growth medium used in our screen. The Com20 was transferred twice in fresh growth medium before being challenged with BPAF or TBBPA, two broad-spectrum anti-gut-bacterial chemicals. In the case of TBBPA, the community was dominated by *Bacteroides thetaiotaomicron*, despite its susceptibility in monoculture. BPAF induced compositional changes concordant with monoculture effects (Fig. 2f), including decreased relative abundances of *B. thetaiotaomicron*, *Bacteroides uniformis*, *Parabacteroides merdae* and *Roseburia intestinalis*, and increased abundances of insensitive species (that is, *Collinsella aerofaciens*, *Coprococcus comes* and *Streptococcus salivarius*). Yet, *F. nucleatum* and *E. rectale* were protected against BPAF in the community (Fig. 2g).

Community-level effects, such as cross-protection, have been described in gut bacterial communities exposed to therapeutic drugs and could partly be attributed to compound bioaccumulation[31,41]. We hypothesized that cross-protection against BPAF in Com20 is due to bioaccumulation, as this has been observed in Bacteroidales monocultures[42]. We measured BPAF concentrations in whole culture, supernatant and pellets of Com20 cultures exposed to 20 µM BPAF using liquid chromatography–tandem mass spectrometry (LC–MS/MS). We detected a 64% reduction in BPAF in supernatant compared with whole culture and recovered BPAF in the pellet (Fig. 2h), indicating that most of BPAF in Com20 is sequestered within cells, explaining cross-protection of some community members.

## Expanded coverage of the chemical landscape

Large-scale surveys of xenobiotic–bacteria interactions, combined with computational analyses, could enable in silico toxicity predictions[43]. However, accurate prediction requires sufficient data spanning the relevant chemical space. To assess this, we compared the extended-connectivity fingerprints (ECFPs) of our pollutant library with those of previously characterized pharmaceutical drugs[25] and 250,000 random PubChem compounds.

Pollutants and pharmaceutical drugs were widely dispersed in uniform manifold approximation and projection (UMAP) space

(Fig. 3a), indicating large chemical diversity for both groups. Notably, pollutants and drugs did not form clear clusters, and only six pollutants had a Tanimoto similarity to drugs of >0.75, highlighting limited molecular fingerprint overlap between the two libraries[44,45] (Fig. 3b). This suggested that labels such as 'pesticide' or 'pharmaceutical drugs' do not capture the molecular basis of anti-gut-bacterial toxicity. To quantify how well pollutant and drug libraries jointly span the chemical space, we noted the Tanimoto similarity of each PubChem compound to its closest match in the libraries. This was significantly increased in the combined library ($P < 2.2 \times 10^{-16}$; Kolmogorov–Smirnov test; Extended Data Fig. 3a), indicating broader chemical-space coverage towards predicting the impact of novel compounds.

## Machine learning predicts antimicrobial effects

To probe the utility of increased chemical-space coverage, we employed machine learning for predicting the anti-gut-bacterial activity of chemicals. We used two datasets: pesticides from our pollutant screen and the drug screen by Maier et al. (Fig. 3c). For each compound, we obtained a comprehensive set of structural features, including molecular fingerprints (Chemistry Development Kit) and deep-learning-based feature embeddings from MoLFormer[46]. The latter is a deep learning model trained on 1.1 billion molecules to predict a wide range of chemical properties, such as substructures, from simplified molecular input line entry system (SMILES) strings. To avoid excessively imbalanced groups for machine learning modelling, we chose seven species that showed >20 hits in both pollutant and drug datasets. Random forest classifier models were trained and evaluated using cross-validation with stratified splits. Classifications were obtained by thresholding probability outputs at the hit frequency of the training dataset—a strategy suitable for imbalanced datasets[47].

We evaluated models using two metrics appropriate for imbalanced datasets: balanced accuracy (the mean of sensitivity and specificity) and average precision (the area under the precision-recall curve)[48]. Models trained and tested on pesticides alone achieved good performance when predicting pesticide toxicity (average balanced accuracy = 0.77; average precision = 0.65; 20× cross-validation; Fig. 3d). Performance varied slightly between species, with the best results obtained for *E. rectale* (balanced accuracy = 0.8; average precision = 0.73) and the weakest predictions for *B. thetaiotaomicron* (balanced accuracy = 0.73; average precision = 0.52). Training models on only molecular descriptor features (that is, excluding MoLFormer embeddings) slightly improved the performance (balanced accuracy = 0.81; average precision = 0.72), indicating that this smaller feature set is sufficient for predictions. Analysis of feature importance (Extended Data Fig. 3) identified molecular features implicated in antibacterial activity. Although a model trained and tested on the combined pesticide–drug dataset showed similar performance, cross-predictions from one dataset to another showed poorer performance (approximate balanced accuracy = 0.7; average precision = 0.2). This indicates that pesticides and drugs are overall different in both chemical and activity space (Fig. 3a). The pesticide toxicity against gut bacteria thus could not have been predicted based on previously available data.

## Conserved genetic response to pollutants with implications for antibiotic resistance

Given the high sensitivity of Bacteroidales to pollutant compounds (Fig. 2a and Extended Data Fig. 2a), we aimed to investigate mechanisms of interaction in species of this order. We used a pooled transposon mutant library of *P. merdae*[49] to identify genes that modulate the impact of xenobiotics on bacterial fitness. The library encompasses barcoded transposon insertion mutants (Tn mutants) of >3,000 non-essential genes, enabling a genome-wide functional analysis. We conducted the competition assay against a panel of ten chemicals. TBBPA, BPAF, imazalil sulfate, fluazinam, closantel and emamectin benzoate were tested at concentrations of ≤20 µM, whereas glyphosate,

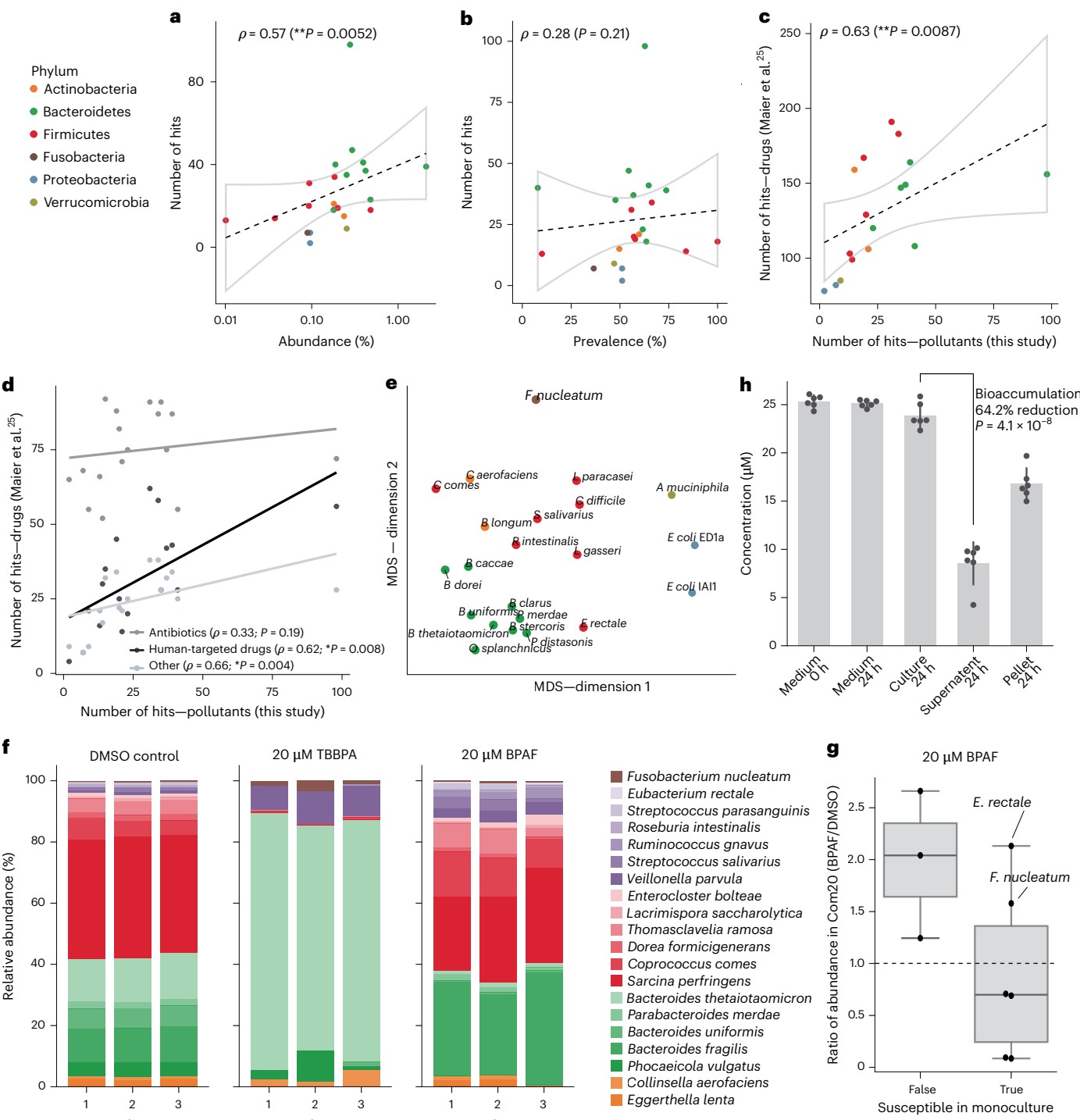

**Fig. 2 | Xenobiotic susceptibility in the microbial community context.**
**a**,**b**, Correlation between average species abundance (**a**) or prevalence (**b**), observed in human cohort studies, and compound sensitivity across gut bacterial strains (*n* = 22 strains). The phyla are colour coded. The lines depict the best linear fit and the grey shapes indicate 95% confidence intervals. *ρ* is the Spearman's rank correlation coefficient (two sided). **c**, Correlation between the number of pollutant compounds inhibiting a strain (this study) and the number of human-targeted drugs inhibiting the same strain in the study by Maier et al.[25] (*n* = 17 strains). **d**, As in **c**, but stratified by drug type. **e**, Multi-dimensional scaling (MDS) of the strain sensitivity profiles of strains (phylum colour coded as in **a**). The distances were computed with binary inhibition classification using the Jaccard metric. MDS was performed with scikit-learn using default settings. **f**, Relative abundances of synthetic community Com20 species after exposure to BPAF and TBBPA, coloured by phylum (*n* = 3 biological replicates). The legend identifies individual species using colour intensity. **g**, Comparison of monoculture versus community susceptibility to BPAF. The data represent ratios of absolute abundance (16S sequencing read counts normalized to the community optical density) between cultures treated with BPAF and the DMSO control. The points are the average of three Com20 replicates, corresponding to species overlapping between monoculture and synthetic community experiments. Central lines, box limits and whiskers represent, median values, upper and lower quartiles, and 1.5× the interquartile range, respectively. **h**, LC–MS/MS measurements of BPAF in synthetic communities (Com20) show bioaccumulation (that is, depletion of BPAF in supernatant compared with whole culture) (*n* = 6 biological replicates; two-sided *t*-test). The data are presented as means ± s.d.

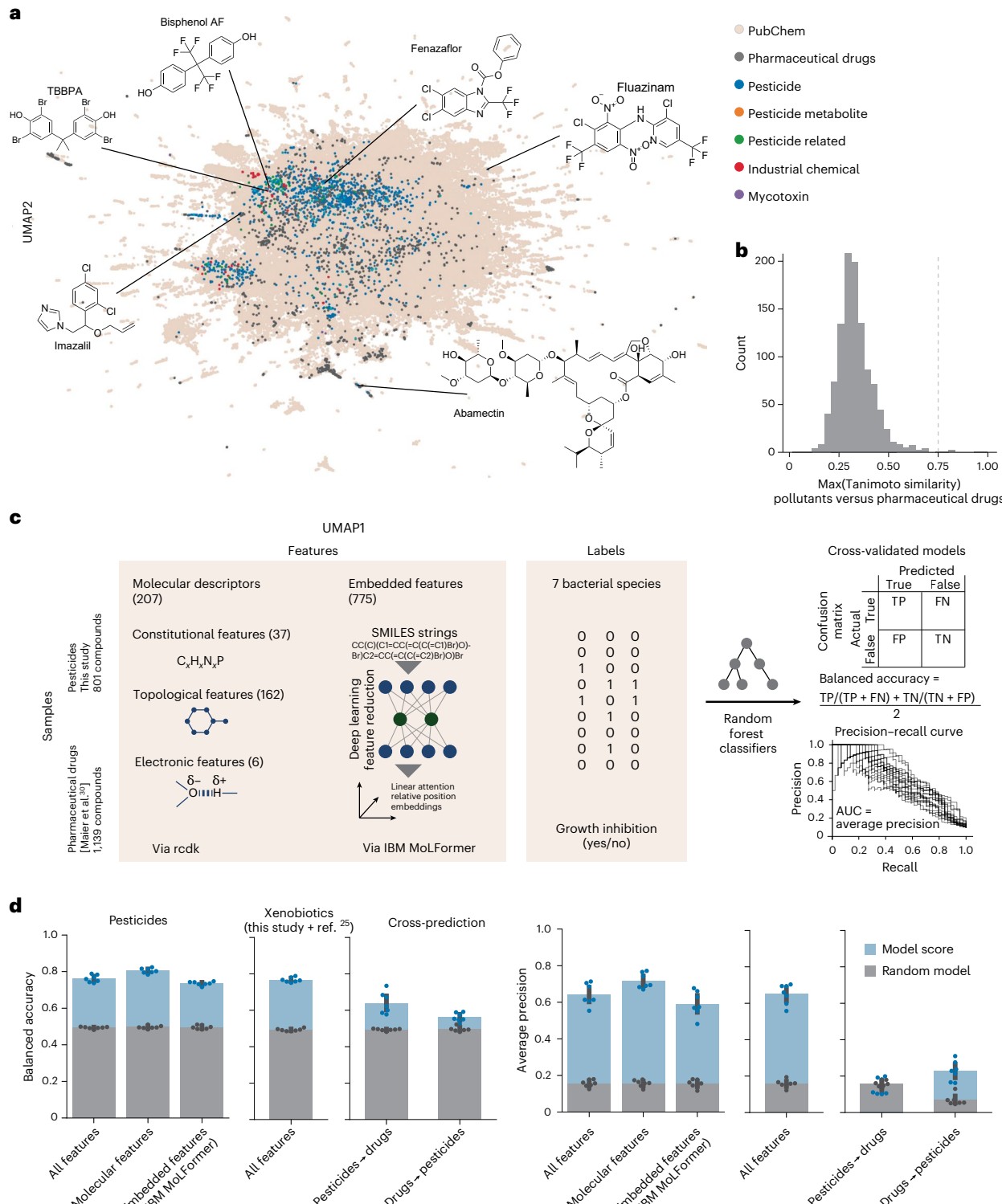

**Fig. 3 | Expanded chemical space of xenobiotic–bacterial growth interactions enables toxicity predictions. a**, UMAP for ECFPs of pollutant library compounds from this study, previously screened pharmaceutical drugs[25] and a random selection of 250,000 compounds from PubChem. **b**, Distribution of similarities between the closest-matching pollutants from this study and drugs used by Maier et al.[25]. The two compound sets are structurally diverse and distinct, as only very few (six) compounds from the library used in this study had a similarity to a drug from ref. 25 of >0.75 (dashed line). Compounds shared between the two studies (*n* = 32; mostly veterinary drugs) were excluded from this analysis. **c**, Machine learning workflow to predict species-specific xenobiotic toxicity from molecular descriptors and embedded features. Random forest classifier models

were trained and evaluated using 20× cross-validation, using performance metrics appropriate for imbalanced datasets. **d**, Performance scores of models trained and tested on various feature and sample sets. Left, models trained on pesticides only (*n* = 801) using different feature sets (*x* axes). Middle, model trained on pesticides and pharmaceutical drugs from Maier et al.[25] (*n* = 1,940). Right, models trained on pesticides (this study) cannot predict the toxicity of pharmaceutical drugs[25] and vice versa. Each dot represents a species (mean of 20× cross-validation splits). Median random model performances are shown as overlaid bars. The error bars represent standard deviation and the dots represent performances obtained by randomly shuffling the labels. FN, false negative; FP, false positive; TN, true negative; TP, true positive.

propiconazole, perfluorooctanoic acid and perfluorononanoic acid (PFNA) were tested at ≥20 μM to ensure sufficient selective pressure (Methods). Cultures inoculated with the pooled library were grown to the early stationary phase, and Tn mutant selection was quantified using barcoded transposon sequencing (TnBarSeq) (Supplementary Data 6).

Gene hits were defined by an absolute log₂[fold change] in mutant abundance of >0.25 and $P_{adj}$ < 0.05, relative to end-point vehicle dimethyl sulfoxide (DMSO) or media controls. The abundance changes were not attributed to global differences in cell doublings (Extended Data Fig. 4a)[50]. Among the compounds tested at ≤20 μM, TBBPA, BPAF and closantel exhibited the strongest effects in library selection, whereas 500 μM PFNA showed the most hits overall. In contrast, 20 μM PFNA, 20 μM perfluorooctanoic acid and 50 μM glyphosate did not show significant hits, consistent with the lack of effect on growth at those concentrations (Extended Data Fig. 4b).

Closantel, an anthelmintic, caused the strongest selection, with >90% of Tn mutants carrying insertions across over 20 different positions of the coding and promoter regions of the gene *NQ542_01170* (Fig. 4a and Supplementary Data 7). This gene encodes a transcriptional regulator homologous to the efflux repressor *acrR* from *B. uniformis* and is co-located with *tolC* and RND-efflux genes, with conserved synteny[51]. Notably, *acrR* Tn mutants were also the top hit under TBBPA, a flame retardant, with >100-fold enrichment compared with the DMSO control, but were not enriched under any other pollutants (Fig. 4b,e).

To further characterize *acrR* Tn mutant phenotypes, we isolated strains with transposon insertions at seven distinct positions within *acrR* coding and promoter regions after growth on closantel or TBBPA. All Tn::*acrR* isolates showed resistance to both chemicals, regardless of the compound they were isolated on (Fig. 4c and Extended Data Fig. 5c). When further evaluating two Tn::*acrR* strains, we observed an eightfold increase in the minimum inhibitory concentration (MIC) for closantel compared with the wild-type strain. Consistent with previous reports linking loss of *acrR* function to ciprofloxacin resistance in diverse bacteria[52], we observed an up to 16-fold increase in the ciprofloxacin MIC for *P. merdae* Tn::*acrR* mutants compared with the wild type (Fig. 4d and Supplementary Data 8). Therefore, resistance to closantel and TBBPA can lead to cross-resistance to antibiotics such as ciprofloxacin.

The differential enrichment of *acrR* Tn mutants suggests a degree of specificity in the efflux of chemicals and/or that efflux alone is insufficient in tackling the other chemicals. However, some transporter Tn mutants showed broad pollutant sensitivity, indicating common pollutant tolerance mechanisms in *P. merdae*. For example, strains with insertions in genes of the *fadL* family of hydrophobic compound transporters (*NQ542_07320* and *NQ542_08455*) were prevalent negative fitness hits across seven xenobiotics, whereas insertion mutants of the RND multidrug efflux locus *NQ542_06240–50* were a significant hit across five chemicals (Extended Data Fig. 4c). Other common negative fitness hits were Tn mutants of the genes encoding components of the F0F1 ATP synthase complex (Fig. 4e), the loss of function of which has been associated with increased sensitivity to polymyxin antibiotics under anaerobic conditions[53]. This is consistent with bisphenol exposure leading to overexpression of F0F1 ATP synthase subunits in *B. thetaiotaomicron*[23]. The strains with Tn insertions in a gene encoding a putative lipid A phosphoethanolamine transferase of the *eptA* gene family (*NQ542_04035*) showed increased sensitivity in half of the tested xenobiotics. This enzyme class often mediates polymyxin resistance via modifications in membrane polarity[54]. Overall, genes implicated in efflux and membrane homeostasis modulated *P. merdae* survival following diverse xenobiotic challenges.

To assess whether the involvement of transporters in xenobiotic response is broadly conserved in the order Bacteroidales, we evaluated mutants of *B. thetaiotaomicron*, a member of a family distant to that of *P. merdae*[55]. We tested closantel, TBBPA, PFNA and BPAF on a curated subset of 185 transporter insertion mutants from the *B. thetaiotaomicron* arrayed transposon loss-of-function mutant library[56]. Growth was monitored over 24 h and we evaluated the response of each Tn insertion mutant by calculating the normalized relative AUC (nrAUC), comparing the xenobiotic treatment with the DMSO control (Extended Data Fig. 6a and Supplementary Data 9). The strongest growth reduction for closantel was observed at 2 and 10 μM for the Tn mutant of *BT2687* encoding an RND-efflux pump (Fig. 4f and Extended Data Fig. 6b). This gene is part of an *acrR*-RND-*tolC* locus homologous to the *P. merdae* Tn::*acrR* closantel hit, suggesting a conserved efflux response (Extended Data Fig. 6c). Additional Tn mutants with conditional growth reduction on closantel included the co-localized genes *BT0562* and *BT0563*, which encode multidrug ABC transporter genes. Loss-of-function Tn mutants of *BT2687*, *BT0562* and *BT0563* have previously been linked to increased susceptibility to antibiotics and biocides in *B. thetaiotaomicron*[57]. Exposure to 250 and 100 μM PFNA inhibited the Tn mutants with insertions at *BT3337* and *BT3338*, which encode an RND-efflux pump needed for biofilm formation[58], and tolerance to fusidic acid, chlorpromazine and thioridazine[57] (Fig. 4f). The top homology and synteny matches for *BT3337* and *BT3338* in *P. merdae* are the RND-encoding loci *NQ542_09525–35*, which were a specific negative fitness hit for PFNA in the pooled screen (Extended Data Fig. 6d). As PFNA can be bioaccumulated by several Bacteroidales[42], we assessed whether the phenotypes of Tn::*BT3337* and Tn::*BT3338* could be caused by increased compound uptake. Testing PFNA bioaccumulation across

**Fig. 4 | Chemical genetic screens in Bacteroidales reveal genes that modulate susceptibility and resistance to xenobiotics. a**, Overview of the closantel experiment and a representative plot of TnBarSeq insertion at the *P. merdae acrR* efflux regulator locus (*NQ542_01170*). The arrows show genes to scale, the black track shows the TA dinucleotide insertion sites and the vertical lines indicate insertion read counts. **b**, Relative abundances of *acrR* locus mutants under 2.5 μM closantel and 20 μM TBBPA, compared with the DMSO control. The bars represent mean values and data points are also shown (n = 2 for closantel and TBBPA; n = 3 for DMSO). **c**, Growth curves under 2 μM closantel of *acrR* coding and promoter region mutants (n = 7 independent insertions), the wild type (WT; n = 6) and a transposon insertion control (n = 2). The original isolation conditions of *acrR* mutants are shown as dashed (TBBPA) and solid lines (closantel). Results for the DMSO control are provided in Extended Data Fig. 5c. **d**, Dose–response curves under closantel and ciprofloxacin of *acrR* mutants and control strains. The points represent the OD₆₀₀ value at 24 h, taken from three independent experiments with technical duplicates. The lines connect median OD₆₀₀ values per genotype. The highest ciprofloxacin dose of 603.6 μM corresponds to 200 μg ml⁻¹. **e**, Heatmap of the top TnBarSeq gene hits. The red and blue scale indicates the log₂[fold change] in mutant abundance relative to that of the end-point control (red and blue represent enrichment and depletion, respectively). Statistical significance was assessed using TRANSIT resampling (permutation based, two sided and Benjamini–Hochberg adjusted). Values of $P_{adj}$ < 0.05 are marked with an asterisk. The functional categories are also colour coded. BCGC, branched α-galactosylceramide; BSCFA, branched short-chain fatty acids. **f**, Growth effects of xenobiotics on arrayed transporter mutants of *B. thetaiotaomicron*. The x axis shows the normalized relative area under the curve (nrAUC, the average rAUC of a mutant in xenobiotic perturbation, divided by the average rAUC in vehicle control), as described in the Methods (n = 185 mutants; two replicates). Hits are defined by an absolute effect on growth of >20% and P < 0.05. **g**, Representative plot of *P. merdae* TnBarSeq insertion in the inositol lipid biosynthesis gene cluster under 500 μM PFNA or DMSO control. Homology with *B. thetaiotaomicron* is shown using clinker. **h**, Growth curves of isolated *P. merdae* transposon mutants with mutation of a putative phosphatidylinositol phosphate phosphatase (PIPPh)-encoding gene (*NQ542_07540*), Results are shown for the wild type and transposon insertion control (Tn::*acrR*) under 250 μM PFNA. Curves for two independent insertion strains are shown, each with two replicates. Tn, transposon.

concentrations with LC–MS/MS showed no differences (Extended Data Fig. 6e,f and Supplementary Data 11), indicating other membrane-related mechanisms. Overall, the shared responses between *P. merdae* and *B. thetaiotaomicron* support conserved mechanisms of pollutant and antibiotic tolerance across Bacteroidales.

## Chemical selection for the loss of beneficial metabolite pathways

*P. merdae* TnBarSeq insertion mutant gene hits showed enrichment in various Kyoto Encyclopedia of Genes and Genomes (KEGG) metabolic pathways for most (seven) of the compounds tested. TBBPA

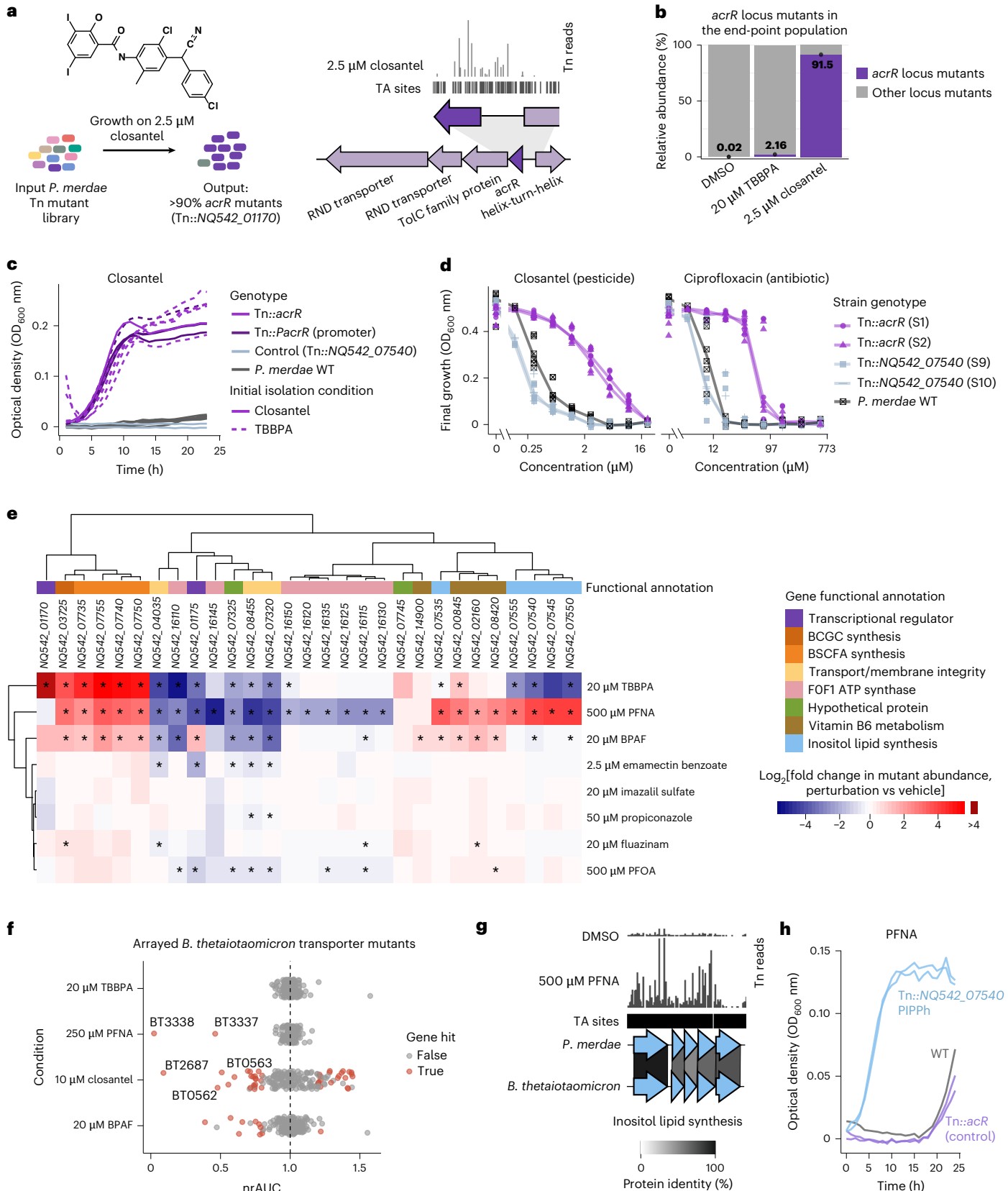

20-μM selection showed a significant enrichment of Tn mutants mapping in the branched-chain amino acid (BCAA) degradation pathway (Extended Data Fig. 4d). The second top locus for positive selection under TBBPA was the *porA* catabolic gene cluster, containing four genes with Tn insertions with >30-fold enrichment (Extended Data Fig. 4e). This pathway is involved in the degradation of BCAAs into branched short-chain fatty acids[59]. Qiao et al.[59] reported on a *P. merdae porA* mutant (with deletion of the *NQ542_07740* gene, encoding pyruvate:ferredoxin oxidoreductase) with loss of its beneficial probiotic property of preventing atherosclerosis in mice. PorA is present in several gut bacteria and conserved across Gram-positive and -negative bacteria, including *Clostridium sporogenes*, where it contributes to immunomodulation[60,61]. The next top hit for positive selection under TBBPA was a gene encoding a putative BCAA aminotransferase (greater than tenfold enrichment; *NQ542_03725*). This gene encodes a close homologue of *B. fragilis* BF9343-3671 (74% identity; 100% coverage), whose knockout mutant stops producing immunomodulatory branched-chain α-galactosylceramide lipids, impacting colonic natural killer T cell regulation in monocolonized mice[62]. Besides TBBPA, *NQ542_03725* mutants also show enrichment under 500 μM PFNA, 20 μM BPAF and 20 μM fluazinam, and the *porA* gene cluster is also selected under 500 μM PFNA and 20 μM BPAF (Fig. 4e). If these xenobiotics select for the loss of bacterial biosynthesis of BCAA-derived beneficial metabolites such as branched short-chain fatty acids and α-galactosylceramides in vivo, they could further impact human health via gut–heart or gut–immune axes.

Loss-of-function Tn insertion mutants of the secondary metabolism genes *NQ542_07535–55* were positively selected under 500 μM PFNA challenge (-16-fold enrichment), but were negative fitness hits under 20 μM TBBPA (Fig. 4e). These genes were identified as being part of a putative inositol lipid biosynthetic gene cluster by homology to *B. thetaiotaomicron* genes (*BT_1522–6*) (Fig. 4g). This gene cluster is widespread in *Bacteroidetes*, and loss of inositol lipid production is associated with changes in the cell capsule and increased resistance to antimicrobial peptides[63]. To validate the phenotype, we isolated two *P. merdae* mutants (with transposon insertions at two locations of the gene *NQ542_07540*, which encodes putative phosphatidylinositol phosphate phosphatase) after enrichment in 500 μM PFNA. These strains showed increased resistance to PFNA and hypersensitivity to TBBPA at a range of concentrations, in agreement with pooled fitness (Fig. 4h and Extended Data Fig. 5d,e). To evaluate whether the increased resistance to PFNA was due to differences in compound uptake, we tested PFNA bioaccumulation, but no growth-independent differences were seen (Extended Data Fig. 5f), indicating the accumulation-independent function of inositol lipid metabolism in PFNA resistance. In summary, our genetic screen suggests that loss-of-function mutations affecting accessory metabolic pathways might be selected for under xenobiotic exposure, potentially altering the metabolic output of the gut microbiota.

## Discussion

Our study uncovered 588 inhibitory interactions between 168 chemicals of concern and common human gut bacteria. Most of these chemicals had not been previously reported to have antibacterial properties. Fungicides and industrial chemicals showed the largest impact, with around 30% exhibiting anti-gut-bacterial activity. For example, although conazole fungicides inhibit ergosterol synthesis, a biochemical pathway unique to fungi, they inhibited some Firmicutes—prominent members of the human gut microbiota. The extent of interactions identified in our study, together with the immense genetic diversity of the gut microbiome[64], suggest that the microbiome may harbour vastly more potential xenobiotic targets than was previously appreciated.

The prevalence of the inhibitory activities of chemical pollutants revealed by our screen, together with those of human-targeted drugs[25,40], challenge the categorical labelling of chemicals into antibiotics, human-targeted drugs, industrial chemicals and pesticides. The antibacterial activity of many xenobiotics is currently not accounted for in assessments of their potential adverse impacts, and toxicity studies are typically set up in ways that are incompatible with capturing gut microbiota effects (Extended Data Fig. 7). To address this gap, toxicology assessment could incorporate tests in single gut bacteria, synthetic communities and gut-on-chip models[65]. Nevertheless, the vast numbers of chemicals in use and the fast pace at which new molecules are brought to market will make it difficult to implement meaningful experimental assessment for all compounds. Towards tackling this, our study demonstrates the feasibility of computational toxicity predictions that can be used at scale and for safe-by-design product development. This proof-of-concept was possible due to the here-expanded coverage of the chemical space, underscoring the need for standardized chemical–bacteria screens.

Our genetic screens using non-model, yet prevalent, gut bacteria provide mechanistic insights into how bacteria maintain fitness under chemical stressors and indicate potential molecular targets. We observe compound-dependent reliance on particular efflux pumps and their regulators for survival in the presence of pollutants. Regulation of efflux pumps stands out as a mechanism conserved between *P. merdae* and *B. thetaiotaomicron*, spanning different Bacteroidales families. This remarkable conservation, together with the previous observation from *E. coli* genetic screens showing the importance of efflux against human-targeted drugs[25,40,66], reinforces the key role of efflux as a general chemical defence mechanism. This mechanistically supports the idea of a general resistant phenotype and that xenobiotics could lead to collateral resistance to antibiotics[67]. In our study, genetic selection against two persistent pollutants—the flame retardant TBBPA and the anthelmintic closantel—resulted in resistance to the antibiotic ciprofloxacin.

Genetic selection in *P. merdae* is enriched for catabolic and biosynthetic genes, suggesting that metabolism plays a fundamental role in response to xenobiotic stress. This enrichment is reminiscent of the prevalence of metabolism-related gene hits observed for non-antibiotic drug screens in *E. coli*[40]. In our study, loss-of-function mutations in genes coding for enzymes involved in producing secondary metabolites with known impact on human health provided a growth advantage under several xenobiotics. This raises the possibility that exposure to chemical pollutants could impact the selection landscape in the gut, leading to metabolic alterations in microbiome–host interaction pathways.

Are our results relevant for real-world exposure scenarios in humans? Many of the compounds in our study (for example, melamine and mepiquat) have been found in micromolar concentrations in human blood[13], implying similar or higher concentrations in the gastrointestinal tract. The concentrations for individuals living in polluted areas and with high occupational exposure (for example, from unprotected pesticide spraying, which is common in much of the developing world), are likely to be much higher. Yet, there are no cohort data available that track both chemical exposure and microbiome dynamics. To enable a large screen and comparison with previous datasets, we tested only a single concentration for most compounds. Although these results clearly highlight compound classes with antibacterial activity, such as fungicides, testing at a single concentration limits the assessment of hazard and risk levels for individual chemicals. Furthermore, the strains used in our study do not span the full diversity of gut bacterial strains and additional screening will be necessary to gain insights into species- and strain-level variation in chemical sensitivity and resistance. Our results therefore currently cannot be directly translated to exposure effects on the gut microbiota in vivo.

Going forward, we envisage that our pollutant–bacteria interaction data will encourage the collection of exposure data in microbiome studies, similar to how in vitro drug–bacteria screens accelerated the recording of drug usage in cohort studies[25,31]. Nevertheless, in vivo

assessment of the impact of chemicals exposure is likely to be difficult due to complex interactions between the microbiota and diet, medications and other lifestyle factors. Genetic and physiological insights gained from our study will therefore help to bridge in vitro and in vivo datasets. The chemical–bacteria interaction map and identified genetic determinants of resistance could be used to guide the selection of compounds and molecular effects to monitor in future microbiome studies.

## Methods

### Chemical library

An arrayed library of pesticides, pesticide metabolites and pesticide-related compounds was obtained from the Chemical Biology Core Facility at the European Molecular Biology Laboratory (EMBL; Heidelberg, Germany). Original compound stocks were ordered from the collection of PESTANAL standards by Merck/Sigma–Aldrich, with catalogue numbers and Chemical Abstracts Service identifiers listed in Supplementary Data 1. An additional plate containing industrial chemicals and mycotoxins was prepared in house. Compounds were dissolved in DMSO at a concentration of 10 mM. The combined library was arranged on 12 96-well plates, which included eight DMSO controls interspersed within the plate. One unique well position of each plate contained a dye (Trypan Blue), serving as a footprint to enable the detection and prevention of plate mix ups and rotation. A working stock concentrated at 2 mM in DMSO was prepared and aliquoted. Libraries were stored at −80 °C.

Extensive metadata were compiled from multiple sources. PubChem Compound Identifiers (CIDs) were primarily retrieved automatically with compound name searches using the PUG–REST interface, and missing CIDs were added manually. PubChem metadata were then retrieved based on CIDs. PubMed hit counts were retrieved in April 2022 using the National Center for Biotechnology Information E-Utilities application programming interface. Compound names were semi-automatically matched to the Pesticide Compendium of the British Crop Production Council to map hierarchical target classifications, and unmatched entries (for example, pesticide metabolites and non-pesticides) were manually curated. Food and Drug Administration and European Food Safety Authority 2018 pesticide monitoring reports were used to annotate detection in common foods and maximum residue level exceedances.

### Bacterial strains and growth conditions

Details of bacterial strains, including their accession numbers in common culture collections and alternative species names, are listed in Supplementary Data 2. All bacteria were grown under anaerobic conditions in a polyvinyl chamber (Coy Laboratory Products) filled with 2% hydrogen and 12% carbon dioxide in nitrogen. The chamber was equipped with a palladium catalyst system for oxygen removal, a dehumidifier and a hydrogen sulfide removal column. Bacteria were grown at 37 °C in modified Gifu anaerobic medium (mGAM; HyServe, Germany, produced by Nissui Pharmaceuticals), prepared according to the manufacturer's instructions and sterilized by autoclaving. Bacterial strains were chosen based on their abundance and prevalence in the healthy human gut[33].

### High-throughput screening and MIC determination

**Growth screening.** Assay plates were prepared by first diluting the frozen pesticide library aliquots (2 mM in DMSO) in mGAM within a master deep-well plate from which 50 μl measures were aliquoted into clear, round-bottom, polystyrene untreated microplates (3795; Corning). Assay plates were then placed in the anaerobic chamber overnight to become anoxic. All liquid handling was performed using a Biomek i7 Automated Workstation (Beckman Coulter).

Each strain was measured in biological triplicates from independent cultures inoculated on separate days. Bacteria were grown for one or two days (depending on the growth rate) in 10 ml media, inoculated directly from frozen glycerol stocks. Cultures were then diluted 100-fold and incubated again for the same amount of time. Optical densities measured at a wavelength of 600 nm ($OD_{600}$) were determined and a dilution with an $OD_{600}$ of 0.1 was prepared. We then added 50 μl of this dilution to each well of the previously prepared assay plates, resulting in a starting $OD_{600}$ of 0.05 and a compound concentration of 20 μM (1% DMSO). Assay plates were sealed with a gas-permeable membrane (Breathe-Easy; Z380059; Merck), which was additionally pierced with a syringe to prevent the build-up of gas, which otherwise occurs for some species. Plates were stacked without lids and the $OD_{600}$ was recorded every hour for 24 h using a stacker (BioStack 4; Agilent BioTek) and plate reader (Epoch 2; Agilent BioTek).

**Statistical analysis of growth.** Data analysis was performed in R version 4.2.2 and RStudio version 1.3.1093. First, for each growth curve, the minimum $OD_{600}$ value was set to 0. Then, the raw AUC was calculated for each well using the bayestestR package and area_under_curve function. Further processing of growth curves was done by plate. For the pesticide library, AUC values were normalized by the median AUC of all control wells (DMSO control) on the respective plate. For the industrial chemicals and mycotoxin library, AUCs were normalized by the median row and column values (to correct for edge effects). Then, for both screens, a z score was calculated as follows: (raw AUC − control median)/control s.d. Each z score was converted to its respective P value and the P value was false discovery rate corrected for the number of compounds tested (n = 1,076). Significance analyses were performed separately for each replicate. A replicate was considered to be significant for growth inhibition if $P_{adj} < 0.05$ and there was a >20% reduction in the normalized AUC. A compound was considered to significantly inhibit a specific strain if at least two out of three replicates were significant.

**MIC determination.** For the validation experiments and determination of the MIC, fresh vials were used, with the same catalogue numbers as in Supplementary Data 1. The results from the MIC screen were prepared and analysed in the same way as those from the high-throughput screen. The MIC was defined as the lowest concentration for which the normalized AUC dropped below 0.1.

### Compound testing in synthetic community Com20

Com20 (ref. 38) was inoculated from a frozen ready-to-use stock (equal member ratio; 10% glycerol containing palladium black; 520810; Sigma–Aldrich) into 5 ml mGAM under anaerobic conditions for two 1:100 overnight passages. Com20 was dispensed into wells of 96-well plates containing 500 μl media with xenobiotic compounds, at a starting $OD_{578}$ of 0.01, sealed with AeraSeal membranes (A9224; Sigma–Aldrich) and anaerobically incubated at 37 °C for 24 h. Pellets from 300 μl of culture were frozen for 16S ribosomal RNA analysis and the $OD_{578}$ was measured as a biomass proxy.

DNA was extracted using a DNeasy UltraClean 96 Microbial Kit (10196-4; Qiagen) and quantified with a Qubit dsDNA BR/HS Assay Kit (Thermo Fisher Scientific). Library preparation and 16S sequencing were conducted at the Next Generation Sequencing Competence Center (Tübingen, Germany) using a two-step PCR protocol. The V4 region was amplified using 515F/806R primers with KAPA HiFi HotStart ReadyMix (Roche) and indexed with unique dual-index barcodes (IDT for Illumina DNA/RNA UD Indexes (Tagmentation)). Libraries were pooled equimolarly, normalized to 10 pM and sequenced on an Illumina MiSeq system (v2 kit; 2 × 250 bp; 20% PhiX).

Reads were processed with the DADA2 package (version 1.21.053) in R (version 4.2.0), using the standard operating procedure (https://benjjneb.github.io/dada2/bigdata.html) with the parameters trimLeft: 23, 24; truncLen: 240, 200; maxEE: 2, 2; truncQ: 11, and only retaining merged reads of 250–256 base pairs. Genus-level taxonomy was first assigned against a curated DADA2-formatted database derived from

the genome taxonomy database (release R06-RS20253)[68] and amplicon sequence variants from expected Com20 genera were then further classified at the species level using full-length 16S ribosomal RNA sequences of all 20 members. Classification was performed with the R package DECIPHER (version 2.24.054) at ≥98% identity.

## Toxicokinetic modelling

**Calculation of estimated administered doses.** To assess the effects of chemical pollutants on gut microbiota and extrapolate in vitro bioactivity data to potential human health risks, we focused on 168 active chemicals. In vitro growth screening concentrations were used to compute estimated administered doses via the IVIVE tool from the Integrated Chemical Environment (https://ice.ntp.niehs.nih.gov/), leveraging the Environmental Protection Agency's httk R package[69] and in-house NICEATM code[70]. IVIVE calculations determined gut concentrations corresponding to bioactivity levels (MICs from growth screening assays). Essential parameters included the toxicokinetic model type, chemical-specific parameters ($\log[P]$, molecular weight, pKa, intrinsic clearance, fraction unbound and tissue partition coefficient), dosing scenarios and exposure routes. OPERA[71] provided parameters for 152 of the 168 chemicals; the reminder were empirically available. The physiologically based toxicokinetic model utilizes multiple compartments for various organs and tissue with perfusion-rate-limited kinetics, simulating elimination primarily via hepatic and renal routes, under a 30-day oral exposure scenario with 24-h intervals. Compartments include the gut, liver and remaining body, simulating dynamic tissue and plasma concentrations for both oral and intravenous exposures[72].

**Curation of in vivo point-of-departure data.** The US Environmental Protection Agency's Toxicity Values Database served as the primary source for condensed in vivo point-of-departure (POD) data, encompassing human health references and animal toxicity values from various studies via the CompTox Chemicals Dashboard (https://comptox.epa.gov/dashboard/). Filters applied to the data available for 126 chemicals included: exposures in mg kg$^{-1}$ bodyweight or mg kg$^{-1}$ bodyweight per day (or convertible units); species-specific conversion factors for rats, mice, dogs and rabbits; the study types sub-chronic, subacute and subchronic; and the POD types lowest observed effect level, lowest adverse effect level, no observed effect level and no observed adverse effect level.

## Comparison of compound features

The SMILES strings for all compounds in PubChem were downloaded, from which 250,000 were selected at random. ECFPs were generated from SMILES strings for all library compounds, previously screened pharmaceutical drugs[73,74] and PubChem compounds using ChemPlot[75]. The ECFPs were then embedded in a two-dimensional representation using UMAP[76], with the number of neighbours and minimum distance set to 80 and 0.35, respectively. The Jaccard/Tanimoto index between compounds was calculated using the cdist function from SciPy[77]. The distributions for the maximum Tanimoto similarity index for each PubChem compound for the pharmaceutical drugs and combined pharmaceutical drugs and pollutants libraries were compared using a two-sided Kolmogorov–Smirnov test.

## Machine learning

Molecular descriptor features were generated with the rdck R package. Embedded features were obtained from a pre-trained version of MoLFormer[46], which takes the SMILES strings as input and outputs a vector with 768 dimensions (https://huggingface.co/ibm-research/MoLFormer-XL-both-10pct). The deterministic flag was enabled to make sure that the model always gives the same output. Machine learning was implemented using scikit-learn version 1.2.2. The data were split into train and test sets in an 8:2 ratio with stratification to ensure equal

hit frequencies in both sets. Twentyfold cross-validation was applied. RandomForestClassifier was used with the options n_estimators=500, max_features='sqrt' and max_depth=None to predict probabilities, which were then thresholded using the hit frequency of the training set[47]. The feature_importances_ attribute was used to access feature importance values.

## TnBarSeq assay and analysis

The dense transposon mutant library of *P. merdae* ATCC 43184 (ref. 49) was used for TnBarSeq mapped to the annotated genome CP102286.1 (GenBank). Two vials of the pooled stock transposon library were used to inoculate mGAM with 30 μg ml$^{-1}$ erythromycin and grown until the mid-exponential phase. This culture was used to inoculate 1 ml cultures at $OD_{600}$ = 0.02 in a 2-ml deep-well plate, with duplicates for each xenobiotic and triplicates for the respective control media with 1% DMSO, 0.2% DMSO or plain media (xenobiotic concentrations are provided in Supplementary Data 6). The growth of the library was tracked by monitoring hourly a 100-μl aliquot of the library (as described in the section 'Growth screening'), and growth curves were analysed using the Growthcurver package in R[78]. The transposon library cultures were grown to the early stationary phase (16 h for all samples except 2.5 μM closantel, which was grown to 48 h). Genomic DNA was extracted using a MagMAX Microbiome Ultra Nucleic Acid Isolation Kit. The transposon barcodes were amplified by PCR with Q5 Hot Start High-Fidelity 2X (New England Biolabs) using indexed primers (the sequences are provided in Supplementary Data 10), and sequenced with an Illumina NextSeq (single-end sequencing with a read length of 75 nucleotides). Barcodes were counted with 2FAST2Q[79] and transposon insertion counts per sample were inputted as wiggle files to TRANSIT, where they were analysed against their respective control using reads in the central 80% of coding regions and normalized using the trimmed total read count, followed by resampling of 20,000 permutations[80] and two-sided statistical testing with Benjamini–Hochberg correction (Supplementary Data 6). Gene hits were considered significant at $P_{adj} < 0.05$ and $|\log_2[\text{fold change}]| > 0.25$. Post-hoc doublings bias correction was performed in R using the EEL_correction script[50]. KEGG terms were assigned with EggNOG Mapper[81] and KEGG term enrichment was performed using a Fisher's exact test in R. Protein homology was mapped with clinker to *B. thetaiotaomicron*[82]; membrane protein families were annotated using TransAAP (hosted by TransportDB)[83] and the Conserved Domains Database at the National Center for Biotechnology Information; and additional relevant homologues were identified using PaperBLAST[84].

## Characterization of mutant strains of *P. merdae* isolates

To enrich for positively selected mutants, the *P. merdae* transposon mutant library was cultured in 20 μM TBBPA, 500 μM PFNA or 2 μM closantel for two passages. Cultures were inoculated at an $OD_{600}$ of 0.02 in 5 ml media in 15-ml tubes and incubated anaerobically for 24 h of growth, then passaged one in 50 into 5 ml fresh media and incubated for an additional 24 h. Following enrichment, cultures were streaked to isolation on anaerobic mGAM agar supplemented with 10 μg ml$^{-1}$ erythromycin, and individual colonies were picked and glycerol stocked. Transposon barcodes of the isolated colonies were identified by Sanger sequencing (Supplementary Data 7) and validated using locus-specific diagnostic PCR (Extended Data Fig. 5a,b; the oligonucleotides are provided in Supplementary Data 10). Growth of the mutants under chemical perturbation was tracked as in the section 'Growth screening' and analysed using the Growthcurver package in R[78].

For MIC testing, mutant strains were passaged twice for 24 h in anaerobic mGAM before being used to inoculate, at an initial $OD_{600}$ of 0.025, 100-μl cultures with twofold dilution series of ciprofloxacin, closantel or DMSO control. All plates contained empty-well controls and no compound control. Growth was tracked for 24 h, and the $OD_{600}$ at 24 h was used to determine the MIC. The MIC was defined as

the lowest concentration for which the blanked-normalized $OD_{600}$ was lower than a cut-off value of <0.1 (>90% inhibition). Three independent replicates were performed on different days, with two technical replicates per strain. The results are provided in Supplementary Data 8.

## Growth screen with arrayed transporter mutants of *B. thetaiotaomicron*

We assembled a subset of the *B. thetaiotaomicron* arrayed mutant library[56], selecting 185 gene mutants annotated as putative transporters. The genes were selected based on annotations from TransAAP[83] or if they included in their functional gene annotation the terms pump, efflux or transporter (Supplementary Data 9). Mutants were re-arrayed in three 96-well plates, each containing empty-well controls.

Inoculum library plates were grown anaerobically in mGAM from glycerol stocks or a first-passage plate for 24 h. Test plates of each condition were prepared at 2× concentration for 20 µM BPAF, 20 µM TBBPA, 2 and 10 µM closantel, 250 µM PFNA and their respective vehicle controls (0.4 or 1.0% DMSO) and left overnight inside the anaerobic chamber to equilibrate. Plates were inoculated by passaging library inoculum at 1:100 for each test condition in two technical replicates. Plates were sealed with gas-permeable membranes and growth was monitored over 24 h, as described in the section 'High-throughput screening and MIC determination'.

Growth curves were fitted using the Growthcurver package in R[78] and the AUC fit was calculated (Supplementary Data 9). One mutant strain that failed growth in the DMSO control was removed from further analysis. To identify mutants whose growth was differentially impacted by xenobiotic treatment, relative AUC values (rAUC) of each mutant were calculated as rAUC = (AUC)/(median AUC of mutants of the plate), assuming a neutral phenotype for most mutants. The average rAUC per technical replicate of mutant strains was calculated. To calculate the nrAUC for each xenobiotic condition, the average rAUC for the mutant gene under the xenobiotic was divided by the average rAUC for the corresponding DMSO control. A Welch's *t*-test was performed and *P* values were corrected for multiple testing using the Benjamini–Hochberg method. Gene mutant hits with conditional growth effects were defined as those with *P* < 0.05 and an nrAUC value corresponding to a 20% increase or decrease.

## Bioaccumulation experiments

Bioaccumulation of BPAF was assessed by reverse-phase LC−MS/MS, as described previously[42]. Com20 cultures were grown anaerobically at 37 °C in mGAM with 1% DMSO and 20 µM BPAF, with controls lacking either BPAF or bacterial cells. After 24 h, at the stationary phase, 800-µl whole-culture samples were mixed and taken, then the cultures were centrifuged (10 min at 3,200*g* and room temperature) and 800 µl of the supernatant was collected (the supernatant sample). The rest of the supernatant was discarded, the pellet was resuspended in 8 ml water and 800 µl transferred to a fresh tube (cell sample). All samples were stored at −80 °C. For LC−MS/MS, samples were thawed and mixed, then 160 µl extracted with 240 µl organic solvent (1:1 MeOH:ACN + 0.1% formic acid with 20 µM amoxicillin, caffeine, ibuprofen and donepezil as internal standards), incubated for 30 min at −20 °C and cleared by centrifugation (5 min at 10,000*g* and 4 °C). The supernatants were then transferred to fresh vials. A standard curve was prepared by serially diluting BPAF in the appropriate matrix (untreated supernatant or cell sample extracts of Com20 cultures from the same experiment). Samples (0.5 µl) were analysed on an Agilent 1290 Infinity II chromatography system coupled to an Agilent 6470 Triple Quadrupole (QQQ) system with a Jet Stream ion source operated in dynamic multiple reaction monitoring mode with a cycle time of 500 ms. The analytical column (Agilent Poroshell 120 EC-C18; 2.1 × 100 mm; 1.9 µm) was operated at 40 °C and a constant flow rate of 0.3 ml min⁻¹. Initial conditions were 65% buffer A (water + 0.1% formic acid + 10 mM

ammonium formate) and 35% buffer B (9:1 acetonitrile:water + 0.1% formic acid + 10 mM ammonium formate). The gradient was as follows: 0 min: 35% B; 0.5 min: 35% B; 4.5 min: 100% B; 5.5 min: stop; post-time: 1.5 min. The source parameters were as follows: gas temperature: 300 °C; gas flow: 10 l min⁻¹; nebulizer: 50 psi; sheath gas temperature: 300 °C; sheath gas flow: 11 l min⁻¹; capillary: 3,500 V (positive) and 3,000 V (negative); nozzle voltage: 2,000 V (positive) and 500 V (negative). BPAF was monitored in negative mode, using a precursor *m/z* of 335 and fragment *m/z* of 265 and 69 (500-ms cycle time). The data were analysed using MassHunter Workstation Quantitative Analysis for QQQ version 10.1.

Bioaccumulation of PFNA was performed as previously described[42]. Strains were inoculated at an $OD_{600}$ of 0.025 in 1-ml cultures with varying PFNA concentrations, plus DMSO and empty-well controls. A 100-µl sample was taken to track growth dynamics in parallel. After 20 h of anaerobic incubation, 80-µl whole-culture and supernatant aliquots were collected and stored at −80 °C. Samples were extracted with solvent mix (1:1 MeOH:ACN + 0.1% formic acid with 20 µM caffeine, ibuprofen and donepezil as the internal standard), followed by 2 h incubation at 4 °C and 30 min incubation at −20 °C, then centrifuged as above. Samples were injected at a 0.3-µl volume in the LC−MS/MS instrument described above. The analytical column (Agilent Poroshell 120 EC-C18; 2.1 × 50 mm; 1.9 µm) was operated at 40 °C with a constant flow rate of 0.8 ml min⁻¹. The initial buffer composition was 70% A (water + 0.1% formic acid) and 30% B (acetonitrile + 0.1% formic acid). The gradient was as follows: 0 min: 30% B; 0.05 min: 30% B; 0.5 min: 100% B; 1 min: 100% B; 1.05 min: 30% B; stop time: 1.5 min. The source parameters were identical to those described above. PFNA was monitored in negative mode, using a precursor *m/z* of 463 and fragment *m/z* of 417.9 and 219 (200 ms cycle time). The data were analysed using MassHunter Workstation Quantitative Analysis for QQQ version 10.1 and a custom R script, and the results are provided in Supplementary Data 11.

## Statistics and reproducibility

No randomization was used in the microbiological experiments, and data collection and analysis were not performed blind to the conditions of the experiments, as no group allocation was needed. No statistical methods were used to determine samples sizes; instead, they were chosen based on previous experience and according to those in the literature[25,56,57]. The data distribution was assumed to be normal, but this was not formally tested. Further statistics and reproducibility information are described in each section.

## Reporting summary

Further information on research design is available in the Nature Portfolio Reporting Summary linked to this article.

# Data availability

The data that support the findings of this study are included in the Supplementary Data. Growth curves, LC−MS/MS raw data and TnBarSeq mapped files used for TRANSIT, as well as the associated processing scripts and intermediate files are available via Mendeley Data (https://doi.org/10.17632/g7hy84t2r6.1). Raw sequencing reads of 16S sequencing of synthetic communities and TnBarSeq data are available from the European Nucleotide Archive (PRJEB97051 and PRJEB76605). Source data are provided with this paper.

# Code availability

Code for the generation and comparison of ECFPs is available from GitHub at https://github.com/MRCToxBioinformatics/Compound_bacteria_screen_chemical_space. Code for machine learning is available from GitHub at https://github.com/yeastbeast/PesticideToxicityPrediction. Code for the chemical–genetic experiments is available from GitHub at https://github.com/I-Roux/Pollutant-chemical-genetics.

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

## Acknowledgements

We thank O. Lemke (Charité – Universitätsmedizin Berlin) for advice on machine learning. We thank C. G. P. Voogdt and A. Typas (EMBL) for providing materials and guidance for the TnBarSeq library experiments. We thank the Chemical Biology Core Facility at EMBL for assembling the pesticide library. We thank EMBL GeneCore for support with the sequencing of the TnBarSeq samples and the Next Generation Sequencing Competence Center (Tübingen, Germany) for the 16S sequencing. We thank A. Shiver and K. C. Huang (Stanford University, USA) for the *B. thetaiotaomicron* mutant library, and T. Davis (Medical Research Council Toxicology Unit, University of Cambridge, UK) for help with mutant library re-arraying. Kiran R. Patil, I.R., S. Kamrad and S.B. received funding from the European Research Council under the European Union's Horizon 2020 research and innovation programme (grant 866028). Kiran R. Patil, R.G., L.F. and A.E.L. were supported by the UK Medical Research Council (project MC_UU_00025/11). L.M. was supported by the German Research Foundation (Cluster of Excellence EXC 2124; grant MA 8164/1-2) and European Research Council (gutMAP project; grant 101076967). N.C.K. and S. Krishna were supported in part by the Intramural Research Program of the National Institutes of Health, National Institute of Environmental Health Sciences.

## Author contributions

I.R., A.E.L., S. Kamrad and Kiran R. Patil conceived of and designed the study. A.E.L. and S. Kamrad performed and analysed the growth experiments. I.R. performed and analysed the chemical–genetic screens and genetic follow-up experiments. Kaustubh R. Patil and S. Kamrad performed machine learning analysis. A.G. and L.M. planned the microbial community experiments and A.G. performed them. T.S. performed the chemoinformatics analyses. S. Krishna and N.C.K. designed the toxicokinetic modelling and S. Krishna performed it. R.G. helped with the data analysis. D.R., L.F. and S.B. contributed to the microbiology experiments. S.B. advised on the microbiology experiments. S. Kamrad and Kiran R. Patil supervised the overall study. I.R., A.E.L., S. Kamrad and Kiran R. Patil wrote the paper. All authors commented on the paper.

## Competing interests

Kiran R. Patil and A.E.L. are co-founders of Cambiotics ApS. The remaining authors declare no competing interests.

## Additional information

**Extended data** is available for this paper at https://doi.org/10.1038/s41564-025-02182-6.

**Correspondence and requests for materials** should be addressed to Stephan Kamrad or Kiran R. Patil.

¹Medical Research Council Toxicology Unit, University of Cambridge, Cambridge, UK. ²Excellence Cluster 'Controlling Microbes to Fight Infections', Interfaculty Institute of Microbiology and Infection Medicine, M3 Research Center, University of Tübingen, Tübingen, Germany. ³Division of Translational Toxicology, National Institute of Environmental Health Sciences, Washington, NC, USA. ⁴Institute of Neuroscience and Medicine, Research Centre Jülich, Jülich, Germany. ⁵Institute of Systems Neuroscience, Medical Faculty, Heinrich Heine University Düsseldorf, Düsseldorf, Germany. ⁶Department of Biochemistry, University of Cambridge, Cambridge, UK. ⁷Present address: Cambiotics ApS, Copenhagen, Denmark. ⁸Present address: Medical Research Council Laboratory of Molecular Biology, Cambridge, UK. ⁹Present address: Physicians Committee for Responsible Medicine, Washington, DC, USA. ¹⁰Present address: Quadram Institute, Norwich, United Kingdom. ¹¹These authors contributed equally: Indra Roux, Anna E. Lindell. ✉e-mail: sk727@cam.ac.uk; kp533@cam.ac.uk

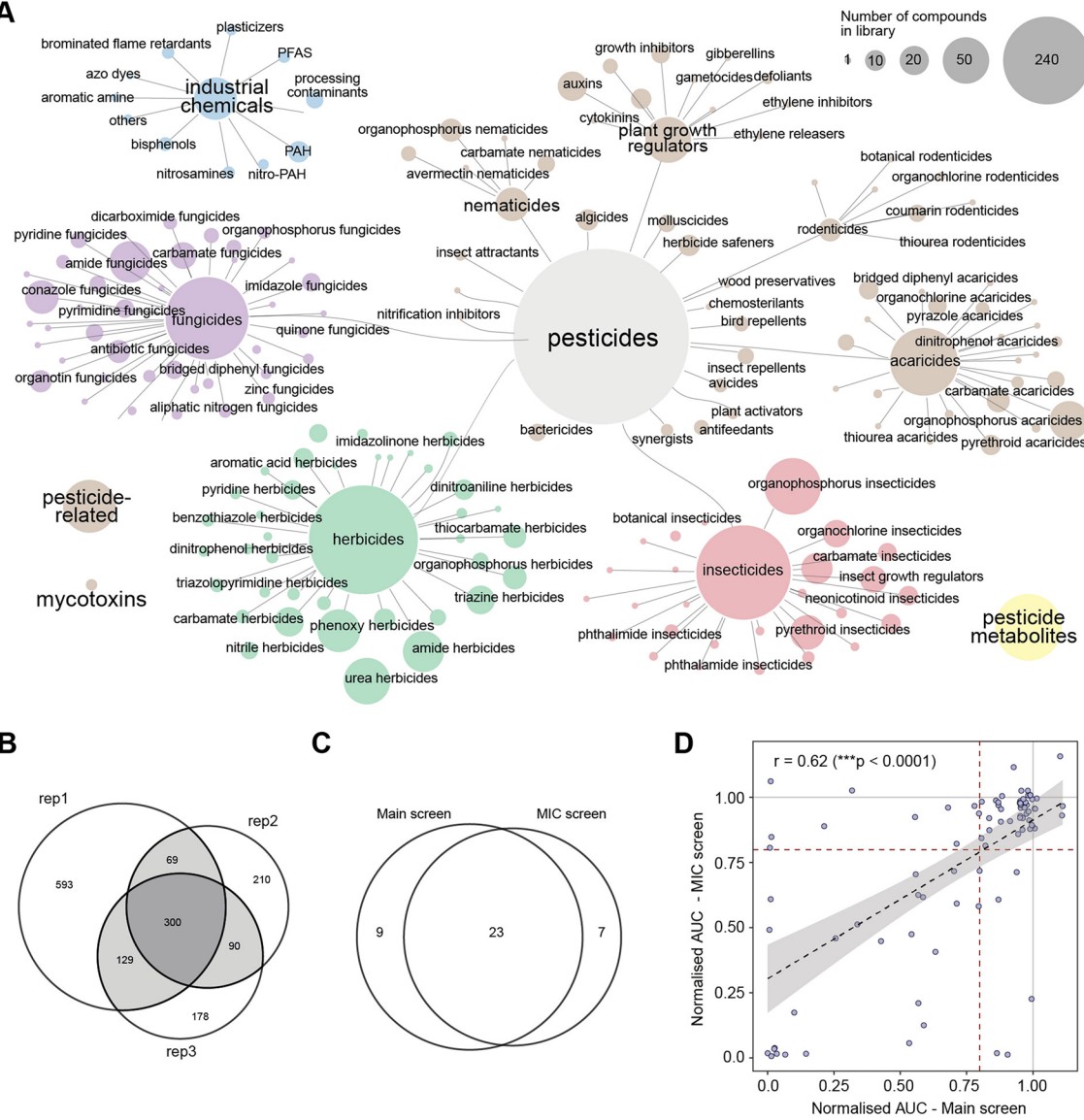

**Extended Data Fig. 1 | Compound library and screening method details.**
**A**) Network illustrating the number of library compounds targeted at different organisms and their chemical subclasses, based on the BCPC Pesticide Compendium. Out of 1076 compounds in total, 829 were pesticides, 119 were pesticide metabolites, 48 were industrial chemicals, 5 were mycotoxins and 76 were other, pesticide-related compounds such as synthetic precursors or compounds that might be found in commercial formulations. **B**) Venn diagram illustrating overlaps of significant interactions between n=3 biological replicates

of the main screen. A compound was considered an overall hit if two out of three biological replicates were statistically significant (padj <0.05) and the area under the curve (AUC) was at least 20% lower than for the DMSO control. **C**) Overlap of hits between main and independent validation screen. **D**) Normalised AUC values in main screen and validation experiment. The Pearson correlation coefficient, two-sided, and associated p-value is shown in the graph. The shaded areas indicate the 95% confidence interval.

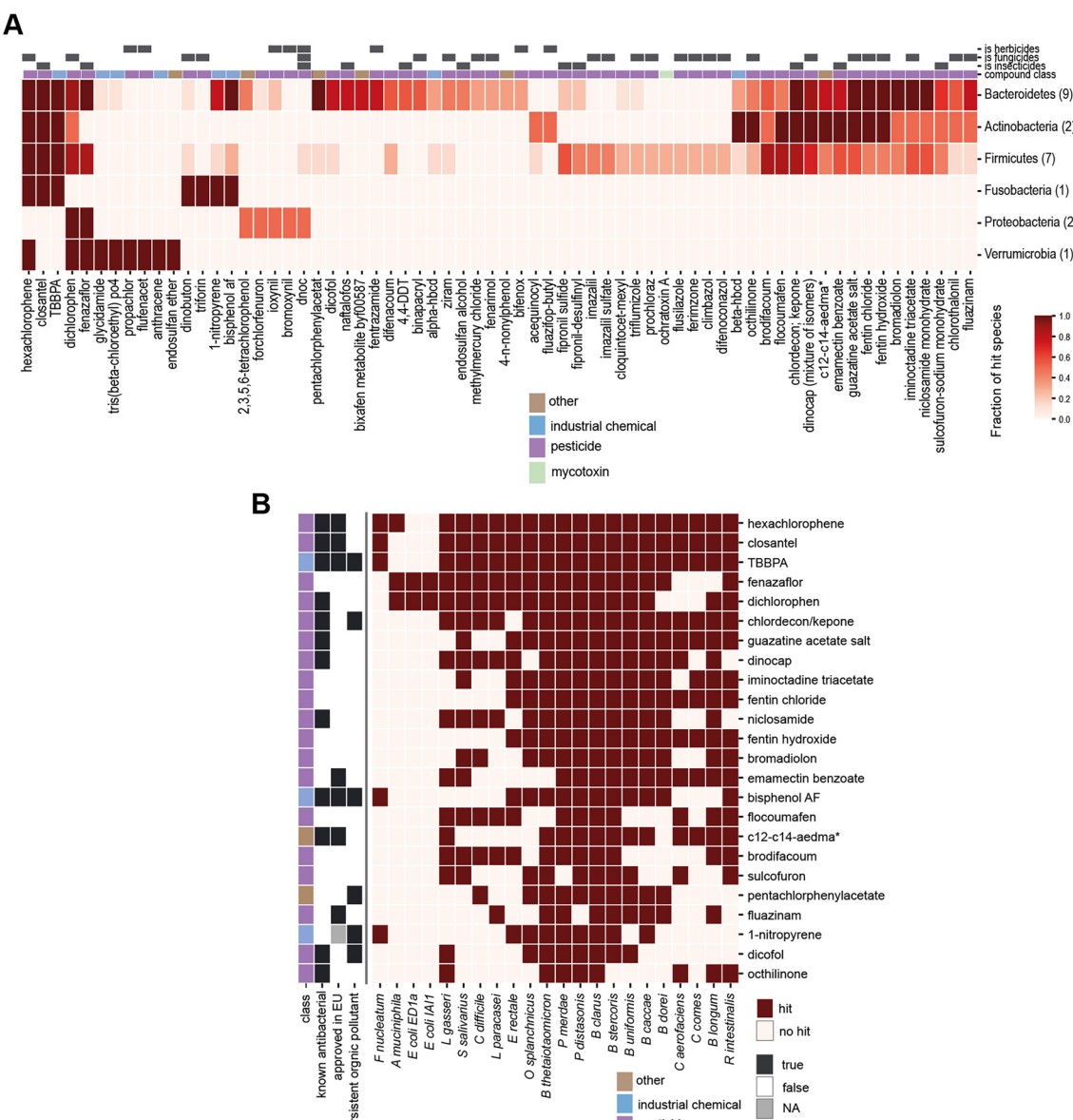

**Extended Data Fig. 2 | Phylogenetic susceptibility profiles and broad spectrum inhibition. A**) Chemical susceptibility profiles grouped by phylum. compounds (x-axis) and phyla (y-axis) were clustered by profile similarity, using Euclidean distance and Ward clustering. The number of strains in each phylum is indicated in parentheses. **B**) Heatmap showing growth inhibition by compounds (20 μM) which were identified to have broad-spectrum activity, inhibiting more than a third of the tested species. Results shown are a subset of those shown in Fig. 1c. Several of these compounds are widely used and/or persistent organic pollutants, raising concerns about their potential anti-gut-bacteria effects. Abbreviation: *C12-C14-Alkyl(ethylbenzyl)dimethylammonium chloride.

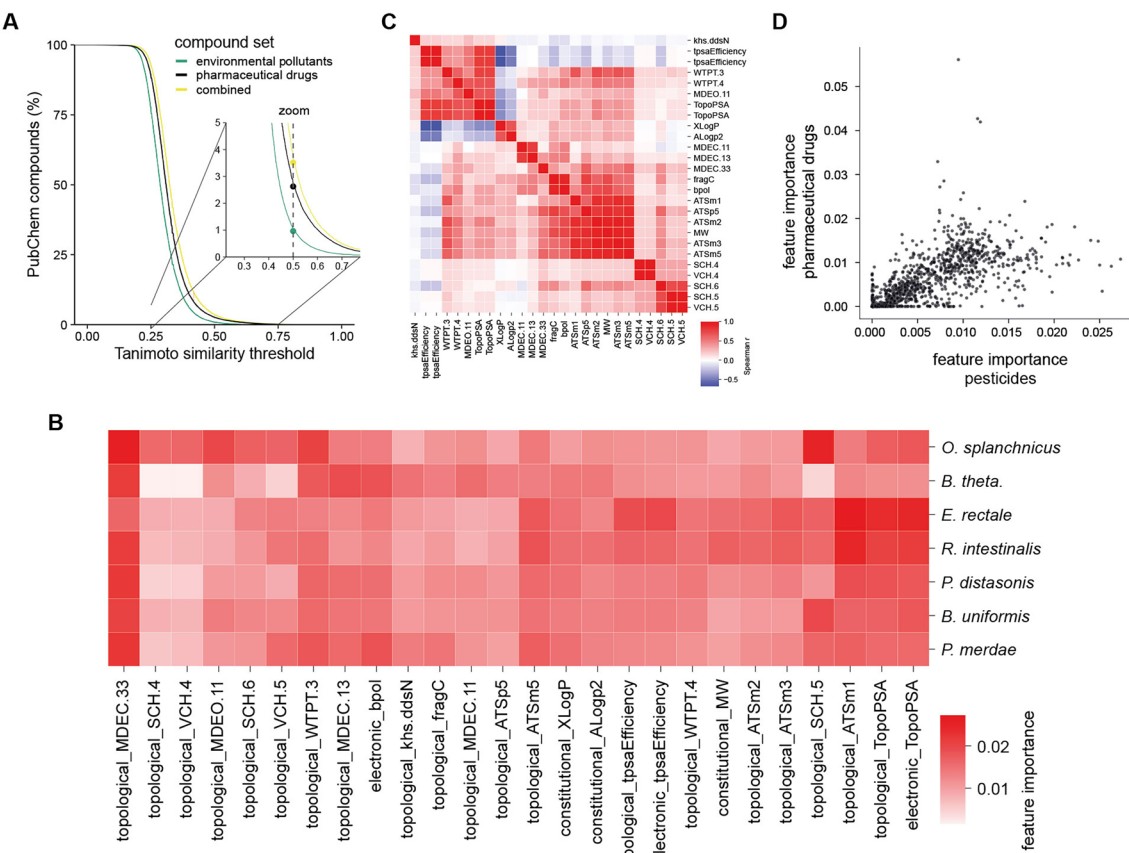

**Extended Data Fig. 3 | Chemoinformatics and machine learning details.**
**A**) Coverage of chemical space is increased by our new, complementary xenobiotic compound library. The plot illustrates the fraction of compounds in the random PubChem subset (y-axis, n=250,000) which have a similar compound (at the Tanimoto similarity threshold indicated on the x-axis), for the compound library of this study (green line), pharmaceutical drugs screened by Maier et al. 2018 (purple line) and the combined dataset. The inset focuses on a Tanimoto threshold of 0.75, where significantly increased coverage of the chemical space is observed (p< 2.2e-16, Kolmogorov-Smirnov test). **B**) Heatmap including all features which belonged to the top10 features in at least one species (ranked by

mean across 20 cross-validation sets). Feature importance was accessed using the feature_importances_attribute of the RandomForestClassifier, which relies on decrease in mean impurity to quantify feature importance. Feature based on molecular distance edge descriptors of carbon atoms (MDEC) (S. Liu et al., 1998) stood out as particularly useful for predicting anti-gut-bacterial activity across different species. **C**) Correlation analysis of the features with each other. **D**) Comparison of feature importances between pesticides and pharmaceutical drugs indicates that similar features are important for both compound classes. Spearman r=0.65, p=0.

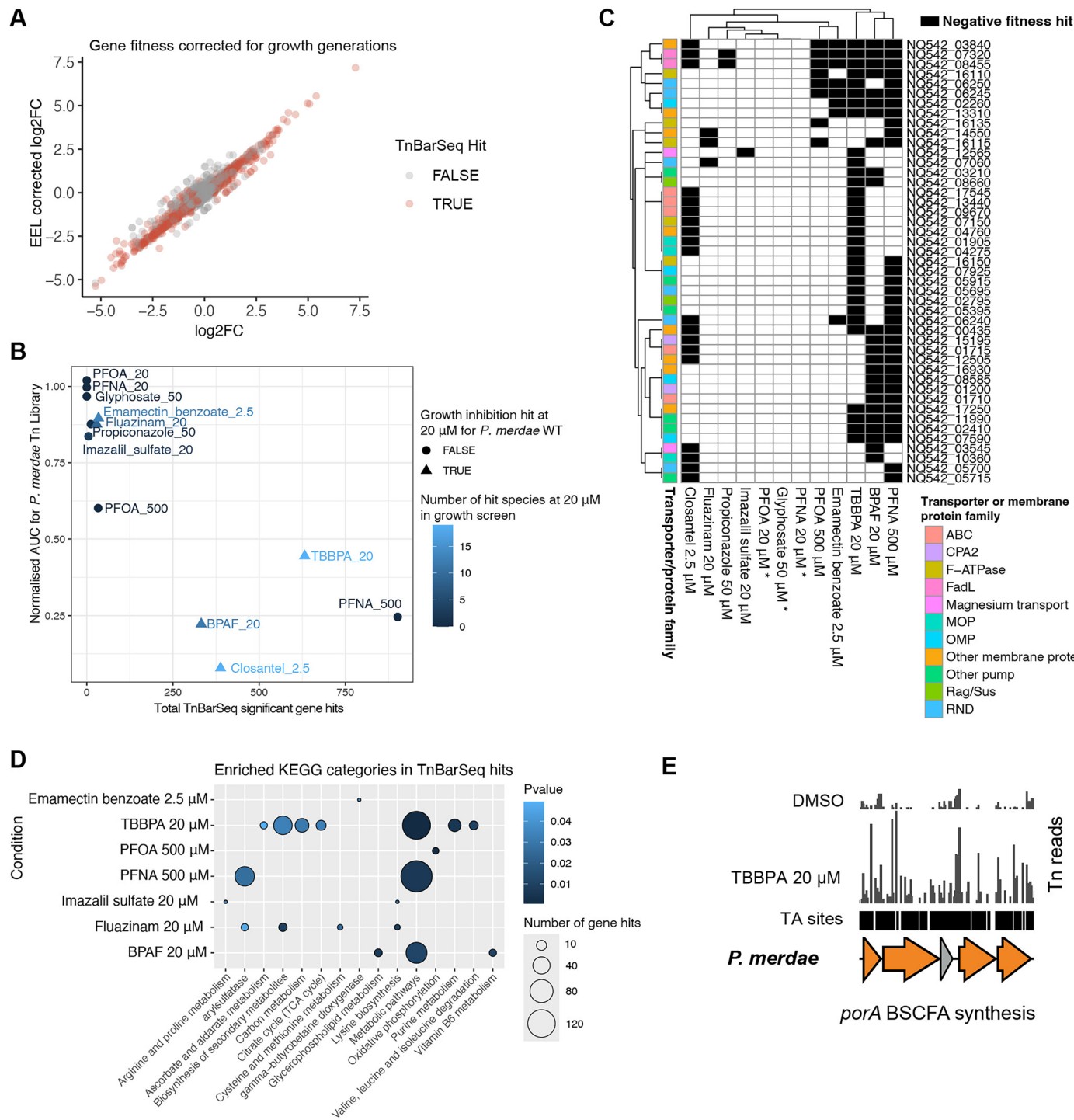

**Extended Data Fig. 4 | Chemical-genetic screen with *P. merdae* pooled library. A**) Post hoc correction for possible growth generation bias in log2FC (EEL method, Kim et al. 2024) show main TnBarSeq hits cannot be explained by differences library growth. Values per gene and condition are found in Supplementary Data 6. **B**) Area under the curve (AUC) for *P. merdae* Tn mutant library under the compounds tested normalised to control, plotted against total TnBarseq hits. The colour scale represents the number of bacterial species in the general growth screen which were an inhibition hit at 20 μM. Point shape indicates whether the compound was a growth hit for wild type *P. merdae* in the previous growth screen. **C**) Putative membrane transporters

and other membrane proteins showing negative fitness in response to more than one xenobiotic. Protein families are indicated in colours. ABC= Adenosine Triphosphate-Binding Cassette. CPA2= Monovalent cation:proton antiporter 2. RND= Resistance Nodulation Division. MOP= Multidrug/oligosachharidyl-lipid/polysaccharide. omp= ompA-H family. **D**) Enriched KEGG Pathway categories among all TnBarSeq hits for compounds tested. Number of gene hits are indicated by circle size, P-value is derived from Fisher's test. **E**) Representative TnBarSeq insertion plot for the porA branched short-chained fatty acid gene cluster (orange) in TBBPA 20 μM and DMSO control.

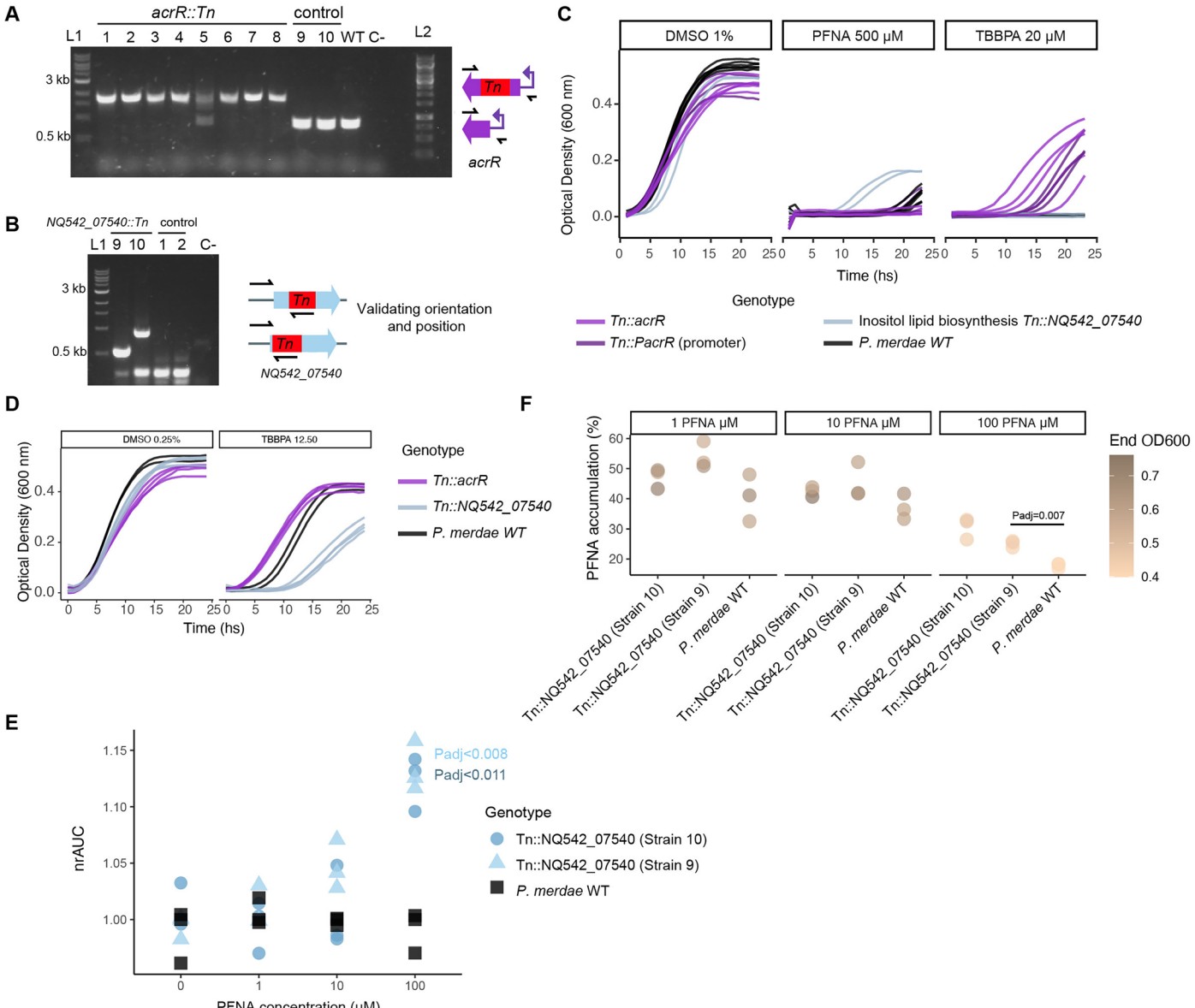

**Extended Data Fig. 5 | Isolate transposon mutant characterization.**
**A**) Diagnostic PCR genotyping of acrR locus transposon mutants
(Tn::NQ542_01170). Transposon insertions are validated as a larger amplicon size
compared to the wild type strain (WT). Mutant insertion location information
in Supplementary Data 7. **B**) Genotyping of transposon mutants located at a
putative phosphatidylinositol phosphate phosphatase (Tn::NQ542_07540)
using primers targeting the junction between transposon and insertion locus
confirms transposon location and orientation predicted by barcode sequencing.
**C**) Growth curves of isolate transposon mutants under xenobiotics. Curves
represent strains with independent insertions on acrR coding region and
promoter regions (purple n=7), Tn::NQ542_07540 from inositol lipid gene cluster
(light blue, n=2) and WT strains for comparison (black, n=6). **D**) Growth curves
at 12.5 μM TBBPA and DMSO control show Tn::NQ542_07540 hypersensitivity
to TBBPA. Curves represent two well replicates of two independent insertion

strains, and *P. merdae* WT control. **E**) Fitness of isolated *P. merdae* mutants
(strain 9 and 10 corresponding to independent insertions at NQ542_07540)
and WT control across a range of concentrations of PFNA (0–100 μM). NrAUC
is normalised to the median AUC of the WT value per concentration, and
the median of DMSO control (0 μM) per genotype. Replicates are shown as
points and P-adjusted values were obtained comparing mutants to WT at each
concentration with Welch's T-test corrected by multiplicity with BH. **F**) PFNA
bioaccumulation in isolated *P. merdae* strains of different genotypes grown on a
range of PFNA concentrations, as analysed by liquid chromatography coupled to
mass spectrometry (LC-MS), see Methods. End OD before compound extraction
is indicated in coloured scale. Replicate values are shown as points. Adjusted
P-values were calculated compared to WT using two-sided Welch's T-test
corrected by multiplicity with BH.

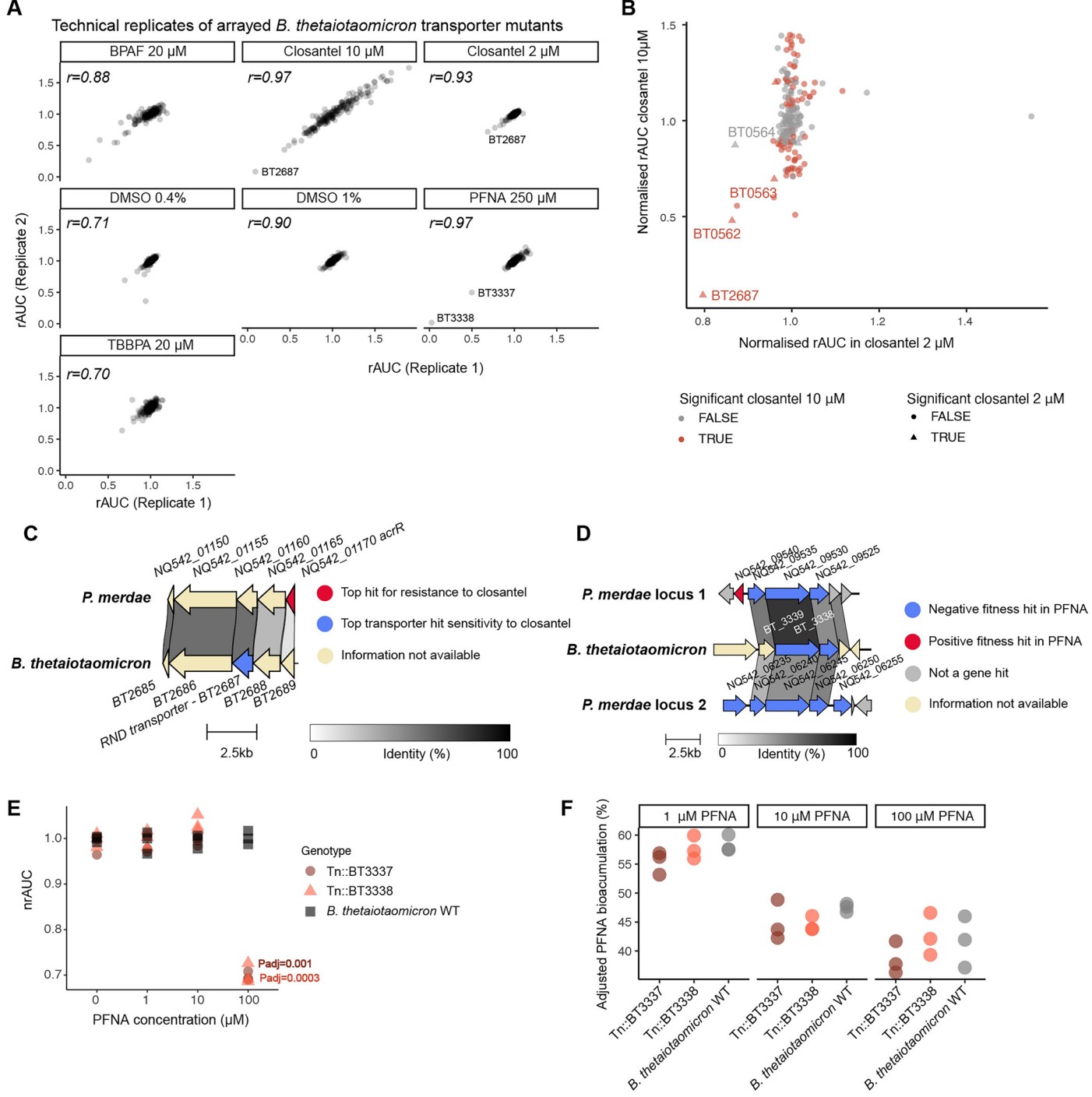

**Extended Data Fig. 6 | B. thetaiotaomicron transporter mutant arrayed screen QC and hit characterization. A**) Scatterplot of relative area under the curve for B. thetaiotaomicron arrayed transporter mutants (rAUC= AUC/ (median AUC of the plate)) across two technical replicate plates per condition. **B**) Scatterplot of *B. thetaiotaomicron* arrayed transporter library in 10 µM and 2 µM closantel. Significant genes P-adj<0.05 by two-sided Welch's T-test corrected by multiplicity with BH. Gene name of main hits in agreement are indicated. Axis show normalised rAUC (per mutant average rAUC xenobiotic / average rAUC DMSO). **C**) Conservation of the acrABR-tolC locus between *B. thetaiotaomicron* and *P. merdae*. Genes in red represent top candidates for positive selection under closantel in *P. merdae* (repressor of efflux acrR), while blue shows the top hit for sensitivity in the arrayed transporter panel of *B. thetaiotaomicron* (RND transporter BT687). Protein homology is shown as gray lines connecting orthologs using Clinker. **D**) Homology and synteny between top RND-efflux

genes negative fitness hits in *P. merdae* and *B. thetaiotaomicron*. **E**) Growth of *B. thetaiotaomicron* PFNA hypersensitive mutants (Tn::BT3337 and Tn::BT3338) and wild type (WT) across a range of concentrations of PFNA (0–100 µM). nrAUC = area under the growth curve (AUC) normalised to the median AUC of the WT value per concentration, and the median of DMSO control (0 µM) per genotype. Replicate values are shown as shapes per strains and P-adjusted values were obtained comparing mutants to WT at each concentration with two-sided Welch's T-test corrected by multiplicity with BH. **F**) PFNA bioaccumulation profiles of strains Tn::BT3337 and Tn::BT3338 and WT control on a range of PFNA concentrations. Adjusted bioaccumulation was calculated by normalising to the biomass of the WT per concentration. Non-significant differences were observed when comparing mutants to WT using two-sided Welch's T-test corrected by multiplicity with BH.

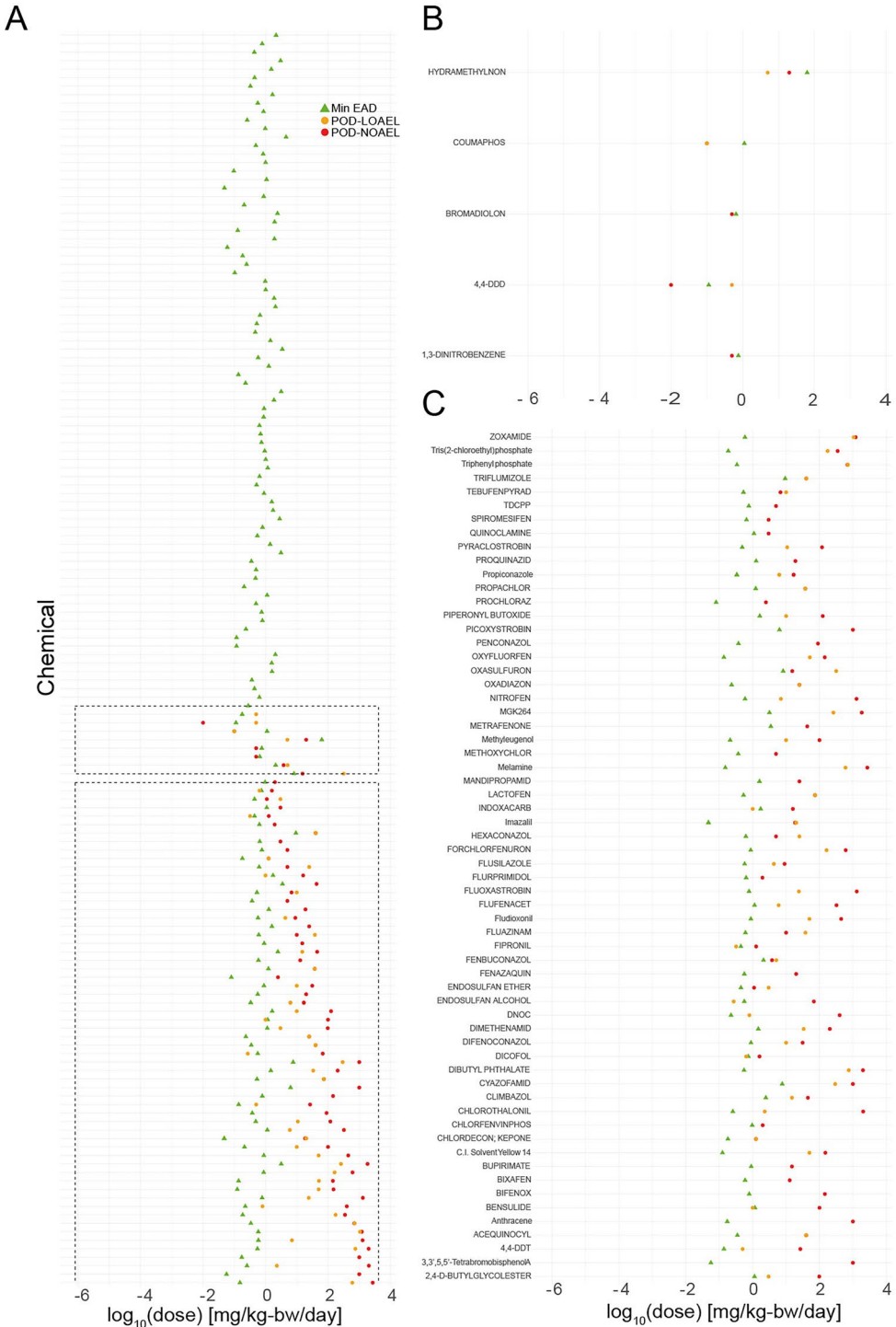

**Extended Data Fig. 7 | Toxicokinetic modelling suggests gut bacteria as sensitive indicators of chemical toxicity. A)** Analysis of minimum equivalent administered oral doses in humans EAD (green triangle), Point of departure - lowest adverse effect level (POD-LOAEL, orange circle) and Point of departure - no adverse effect level (POD-NOAEL, orange circle), for 149 chemicals. Boxes highlight chemicals with negative and positive POD ratios, enlarged in panel **B** and **C** respectively. To contrast the observed sensitivity of gut bacteria to chemicals against animal toxicity estimates, we applied physiologically based toxicokinetic (PBTK) modelling to 168 chemicals. Using a 30-day exposure scenario with 24 hour dose intervals, we used *in vitro* to *in vivo* extrapolation (IVIVE) to estimate equivalent administered oral doses in humans (EADs) that result in concentrations of xenobiotics in the gut at which we observed bacterial growth inhibition *in vitro* (Supplementary Data 12). We compared these EADs

to the doses at which toxicity was observed in a curated database of animal experiments (ToxValDB). This comparison is captured as the POD$_{ratio}$, which is the ratio of lowest dose at which an adverse effect was observed in mice (Point of departure - lowest adverse effect level: POD-LOAEL) to the previously determined EAD. We detected a positive $\log_{10}$(POD$_{ratio}$) for the vast majority of compounds in our analysis (56 out of 68 compounds for which animal data was available). A positive $\log_{10}$(POD$_{ratio}$) indicates that our *in vitro* screen picked up effects at lower equivalent doses than required to elicit toxicity in animals. While this is not unexpected, as animal toxicology does not usually consider the microbiome, this indicates that *in vitro* screening can be a sensitive method to assess potential adverse effects on microbiota not typically picked up in animal toxicity studies. More consideration and innovative study designs are required to better capture the microbiota-damaging toxic effects of xenobiotics in animal studies.

# Reporting Summary

## Statistics

For all statistical analyses, confirm that the following items are present in the figure legend, table legend, main text, or Methods section.

| n/a | Confirmed | |
|---|---|---|
| ☐ | ☒ | The exact sample size (n) for each experimental group/condition, given as a discrete number and unit of measurement |
| ☐ | ☒ | A statement on whether measurements were taken from distinct samples or whether the same sample was measured repeatedly |
| ☐ | ☒ | The statistical test(s) used AND whether they are one- or two-sided <br> *Only common tests should be described solely by name; describe more complex techniques in the Methods section.* |
| ☒ | ☐ | A description of all covariates tested |
| ☐ | ☒ | A description of any assumptions or corrections, such as tests of normality and adjustment for multiple comparisons |
| ☐ | ☒ | A full description of the statistical parameters including central tendency (e.g. means) or other basic estimates (e.g. regression coefficient) AND variation (e.g. standard deviation) or associated estimates of uncertainty (e.g. confidence intervals) |
| ☐ | ☒ | For null hypothesis testing, the test statistic (e.g. $F$, $t$, $r$) with confidence intervals, effect sizes, degrees of freedom and $P$ value noted <br> *Give P values as exact values whenever suitable.* |
| ☒ | ☐ | For Bayesian analysis, information on the choice of priors and Markov chain Monte Carlo settings |
| ☒ | ☐ | For hierarchical and complex designs, identification of the appropriate level for tests and full reporting of outcomes |
| ☐ | ☒ | Estimates of effect sizes (e.g. Cohen's $d$, Pearson's $r$), indicating how they were calculated |

*Our web collection on statistics for biologists contains articles on many of the points above.*

## Software and code

Policy information about availability of computer code

| Data collection | Plate reader control software: BioTek Gen5 Software v3.11.19 |
|---|---|
| Data analysis | R (v4.2.2, v4.2.0, v4.2.3) and RStudio Version v1.3.1093. <br> Python (v.3.x) <br> 2FAST2Q (https://veeninglab.com/2fast2q) <br> TRANSIT (v3.3.4) <br> TransAAP (https://www.membranetransport.org/transportDB2/TransAAP_login.html) <br> Clinker (https://cagecat.bioinformatics.nl/) <br> EggNOG mapper (http://eggnog-mapper.embl.de/) <br> PaperBLAST (https://papers.genomics.lbl.gov/cgi-bin/litSearch.cgi) <br> IVIVE tool at Integrated Chemical Environment (https://ice.ntp.niehs.nih.gov/) <br> CompTox Chemicals Dashboard (https://comptox.epa.gov/dashboard/) <br> Genome taxonomy database (GTDB) release R06-RS20253 <br> PubChem (https://pubchem.ncbi.nlm.nih.gov/) <br> EEL_Correction script (https://github.com/horiatodor/EEL_correction) <br> GrowthCurver in R version 0.3.1 <br> DECIPHER v. 2.24.054 <br> DADA2 v. 1.21.053 <br> MassHunter Workstation Quantitative Analysis for QQQ v10.1 |

For manuscripts utilizing custom algorithms or software that are central to the research but not yet described in published literature, software must be made available to editors and reviewers. We strongly encourage code deposition in a community repository (e.g. GitHub). See the Nature Portfolio guidelines for submitting code & software for further information.

# Data

Policy information about availability of data

All manuscripts must include a data availability statement. This statement should provide the following information, where applicable:
- Accession codes, unique identifiers, or web links for publicly available datasets
- A description of any restrictions on data availability
- For clinical datasets or third party data, please ensure that the statement adheres to our policy

> Data that support the findings of this study are included in Supplementary Information. Growth curves, LC-MS2 raw data, TnBarSeq mapped files as well as associated processing scripts and intermediate files are available via Mendeley Data DOI: 10.17632/g7hy84t2r6.1 Raw sequencing reads of 16S sequencing of synthetic communities and TnBarSeq data were uploaded to the European Nucleotide Archive (PRJEB97051and PRJEB76605).

# Research involving human participants, their data, or biological material

Policy information about studies with human participants or human data. See also policy information about sex, gender (identity/presentation), and sexual orientation and race, ethnicity and racism.

| | |
|---|---|
| Reporting on sex and gender | *Use the terms sex (biological attribute) and gender (shaped by social and cultural circumstances) carefully in order to avoid confusing both terms. Indicate if findings apply to only one sex or gender; describe whether sex and gender were considered in study design; whether sex and/or gender was determined based on self-reporting or assigned and methods used.*<br>*Provide in the source data disaggregated sex and gender data, where this information has been collected, and if consent has been obtained for sharing of individual-level data; provide overall numbers in this Reporting Summary. Please state if this information has not been collected.*<br>*Report sex- and gender-based analyses where performed, justify reasons for lack of sex- and gender-based analysis.* |
| Reporting on race, ethnicity, or other socially relevant groupings | *Please specify the socially constructed or socially relevant categorization variable(s) used in your manuscript and explain why they were used. Please note that such variables should not be used as proxies for other socially constructed/relevant variables (for example, race or ethnicity should not be used as a proxy for socioeconomic status).*<br>*Provide clear definitions of the relevant terms used, how they were provided (by the participants/respondents, the researchers, or third parties), and the method(s) used to classify people into the different categories (e.g. self-report, census or administrative data, social media data, etc.)*<br>*Please provide details about how you controlled for confounding variables in your analyses.* |
| Population characteristics | *Describe the covariate-relevant population characteristics of the human research participants (e.g. age, genotypic information, past and current diagnosis and treatment categories). If you filled out the behavioural & social sciences study design questions and have nothing to add here, write "See above."* |
| Recruitment | *Describe how participants were recruited. Outline any potential self-selection bias or other biases that may be present and how these are likely to impact results.* |
| Ethics oversight | *Identify the organization(s) that approved the study protocol.* |

Note that full information on the approval of the study protocol must also be provided in the manuscript.

# Field-specific reporting

Please select the one below that is the best fit for your research. If you are not sure, read the appropriate sections before making your selection.

☒ Life sciences    ☐ Behavioural & social sciences    ☐ Ecological, evolutionary & environmental sciences

For a reference copy of the document with all sections, see nature.com/documents/nr-reporting-summary-flat.pdf

# Life sciences study design

All studies must disclose on these points even when the disclosure is negative.

| | |
|---|---|
| Sample size | The growth screen sample size (n=3) was chosen based on experimental feasibility and convention. The chemical-genetic screens were done with at least two replicates, as has been used before in similar genetic screens in Bacteroidales (Arjes et al 2022; Liu et al, 2021) |
| Data exclusions | No data were excluded from the growth screen. |
| Replication | In the main growth screen, to ensure reproducibility we acquired three independent biological replicates (independent day replicates) across separate experiment days. We further verified the reproducibility by conducting an independent validation screen which confirmed the majority of the original hits. For chemical-genetic screens there were follow up experiments that replicated expected mutant enrichment, and the expected phenotype of isolate mutant strains. |
| Randomization | Randomization is not relevant to this study as no group allocation took place. |

| Randomization | |
|---|---|
| Blinding | No blinding took place in this study. Blinding is not relevant to this study as no group allocation took place. |

# Reporting for specific materials, systems and methods

We require information from authors about some types of materials, experimental systems and methods used in many studies. Here, indicate whether each material, system or method listed is relevant to your study. If you are not sure if a list item applies to your research, read the appropriate section before selecting a response.

## Materials & experimental systems

| n/a | Involved in the study |
|---|---|
| ☒ | ☐ Antibodies |
| ☒ | ☐ Eukaryotic cell lines |
| ☒ | ☐ Palaeontology and archaeology |
| ☒ | ☐ Animals and other organisms |
| ☒ | ☐ Clinical data |
| ☒ | ☐ Dual use research of concern |
| ☒ | ☐ Plants |

## Methods

| n/a | Involved in the study |
|---|---|
| ☒ | ☐ ChIP-seq |
| ☒ | ☐ Flow cytometry |
| ☒ | ☐ MRI-based neuroimaging |

## Plants

| Seed stocks | *Report on the source of all seed stocks or other plant material used. If applicable, state the seed stock centre and catalogue number. If plant specimens were collected from the field, describe the collection location, date and sampling procedures.* |
|---|---|
| Novel plant genotypes | *Describe the methods by which all novel plant genotypes were produced. This includes those generated by transgenic approaches, gene editing, chemical/radiation-based mutagenesis and hybridization. For transgenic lines, describe the transformation method, the number of independent lines analyzed and the generation upon which experiments were performed. For gene-edited lines, describe the editor used, the endogenous sequence targeted for editing, the targeting guide RNA sequence (if applicable) and how the editor was applied.* |
| Authentication | *Describe any authentication procedures for each seed stock used or novel genotype generated. Describe any experiments used to assess the effect of a mutation and, where applicable, how potential secondary effects (e.g. second site T-DNA insertions, mosiacism, off-target gene editing) were examined.* |

