## [Peer Review File · Nature Microbiology]

Industrial and agricultural chemicals exhibit antimicrobial activity against human gut bacteria in vitro

Corresponding Author: Professor Kiran Patil

Version 0:

Reviewer comments:

Reviewer #1

(Remarks to the Author)

The authors performed a large-scale screening for effects of chemical pollutants on a small collection of highly prevalent gut microbiome strains, identifying hundreds of potential growth inhibition effects of pollutants on gut strains. Next, they evaluated the effective dose on which the pollutant effects inhibit strain growth. They also performed the analysis on bacterial communities rather than individual bacteria, and even used transposon knock out model to identify genes that define strain susceptibility to pollutants.

Basically, any of the follow-up experiments I can think of are already done.

This is an incredible study both in terms of scientific and public importance. I could think only of some textual corrections, like:

p111: Contamination ... are -> contamination ... is

p.14|28: The gut microbiota are - could be use to use 'is' here

p15|32: anti-gut bacterial - should be rephrased to just 'antibacterial',

And some others. Overall, the quality of text is quite good, maybe just a bit of proofreading could help.

Reviewer #2

(Remarks to the Author)

This study looked at the impact of > 1000 pollutant chemicals on the growth of 22 gut bacterial strains in vitro. A total of 168 chemicals, most of which were not previously known to have antibacterial effects, reduced growth of at least one bacterial strain. A subset of these compounds was also tested in a synthetic microbial community and shown to alter community composition. The authors proceeded to analyse the genetic basis of such off-target effect of these pollutants on the microbiota. Membrane transport and polarity were identified as key modulators of off-target toxicity. Chemical resistance mutants included those deficient in the biosynthesis of beneficial metabolites, such as branched short-chain fatty acids, which are relevant to human health. However, the effective capacity of chemicals such as TBPPA to reduce the production of such metabolites by microbes such as *P. merdae* awaits further validation.

This is an important study that expands our understanding of the ability of xenobiotics to impact the human microbiota. The study was carefully designed, controls seemed largely appropriate, and effort was made to replicate results for compounds with hits. I have a few points related, e.g., to the relevance of the concentration tested or presentation of results, that I would like to see clarified/discussed before publication.

1. It is mentioned in the introduction that some pollutants have been measured in human blood or urine. How do available concentrations from urine and blood compare with the 20 μ M used in the screen? or with the concentrations determined by the toxicokinetic modelling applied in this study?

2. Page 5, lines 10-13 and Supplementary Data 3: I could only find reported whether a compound significantly impacted growth (> 20% reduction in AUC, True or False), and not the magnitude of such reduction. Why? A 90% inhibition in growth may have quite different repercussions, e.g. to the microbiota community and even the host, compared to a 20.1% growth inhibition. These results would be of interest to the research community.

In addition, in such a large collection of compounds, is it true that none had a positive impact on bacterial growth compared to DMSO? Can the authors report on that or explain in the manuscript why was growth promotion not investigated (e.g., due to the experimental set-up or rich medium used).

3. Figure 4A, B: the pollutants tested are chemically distinct from previously tested pharmaceutical drugs. But are pollutants with microbiota hits and pharmaceuticals with microbiota hits chemically distinct? Or is there an overlap/similarity in terms of their chemical structures?

Of the 168 compounds shown to affect at least one microbiome member, only a subset (24) shows a broad toxicity range. What is the chemical diversity of compounds with such a broad spectrum? Can the authors highlight these compounds in Figure 4A?

4. Section “Chemicals alter the composition of synthetic gut bacterial communities”: what was the rationale for selecting a synthetic community containing bacterial strains distinct from the strains tested in pure culture?

Other points:

-Materials and methods, under “statistical analysis of growth”: what do the authors mean by “control” wells? Why was AUC normalisation for industrial chemicals and mycotoxins done differently – there were no edge effects for the other plates?

-Figure 1 C: In addition to the total number of compounds tested, could be useful to add the total number of compounds tested from each type to each of the main coloured rectangles.

-Page 9, line 31: please define “KS test”.

Reviewer #3

(Remarks to the Author)

In this manuscript, Lindell et al present a large screen of more than 1000 pollutants and non-pharmaceutical chemicals against 22 members of the human gut microbiome, with a direct readout of growth inhibition. They uncover hundreds of potential interactions, follow up on a few of them to calculate MICs, and perform a genetic screen to identify resistance mechanisms to these chemicals.

The experiments are executed in a similar manner to what was performed previously by the same authors with pharmaceuticals, and the direct screen results are compared to those from the earlier study. I do have several concerns about the initial rationale of the study design, relevance to the real world, and the overinterpretation of the results.

1. I understand that a 20 μM concentration of the chemicals in the screen was selected as a means of comparison to the previous pharmaceuticals screen. While this concentration may be relevant to pharmaceuticals that are given at high dosages and over extended periods of time, it is unclear if this concentration is at all relevant to pollutants that are typically found in trace amounts in food and water. This is also exaggerated by the fact that the authors present their data in a binary mode (significant/not significant) based on a very limited inhibition cutoff (20%), without clear reflection of the magnitude of inhibition in their hits. Based on the concentrations shown in the few cases of MIC determination, it looks like most of the cases will have limited growth inhibition at micromolar concentrations. The authors should either rationalize the use of such high concentration based on real world exposure data, or repeat the screen at a lower concentration. This statement is really misleading given the presented data from the MIC experiment: “Many compounds show strong inhibitory effects at substantially lower concentrations than were tested in the main screen. E.g., imazalil sulfate and prochloraz inhibited *Eubacterium rectale*, a prevalent species that produces health-associated metabolite butyrate, at a concentration as low as 2.5 μM ”. MICs are not even presented in the main text, even though the goal of the study was to determine MICs.

2. Figure 2a shows the proportions of strains from each phylum that are sensitive to the different chemicals. This analysis is completely biased by the small sample size of the bacterial strains tested here, and the limited sampling in most Phyla. For example, a 9/9 of the Bacteroidetes shows exactly the same result as a 1/1 of the Fusobacteria, which is not a meaningful number to base a whole Phylum results on. Proteobacteria are represented by two strains, both of which are *E. coli* – proportion there means nothing. Verrucomicrobia is also 1, and Actinobacteria is 2....This entire analysis is flawed in my opinion.

3. An in depth structural analysis of the chemicals that show broad versus specific effects is more useful than the information about their use in EU or US, and may reveal actual features of these molecules that can predict their toxicity (this was partially done with one example in the manuscript, benzoate).

4. Only 75% replicated between the original screen and the MIC experiment, can the author explain the reason behind the failure of the remaining 25%?

5. The concentration of the two xenobiotics used to measure effects on the synthetic community was not described – but assuming it is 20 μM , then again it is the same issue as in the point above, this concentration needs to be physiologically relevant, and it is somewhat of an overinterpretation to assume that effects in monoculture are translatable to community effects based on the results of only two xenobiotics tested. The ultimate measure here is of course the exposure of a human-relevant dose in a mammalian setting (mice) and the quantification of microbiome changes in this setting. Such test would take into account real exposure in a mammalian gut, and the concentration/time needed to observe any significant changes.

6. Were the strains used in the Com20 the same exact strains as the ones in the monoculture screen?

7. “In the case of therapeutic drugs, the emergent community-scale effects are due to bacterial metabolism of drugs and the impact of drugs on bacterial metabolism. The results from the BPAF and TBBPA suggest that similar mechanisms are applicable in the case of chemical contaminants.” This is an unsupported statement without any mechanistic insights into metabolism of chemical contaminants by members of the microbiome, or effect of these chemicals on microbial metabolism.

8. The authors present the modeling results as a support for the relevance of their findings, yet even in the modeling experiment they appear to simulate an unrealistic dosage scheme for a pollutant. I am not even sure how they calculated the concentrations

to be used in their repeated exposures. The description appears to indicate that they selected a regiment that would yield an inhibitory dose, which is circular in nature. Isn't the modeling supposed to test whether a realistic dose can lead to gut exposure levels that result in inhibition?

9. "We noted a positive correlation of number of compounds affecting a species with its abundance in the human gut" – what is the rationale to perform this correlation? It is heavily confounded by the sampling of the molecules tested and the sampling of the microbes tested. It is also irrelevant: abundant microbes do not have anything specific to them to make them more or less susceptible to the random set of compounds tested in this study. Same with the correlation between microbes affected by pharmaceuticals and non-pharmaceuticals. This statement is misleading: "The correlation in the case of human-targeted drugs indicates that bacteria use a common and non-specific mechanism to tackle non-antibiotic drugs and chemical pollutants." – bacteria do not evolve to "tackle" specific molecules they have never been exposed to! And certainly do not use common and non-specific mechanisms to interact with hundreds of molecules. I encourage the authors to refrain from making such broad statements that are not based on real biological rationales nor on newly collected data.

10. There are several issues with the genetic screen to identify resistance/sensitivity determinants. First, some unrealistic concentrations were used, such as 50 μM and 500 μM (even 20 μM is high). Second, the effects on efflux, membrane integrity, transport are all expected with chemical stress, and appear to be non-specific in most cases. The only unexpected result is the fact that some metabolic pathway mutants appear to mediate a growth advantage under treatment conditions. It is not clear if this result is simply a reflection of these mutants being faster in growth than WT, and therefore appear to be enriched when overall growth inhibition is at play, or if there is anything specific to this interaction. The typical follow-up experiment to confirm these findings is to isolate (or remake) these specific mutants and compare their MIC or growth curves to the WT in the presence of different concentrations of the xenobiotic. Without such confirmation, the results are hard to interpret.

11. A better experiment than the transposon one is to evolve resistant mutants, by treating the WT with sub MIC concentrations in subsequent rounds and isolating strains that grow undeterred in the presence of such concentrations. This is more of a reflection to a real life setting in the human microbiome. Finally, the authors claim that exposure to pollutants can lead to mutants that lost the functionality of host-relevant metabolic pathways. It is relatively simple to check in bacterial isolates or metagenomic data from the human microbiome if these mutants do indeed exist, where human populations are actually exposed to the studied chemicals for a long time and under the biologically relevant concentrations. Absent this data or similar one presented from new experiments, the current results are indeed confounded by the unrealistically high concentrations and exposures used.

12. This may appear trivial but it is an extremely serious issue, especially in a high-profile microbiology journal that should abide to the highest standards: I encourage the authors to study the real definition of "commensals" and "commensalism" before using this terminology. Commensalism, mutualism, parasitism, symbiosis, etc., are all well-bound terms with accurate ecological definitions, and should not be casually used. I don't believe that we have experimental data to support calling any of the 22 organisms tested in this study "commensals" – obviously not *C. difficile*, but actually neither the remaining 21 strains.

Decision Letter:

13th December 2024

Dear Kiran,

Thank you for your patience while your manuscript "Off-purpose activity of industrial and agricultural chemicals against human gut bacteria" was under peer-review at Nature Microbiology. It has now been seen by 3 referees, whose expertise and comments you will find at the end of this email. Although they find your work of some potential interest, they have raised a number of concerns that will need to be addressed before we can consider publication of the work in Nature Microbiology.

Thank you for preparing a revision plan in response to these concerns. After discussion amongst the editorial team, we felt that these responses were reasonable. Please complete the proposed revisions including the in vitro experimental work and discuss the limitations clearly in the paper. Please also clearly explain in the main text that existing data on this topic is limited.

Should further experimental data allow you to address these criticisms, we would be happy to look at a revised manuscript.

Please include a data availability statement as a separate section after Methods but before references, under the heading "Data Availability". This section should inform readers about the availability of the data used to support the conclusions of your study. This information includes accession codes to public repositories (data banks for protein, DNA or RNA sequences, microarray,

proteomics data etc...), references to source data published alongside the paper, unique identifiers such as URLs to data repository entries, or data set DOIs, and any other statement about data availability. At a minimum, you should include the following statement: "The data that support the findings of this study are available from the corresponding author upon request", mentioning any restrictions on availability. If DOIs are provided, we also strongly encourage including these in the Reference list (authors, title, publisher (repository name), identifier, year). For more guidance on how to write this section please see: <http://www.nature.com/authors/policies/data/data-availability-statements-data-citations.pdf>

* If you have not done so already we suggest that you begin to revise your manuscript so that it conforms to our Article format instructions at <http://www.nature.com/nmicrobiol/info/final-submission>. Refer also to any guidelines provided in this letter.

When submitting the revised version of your manuscript, please pay close attention to our [href="https://www.nature.com/nature-portfolio/editorial-policies/image-integrity">Digital Image Integrity Guidelines](https://www.nature.com/nature-portfolio/editorial-policies/image-integrity) and to the following points below:

Link Redacted

Note: This url links to your confidential homepage and associated information about manuscripts you may have submitted or be reviewing for us. If you wish to forward this e-mail to co-authors, please delete this link to your homepage first.

Nature Microbiology is committed to improving transparency in authorship. As part of our efforts in this direction, we are now requesting that all authors identified as 'corresponding author' on published papers create and link their Open Researcher and Contributor Identifier (ORCID) with their account on the Manuscript Tracking System (MTS), prior to acceptance. This applies to primary research papers only. ORCID helps the scientific community achieve unambiguous attribution of all scholarly contributions. You can create and link your ORCID from the home page of the MTS by clicking on 'Modify my Springer Nature account'. For more information please visit [please visit www.springernature.com/orcid](http://www.springernature.com/orcid).

If you wish to submit a suitably revised manuscript we would hope to receive it within 6 months. If you cannot send it within this time, please let us know. We will be happy to consider your revision, even if a similar study has been accepted for publication at Nature Microbiology or published elsewhere (up to a maximum of 6 months).

Yours sincerely,

Reviewer Expertise:

Referee #1: microbiome, bioinformatics, omics

Referee #2: microbial xenobiotic metabolism

Referee #3: microbial xenobiotic metabolism

Reviewer Comments:

Reviewer #1 (Remarks to the Author):

The authors performed a large-scale screening for effects of chemical pollutants on a small collection of highly prevalent gut microbiome strains, identifying hundreds of potential growth inhibition effects of pollutants on gut strains. Next, they evaluated the effective dose on which the pollutant effects inhibit strain growth. They also performed the analysis on bacterial communities

rather than individual bacteria, and even used transposon knock out model to identify genes that define strain susceptibility to pollutants.

Basically, any of the follow-up experiments I can think of are already done.

This is an incredible study both in terms of scientific and public importance. I could think only of some textual corrections, like:

p111. Contamination ... are -> contamination ... is

p14|28: The gut microbiota are - could be use to use 'is' here

p15|32: anti-gut bacterial - should be rephrased to just 'antibacterial',

And some others. Overall, the quality of text is quite good, maybe just a bit of proofreading could help.

Reviewer #2 (Remarks to the Author):

This study looked at the impact of > 1000 pollutant chemicals on the growth of 22 gut bacterial strains in vitro. A total of 168 chemicals, most of which were not previously known to have antibacterial effects, reduced growth of at least one bacterial strain. A subset of these compounds was also tested in a synthetic microbial community and shown to alter community composition. The authors proceeded to analyse the genetic basis of such off-target effect of these pollutants on the microbiota. Membrane transport and polarity were identified as key modulators of off-target toxicity. Chemical resistance mutants included those deficient in the biosynthesis of beneficial metabolites, such as branched short-chain fatty acids, which are relevant to human health. However, the effective capacity of chemicals such as TBPPA to reduce the production of such metabolites by microbes such as *P. merdae* awaits further validation.

This is an important study that expands our understanding of the ability of xenobiotics to impact the human microbiota. The study was carefully designed, controls seemed largely appropriate, and effort was made to replicate results for compounds with hits. I have a few points related, e.g., to the relevance of the concentration tested or presentation of results, that I would like to see clarified/discussed before publication.

1. It is mentioned in the introduction that some pollutants have been measured in human blood or urine. How do available concentrations from urine and blood compare with the 20 μ M used in the screen? or with the concentrations determined by the toxicokinetic modelling applied in this study?

2. Page 5, lines 10-13 and Supplementary Data 3: I could only find reported whether a compound significantly impacted growth (> 20% reduction in AUC, True or False), and not the magnitude of such reduction. Why? A 90% inhibition in growth may have quite different repercussions, e.g. to the microbiota community and even the host, compared to a 20.1% growth inhibition. These results would be of interest to the research community.

In addition, in such a large collection of compounds, is it true that none had a positive impact on bacterial growth compared to DMSO? Can the authors report on that or explain in the manuscript why was growth promotion not investigated (e.g., due to the experimental set-up or rich medium used).

3. Figure 4A, B: the pollutants tested are chemically distinct from previously tested pharmaceutical drugs. But are pollutants with microbiota hits and pharmaceuticals with microbiota hits chemically distinct? Or is there an overlap/similarity in terms of their chemical structures?

Of the 168 compounds shown to affect at least one microbiome member, only a subset (24) shows a broad toxicity range. What is the chemical diversity of compounds with such a broad spectrum? Can the authors highlight these compounds in Figure 4A?

4. Section "Chemicals alter the composition of synthetic gut bacterial communities": what was the rationale for selecting a synthetic community containing bacterial strains distinct from the strains tested in pure culture?

Other points:

-Materials and methods, under "statistical analysis of growth": what do the authors mean by "control" wells? Why was AUC normalisation for industrial chemicals and mycotoxins done differently – there were no edge effects for the other plates?

-Figure 1 C: In addition to the total number of compounds tested, could be useful to add the total number of compounds tested from each type to each of the main coloured rectangles.

-Page 9, line 31: please define "KS test".

Reviewer #3 (Remarks to the Author):

In this manuscript, Lindell et al present a large screen of more than 1000 pollutants and non-pharmaceutical chemicals against 22 members of the human gut microbiome, with a direct readout of growth inhibition. They uncover hundreds of potential interactions, follow up on a few of them to calculate MICs, and perform a genetic screen to identify resistance mechanisms to these chemicals.

The experiments are executed in a similar manner to what was performed previously by the same authors with pharmaceuticals, and the direct screen results are compared to those from the earlier study. I do have several concerns about the initial rationale of the study design, relevance to the real world, and the overinterpretation of the results.

1. I understand that a 20 μM concentration of the chemicals in the screen was selected as a means of comparison to the previous pharmaceuticals screen. While this concentration may be relevant to pharmaceuticals that are given at high dosages and over extended periods of time, it is unclear if this concentration is at all relevant to pollutants that are typically found in trace amounts in food and water. This is also exaggerated by the fact that the authors present their data in a binary mode (significant/not significant) based on a very limited inhibition cutoff (20%), without clear reflection of the magnitude of inhibition in their hits. Based on the concentrations shown in the few cases of MIC determination, it looks like most of the cases will have limited growth inhibition at micromolar concentrations. The authors should either rationalize the use of such high concentration based on real world exposure data, or repeat the screen at a lower concentration. This statement is really misleading given the presented data from the MIC experiment: "Many compounds show strong inhibitory effects at substantially lower concentrations than were tested in the main screen. E.g., imazalil sulfate and prochloraz inhibited *Eubacterium rectale*, a prevalent species that produces health-associated metabolite butyrate, at a concentration as low as 2.5 μM ". MICs are not even presented in the main text, even though the goal of the study was to determine MICs.

2. Figure 2a shows the proportions of strains from each phylum that are sensitive to the different chemicals. This analysis is completely biased by the small sample size of the bacterial strains tested here, and the limited sampling in most Phyla. For example, a 9/9 of the Bacteroidetes shows exactly the same result as a 1/1 of the Fusobacteria, which is not a meaningful number to base a whole Phylum results on. Proteobacteria are represented by two strains, both of which are *E. coli* – proportion there means nothing. Verrucomicrobia is also 1, and Actinobacteria is 2.... This entire analysis is flawed in my opinion.

3. An in depth structural analysis of the chemicals that show broad versus specific effects is more useful than the information about their use in EU or US, and may reveal actual features of these molecules that can predict their toxicity (this was partially done with one example in the manuscript, benzoate).

4. Only 75% replicated between the original screen and the MIC experiment, can the author explain the reason behind the failure of the remaining 25%?

5. The concentration of the two xenobiotics used to measure effects on the synthetic community was not described – but assuming it is 20 μM , then again it is the same issue as in the point above, this concentration needs to be physiologically relevant, and it is somewhat of an overinterpretation to assume that effects in monoculture are translatable to community effects based on the results of only two xenobiotics tested. The ultimate measure here is of course the exposure of a human-relevant dose in a mammalian setting (mice) and the quantification of microbiome changes in this setting. Such test would take into account real exposure in a mammalian gut, and the concentration/time needed to observe any significant changes.

6. Were the strains used in the Com20 the same exact strains as the ones in the monoculture screen?

7. "In the case of therapeutic drugs, the emergent community-scale effects are due to bacterial metabolism of drugs and the impact of drugs on bacterial metabolism. The results from the BPAF and TBBPA suggest that similar mechanisms are applicable in the case of chemical contaminants." This is an unsupported statement without any mechanistic insights into metabolism of chemical contaminants by members of the microbiome, or effect of these chemicals on microbial metabolism.

8. The authors present the modeling results as a support for the relevance of their findings, yet even in the modeling experiment they appear to simulate an unrealistic dosage scheme for a pollutant. I am not even sure how they calculated the concentrations to be used in their repeated exposures. The description appears to indicate that they selected a regimen that would yield an inhibitory dose, which is circular in nature. Isn't the modeling supposed to test whether a realistic dose can lead to gut exposure levels that result in inhibition?

9. "We noted a positive correlation of number of compounds affecting a species with its abundance in the human gut" – what is the rationale to perform this correlation? It is heavily confounded by the sampling of the molecules tested and the sampling of the microbes tested. It is also irrelevant: abundant microbes do not have anything specific to them to make them more or less susceptible to the random set of compounds tested in this study. Same with the correlation between microbes affected by pharmaceuticals and non-pharmaceuticals. This statement is misleading: "The correlation in the case of human-targeted drugs indicates that bacteria use a common and non-specific mechanism to tackle non-antibiotic drugs and chemical pollutants." – bacteria do not evolve to "tackle" specific molecules they have never been exposed to! And certainly do not use common and non-specific mechanisms to interact with hundreds of molecules. I encourage the authors to refrain from making such broad statements that are not based on real biological rationales nor on newly collected data.

10. There are several issues with the genetic screen to identify resistance/sensitivity determinants. First, some unrealistic concentrations were used, such as 50 μM and 500 μM (even 20 μM is high). Second, the effects on efflux, membrane integrity, transport are all expected with chemical stress, and appear to be non-specific in most cases. The only unexpected result is the fact that some metabolic pathway mutants appear to mediate a growth advantage under treatment conditions. It is not clear if this result is simply a reflection of these mutants being faster in growth than WT, and therefore appear to be enriched when overall growth inhibition is at play, or if there is anything specific to this interaction. The typical follow-up experiment to confirm these findings is to isolate (or remake) these specific mutants and compare their MIC or growth curves to the WT in the presence of different concentrations of the xenobiotic. Without such confirmation, the results are hard to interpret.

11. A better experiment than the transposon one is to evolve resistant mutants, by treating the WT with sub MIC concentrations in subsequent rounds and isolating strains that grow undeterred in the presence of such concentrations. This is more of a reflection to a real life setting in the human microbiome. Finally, the authors claim that exposure to pollutants can lead to mutants that lost the functionality of host-relevant metabolic pathways. It is relatively simple to check in bacterial isolates or metagenomic data from the human microbiome if these mutants do indeed exist, where human populations are actually exposed to the studied

chemicals for a long time and under the biologically relevant concentrations. Absent this data or similar one presented from new experiments, the current results are indeed confounded by the unrealistically high concentrations and exposures used.

12. This may appear trivial but it is an extremely serious issue, especially in a high-profile microbiology journal that should abide to the highest standards: I encourage the authors to study the real definition of "commensals" and "commensalism" before using this terminology. Commensalism, mutualism, parasitism, symbiosis, etc., are all well-bound terms with accurate ecological definitions, and should not be casually used. I don't believe that we have experimental data to support calling any of the 22 organisms tested in this study "commensals" – obviously not *C. difficile*, but actually neither the remaining 21 strains.

Version 1:

Reviewer comments:

Reviewer #2

(Remarks to the Author)

The authors have addressed all the concerns raised. They have: justified the chosen concentration of chemicals for the screening with an additional analysis of chemical concentrations in foods and plasma; made growth data available; used machine learning and chemoinformatic analyses to assess whether chemical features predict bacterial toxicity; explained the rationale behind using the Com20 community to study the effects of chemicals at the community level. This highly relevant study advances our understanding of how industrial and agricultural chemicals can affect the gut microbiota. I would like to commend the authors for their detailed work, which is of great value to the field.

Reviewer #3

(Remarks to the Author)

The manuscript has been greatly improved, overinterpretations have been toned down, and missing experimental procedures have been included in the revised manuscript. I thank the authors for the care they put into their response, and the explanation at the beginning of the rebuttal and in various parts of the revised manuscript, especially about the motivations of the experiments and interpretations of the results.

I am sorry to keep pushing on the following point, though. It is the authors' choice to keep using "commensal" to describe the panel, which is still erroneous. Can you really prove that these microbes do not provide any benefit nor harm to the host, after all we know from gut microbiome studies? Having been used erroneously in the past does not make it right.

Decision Letter:

Our ref: NMICROBIOL-24092928A

1st September 2025

Dear Kiran,

Thank you for submitting your revised manuscript "Off-purpose activity of industrial and agricultural chemicals against human gut bacteria" (NMICROBIOL-24092928A). It has now been seen by the original referees and their comments are below. The reviewers find that the paper has improved in revision, and therefore we'll be happy in principle to publish it in Nature Microbiology, pending minor revisions to satisfy the referees' final requests and to comply with our editorial and formatting guidelines.

Thank you again for your interest in Nature Microbiology Please do not hesitate to contact me if you have any questions.

Sincerely,

Reviewer #2 (Remarks to the Author):

The authors have addressed all the concerns raised. They have: justified the chosen concentration of chemicals for the screening with an additional analysis of chemical concentrations in foods and plasma; made growth data available; used machine learning and chemoinformatic analyses to assess whether chemical features predict bacterial toxicity; explained the rationale behind using the Com20 community to study the effects of chemicals at the community level. This highly relevant study advances our understanding of how industrial and agricultural chemicals can affect the gut microbiota. I would like to commend the authors for their detailed work, which is of great value to the field.

Reviewer #3 (Remarks to the Author):

The manuscript has been greatly improved, overinterpretations have been toned down, and missing experimental procedures have been included in the revised manuscript. I thank the authors for the care they put into their response, and the explanation at the beginning of the rebuttal and in various parts of the revised manuscript, especially about the motivations of the experiments and interpretations of the results.

I am sorry to keep pushing on the following point, though. It is the authors' choice to keep using "commensal" to describe the panel, which is still erroneous. Can you really prove that these microbes do not provide any benefit nor harm to the host, after all we know from gut microbiome studies? Having been used erroneously in the past does not make it right.

Version 2:

Decision Letter:

8th October 2025

Dear Professor Patil,

I am pleased to accept your Article "Industrial and agricultural chemicals exhibit antimicrobial activity against human gut bacteria in vitro" for publication in Nature Microbiology. Thank you for having chosen to submit your work to us and many congratulations.

Authors may need to take specific actions to achieve compliance with funder and institutional open access mandates. If your research is supported by a funder that requires immediate open access (e.g. according to [a href="https://www.springernature.com/gp/open-science/plan-s-compliance"> Plan S principles](https://www.springernature.com/gp/open-science/plan-s-compliance) or the [a href="https://www.springernature.com/gp/open-science/us-federal-agency-compliance"> NIH public access policy](https://www.springernature.com/gp/open-science/us-federal-agency-compliance)) then you should select the gold OA route, and we will direct you to the compliant route where possible. Because authors warrant under our subscription licensing terms that they haven't committed to licensing any version of their article under a licence inconsistent with the terms of our agreement – including the applicable embargo period – publication under the subscription model isn't

suitable for authors whose funders require no embargo.

With kind regards,

P.S. Click on the following link if you would like to recommend Nature Microbiology to your librarian
<http://www.nature.com/subscriptions/recommend.html#forms>

** Visit the Springer Nature Editorial and Publishing website at http://editorial-jobs.springernature.com?utm_source=ejP_NMicro_email&utm_medium=ejP_NMicro_email&utm_campaign=ejp_NMicro for more information about our career opportunities. If you have any questions please click [here](mailto:editorial.publishing.jobs@springernature.com).

Off-purpose activity of industrial and agricultural chemicals against human gut bacteria

We thank the reviewers for their constructive comments. These have enriched the results and helped us improve the clarity. We have addressed each point as described below, and our responses are marked in green.

Summary of the key changes and the aim of the study

The importance of studying the impact of widespread chemicals on gut bacteria is self-evident. The question is how do we study chemical-bacteria interactions in a manner that is relevant for applications such as policy and better chemical design? The 'relevance' can have multiple facets, the two main ones being, a) inferring a safe limit for existing chemicals from a microbiota perspective, and b) uncovering the general principles underlying chemical-bacteria interactions.

The former indeed would require, as indicated by the reviewer #3, studying chemicals at concentrations that we know, or estimate, to be similar to those encountered by the gut microbiota. This, however, means that we must first measure faecal (or, preferably, colon) concentrations for thousands of chemicals across populations with different exposure levels since no such data is available even for a small subset of these chemicals. Each chemical then must be tested individually against a panel of gut bacteria at different concentrations. And even if such a mega effort is carried out, it is likely that much of this work will be futile since the chemicals used in agriculture and industry constantly change as the regulatory landscape and the market forces rapidly evolve. Therefore, this approach will be of limited temporal and scale relevance.

Our systematic approach to study chemical-bacteria interactions is the first essential step towards overcoming the above limitations and offers a scalable approach for building a fundamental knowledgebase with broad applicability. We acknowledge that our original manuscript did not convey this point clearly enough and did not demonstrate the usefulness of the data. In this revision we have implemented a machine learning workflow to predict antimicrobial toxicity, paving the way for computational toxicity assessment. This is analogous to how solving protein structures under 'physiologically irrelevant' conditions outside of their complex cellular context has provided deep insights into cell biology and paved the way to predictive models that are revolutionizing modern biology.

The choice of the concentration used in our study is motivated by the previous studies on pharmaceutical compounds (Maier et al. *Nature* 2018, Klünemann et al. *Nature* 2021, Gracia-Santamarina et al. *Cell* 2024) and thus enables direct comparison to these rich datasets and their integration into a single machine learning framework (New Figures 3C-D). It also should be noted that the distinction between pharmaceutical drugs and, say, pesticides, is somewhat arbitrary from a chemical perspective as illustrated in Fig. 3A in the manuscript. Screening chemical pollutants at the same concentration opens opportunities for learning patterns and principles that can be used to assess, e.g., pesticide and drug candidates that are yet to enter the market, or to help improve the specificity of new chemicals. Importantly, at 20 μM the majority of the hits show over 50% growth inhibition, with a quarter of the hits showing strong inhibition (<0.1 normalised AUC) with examples across diverse taxa.

We note that concentrations of some chemical pollutants in the blood have been found to be in the μM range for some individuals in Germany in a recent study [Braun et al. (2024) *Science*]. Importantly, for some strong inhibition hits, such as closantel, regulatory limits in food are in the μM range, with observations in dairy products to be able to reach this range. See the point-by-point response below for the key statistics. The concentrations for individuals living in highly polluted areas and with high occupational exposure (e.g. unprotected pesticide spraying, which is unfortunately common in much of the developing world), is likely to be much higher. While the relationship between blood and colon concentrations is compound and individual dependent, estimates for pharmaceuticals commonly predict higher colon than blood concentrations. There will, of course, be variation based on the individual and the chemical in question, which further emphasizes the need for systematic analysis. In the revision, we have provided a detailed summary of the findings from literature searches.

Finally, we do recognise that our original manuscript fell short of articulating and illustrating the value of a systematic approach. We have now conducted chemoinformatics/ML analysis and mechanistic follow up on genetic screens, both of which help illustrate the fundamental value of our unique dataset. For the former, we collaborated with ML expert/s and use state-of-the art tools.

Summary of the original revision plan with new analyses/experiments

Critical reviewers point	Summary of the original revision plan	Results of the revision/implemented changes
---------------------------------------	---

R2.3 and R3.3 - Chemical similarity	Chemoinformatics and ML analysis for identifying chemical features enriched in broad-spectrum compounds, and for assessing the predictive power of the combined drug-chemical dataset. For this, we will use state-of-the-art ML tools and the recently released chemical feature library by IBM - MolFormer.	We have successfully implemented a machine learning framework based on random forest classifiers which accurately predict toxicity in species and compound-class specific models. We describe the performance across different compound and feature sets and identify subsets of important structural features.
R3.11 - Chemical-genetic screen validation (mutant library)	Extensive experimental follow-up through isolating gene KO mutants in P. merdae as well as using orthologs in related Bacteroidales. We will test fitness effects at different xenobiotic concentrations as well as metabolic changes.	We have isolated P. merdae mutant strains, and characterised their growth under different xenobiotic concentrations, validating TnBarSeq interpretation. Further, we show that some mutants develop cross-resistance to antibiotics. To provide further context of genetic hits in Bacteroidales, we have performed a comprehensive analysis of B. thetaiotaomicron arrayed transporter gene mutants under four pollutants. This revealed conserved compound-dependent genetic responses in efflux pathways. We also explored mutant strain phenotype with mass spectrometry to evaluate possible changes on compound bioaccumulation.
R3.4 - Validation growth screen hit overlap	Assessment of compound purity using mass-spectrometry and detailed chemical sensitivity analysis for the bacteria in question.	We have conducted LC-MS measurements of original library stocks and our own stocks used for validation/MIC screen with a broad Agilent pesticide method. Untargeted analysis detects no significant contaminants. Targeted analysis confirms similar concentration of both stocks.

		We also further analysed the discrepancies to reveal species-specificity and the impact of modulating hit calling thresholds.
R2.1 and R3.1 - Concentration/doses and parameters of growth inhibition	Detailed outline of why the 20 μM dose was chosen and the value of a systematic pan-xenobiotic toxicity dataset. Table summarising extensive literature searches for chemical concentration measurements in human samples.	We have further clarified the rationale behind the chosen dose in the main text. We have analysed a recent study from Braun et al (2024), Science which measured xenobiotic concentrations in the plasma of German mothers. The majority of samples contained at least one xenobiotic at a concentration $>10 \mu\text{M}$ (including compounds from our library), supporting the biological relevance of micromolar concentrations for screening. We also include mentions to hit compounds with regulatory limits in the μM level.
R3.7 - Mechanism of BPAF disruption of community	Mechanistic follow up: Measurements of BPAF bioaccumulation in community, which is a likely cause for cross protection, and further follow-up experiments.	LC-MS ² measurements indicate BPAF bioaccumulation in community as likely cause for the observed cross-protection.

Point-by-point response

Reviewer #1:

Remarks to the Author:

The authors performed a large-scale screening for effects of chemical pollutants on a small collection of highly prevalent gut microbiome strains, identifying hundreds of potential growth inhibition effects of pollutants on gut strains. Next, they evaluated the effective dose on which the pollutant effects inhibit strain growth. They also performed the analysis on bacterial communities rather than individual bacteria, and even used transposon knock out model to identify genes that define strain susceptibility to pollutants.

Basically, any of the follow-up experiments I can think of are already done. This is an incredible study both in terms of scientific and public importance. I could think only of some textual corrections, like:
p111. Contamination ... are -> contamination ... is
p.14128: The gut microbiota are - could be use to use 'is' here
p15132: anti-gut bacterial - should be rephrased to just 'antibacterial',

And some others. Overall, the quality of text is quite good, maybe just a bit of proofreading could help.

We thank the reviewer for their positive evaluation of our manuscript and constructive suggestions for improvement. We have addressed the language issues and extensively edited the entire text to improve grammar and clarity.

Reviewer #2:

Remarks to the Author:

This study looked at the impact of > 1000 pollutant chemicals on the growth of 22 gut bacterial strains in vitro. A total of 168 chemicals, most of which were not previously known to have antibacterial effects, reduced growth of at least one bacterial strain. A subset of these compounds was also tested in a synthetic microbial community and shown to alter community composition. The authors proceeded to analyse the genetic basis of such off-target effects of these pollutants on the microbiota. Membrane transport and polarity were identified as key modulators of off-target toxicity. Chemical resistance mutants included those deficient in the biosynthesis of beneficial metabolites, such as branched short-chain fatty acids, which are relevant to human health. However, the effective capacity of chemicals such as TBPPA to reduce the production of such metabolites by microbes such as *P. merdae* awaits further validation.

This is an important study that expands our understanding of the ability of xenobiotics to impact the human microbiota. The study was carefully designed, controls seemed largely appropriate, and effort was made to replicate results for compounds with hits. I have a few points related, e.g., to the relevance of the concentration tested or presentation of results, that I would like to see clarified/discussed before publication.

We thank the reviewer for the constructive comments, which have helped us greatly to improve the clarity and completeness of our manuscript.

We have conducted several follow-up experiments on our genetic screen in *P. merdae*. We validated growth inhibition and promotion phenotypes in *P. merdae* isolate mutant strains across a range of xenobiotic concentrations. We also demonstrated that some of these mutants develop cross-resistance to antibiotics such as ciprofloxacin. Additionally, we evaluated *B. thetaiotaomicron* isolate mutants (Arjes et al., 2022) under pollutant stress and found that some compound-dependent efflux responses are conserved between *P. merdae* and *B. thetaiotaomicron*. The results from the new experiments are shown in the updated Figure 4, and Supporting Figures 4-6, and the corresponding updated main text.

Figure 4: Chemical-genetic screens in Bacteroidales reveal genes that modulate susceptibility and resistance to xenobiotics.

- Overview of closantel experiment and representative TnBarSeq insertion plot. The genetic context of *acrR* efflux regulator (*NQ542_01170*) is shown with genes represented as coloured arrows in scale. The black track above represents TA dinucleotide sites, which are potential transposon insertions sites, and black lines represent TnBarSeq insertion sites and read count.
- Relative abundance of transposon mutants in the *acrR* locus. Mutants show strong positive selection under closantel 2.5 μM and TBBPA 20 μM.
- Growth curves of isolated transposon mutants and wild type *P. merdae* under closantel (2 μM). Strains include independent insertions in *acrR* coding and promoter regions (n=7), wild type strains (WT, n=6), and Tn insertion control (n=2). Isolation condition of *acrR* mutants is represented as solid line for closantel or dashed line for TBBPA. DMSO control shown in Supplementary Figure 6C.
- Dose-response curves under closantel and ciprofloxacin of *acrR* mutants and control strains. Data points represent optical density values at 24 hours of culture from technical duplicates across three independent experiments. The two-fold dilution series highest tested concentration of ciprofloxacin 603.6 μM is equivalent to 200 μg/mL.

- E) Heatmap of representative TnBarSeq top gene hits. The colour represents the $\log_2(\text{fold-change})$ in Tn mutant abundance compared to end point control. Red indicates positive conditional fitness (resistance) and blue conditional negative fitness (hypersensitivity). Statistically significant hits $P_{\text{adj}} < 0.05$ absolute $\log_2(\text{fold-change}) > 0.25$ are marked with an asterisk. Gene functional annotation categories are colour coded.
- F) Effect of xenobiotics on growth of an arrayed transporter mutants of *B. thetaiotaomicron*. Axis show normalised relative area under the curve per mutant (nrAUC= per mutant average rAUC xenobiotic / average rAUC DMSO), as described in methods. Hits are defined by an absolute effect in growth of >20% and $p\text{-value} < 0.05$.
- G) Representative TnBarSeq insertion plot on the inositol lipid biosynthesis gene cluster in PFNA (500 μM) and DMSO control. Homology to the characterised inositol lipid gene cluster from *B. thetaiotaomicron* is shown using Clinker.
- H) Growth curves of isolated *P. merdae* transposon mutants of the putative phosphatidylinositol phosphate phosphatase (PIPPh) gene *NQ542_07540* from the inositol lipid biosynthetic gene cluster, WT, and transposon insertion control (*Tn::acrR*) under PFNA (250 μM). Curves for two independent insertion strains are shown, and values are the average of two technical replicates.

2.1 - Concentrations

It is mentioned in the introduction that some pollutants have been measured in human blood or urine. How do available concentrations from urine and blood compare with the 20 μM used in the screen? or with the concentrations determined by the toxicokinetic modelling applied in this study?

We appreciate this point. We now better explain the reasoning of our choice of dose (please see the beginning of this letter, "Summary of key changes and aim of the study"). Additionally, we have conducted a search of the literature and provide examples of observed concentrations. For example, Braun et al, *Science* (2024) report concentrations of xenobiotics in the plasma of pregnant women and identify hundreds of instances where concentrations fall in the same order of magnitude as tested in our screen, many from compounds we have tested. Additionally, we now cite studies which have reported micromolar concentrations of closantel, one of the most active compounds on our screen, in dairy products derived from animals treated with this antiparasitic compound. Importantly, the regulatory Maximum Residue Limits (MLR) for closantel in Great Britain for some products of animal origin are above the μM level.

In main text:

"To further contextualise this dose, we analysed a recently published study of xenobiotic plasma concentrations (Braun et al., 2024). Out of 592 samples, the majority (305) contained at least one xenobiotic at a concentration of 10 μM or higher. Among the measured pesticides, spinosyn A, melamine, mepiquat, acrylamide, and amitrole, which are all included in our compound library, had the highest number of samples with concentrations above 2 μM in plasma (45, 40, 39, 34 and 33 respectively). Pharmacokinetic studies suggest that colonic concentrations to be in the same range or higher than observed in plasma (Hens et al., 2016; Maier et al., 2018). The relevance of micromolar range doses is further supported by micromolar range regulatory limits in food for certain compounds such as the antiparasitic closantel that can reach micromolar concentrations in, for

example, dairy products (Iezzi et al., 2014; Power et al., 2013; Veterinary Medicines Directorate, 2022).“

2.2. Report full AUC data

Page 5, lines 10-13 and Supplementary Data 3: I could only find reported whether a compound significantly impacted growth (> 20% reduction in AUC, True or False), and not the magnitude of such reduction. Why? A 90% inhibition in growth may have quite different repercussions, e.g. to the microbiota community and even the host, compared to a 20.1% growth inhibition. These results would be of interest to the research community.

In addition, in such a large collection of compounds, is it true that none had a positive impact on bacterial growth compared to DMSO? Can the authors report on that or explain in the manuscript why was growth promotion not investigated (e.g., due to the experimental set-up or rich medium used).

We apologise for our oversight regarding the availability of this data. We have included AUC, normalised AUC, z-score, p-value and adjusted p-values in **Supplementary Table 3** and provide the full raw growth data in a Mendeley Data repository (see **Data Availability**). We also include an additional figure to illustrate that most hits show over 50% growth inhibition, with a quarter of hits showing over 90% inhibition (**Figure 1E**).

Supplementary Data 3. Growth screen results (related to Fig 1C)

Strain ID	Strain	Compound	Compound ID	Median AUC	Median normalised AUC	Median zscore	Median p-value	Median adj-p-value	Significant growth inhibition	
NT5021	Akkermansia muciniphila	(-)-NICOTINE	3_F7		0.373	0.7534	-0.7964	0.42578784	0.98908	FALSE
NT5021	Akkermansia muciniphila	(+)-DICLOFOP	12_C6		0.571	0.81166	-1.6214	0.1049253	0.57016	FALSE
NT5021	Akkermansia muciniphila	(AMINOMETHYL)PHOSPHONIC ACID	4_G3		0.537	1.04251	0.54336	0.51035547	0.93345	FALSE
NT5021	Akkermansia muciniphila	(E)-METOMINOSTROBIN	7_F5		0.716	0.88293	-0.9266	0.3541463	0.9951	FALSE
NT5021	Akkermansia muciniphila	(E)-MEVINPHOS SOLUTION	9_A1		0.849	1.08498	0.97894	0.32761074	0.96316	FALSE
NT5021	Akkermansia muciniphila	(S)-3-ANILINO-5-METHYL-5-PHENYLIMIDAZOLI	7_E4		0.704	0.93122	-0.713	0.47587066	0.9951	FALSE
NT5021	Akkermansia muciniphila	(Z)-METOMINOSTROBIN	7_E5		0.7075	0.87265	-0.9402	0.34709631	0.95258	FALSE
NT5021	Akkermansia muciniphila	(Z)-MEVINPHOS SOLUTION	8_H1		0.8865	1.05473	0.41294	0.67314857	0.95159	FALSE
NT5021	Akkermansia muciniphila	1-NAPHTHOL	4_C9		0.498	1.00252	0.01882	0.30182049	0.99694	FALSE
NT5021	Akkermansia muciniphila	1-NAPHTHYLACETAMIDE	9_G4		0.7865	1.00773	0.10305	0.9179264	0.9951	FALSE
NT5021	Akkermansia muciniphila	1-NAPHTHYLACETIC ACID	7_E11		0.7975	0.98366	-0.1207	0.82102439	0.9951	FALSE

In main text:

“Circa 150 chemical-bacteria interactions featured over 90% decrease in growth, uncovering the strong off-purpose anti-gut bacterial activity of 33 compounds (**Figure 1E**).”

New Figure 1E:

E) Distribution showing the strength of growth inhibition estimated as mean normalised AUC values (relative to plate-matched DMSO-controls) of hits.

We consider growth promotion out of scope of this study, but we are making raw data including growth curves available. To contextualise from Supplementary Data 3, while 168 compounds have significant growth inhibition (>20% inhibition), only 17 compounds suggest cases of growth promotion (>20% increase). Previous reports in drug-bacteria interactions also indicate small number of such effects (e.g. in Klunemann et al. 2021 study). We do not follow up on those cases in the present study since our focus is on inhibitory effects but we do agree that following these in the future may hold some insights for chemical-bacteria interactions.

2.3. Chemical similarity

Figure 4A, B: the pollutants tested are chemically distinct from previously tested pharmaceutical drugs. But are pollutants with microbiota hits and pharmaceuticals with microbiota hits chemically distinct? Or is there an overlap/similarity in terms of their chemical structures?

Of the 168 compounds shown to affect at least one microbiome member, only a subset (24) shows a broad toxicity range. What is the chemical diversity of compounds with such a broad spectrum? Can the authors highlight these compounds in Figure 4A?

Thank you for these suggestions. We have generated a new UMAP plot, shown below, with broad spectrum chemicals (>1/3 of species inhibited) highlighted which shows that these are distinct and covering a wide chemical space.

To address this more comprehensively and in an unbiased manner, we now include a new section '**Machine learning predicts antimicrobial effects**' where we performed in-depth chemoinformatic analyses and combined this with machine learning approaches for identifying assessing whether chemical features are predictive of anti-bacterial effects and to identify specific chemical features enriched in broad-spectrum compounds (see new **Figure 3**, reproduced below). Additional analysis of chemical features is reported in new **Supporting Figure 3**.

Figure 3: The chemical landscape of xenobiotic-bacteria growth interactions

A) Uniform Manifold Approximation and Projection (UMAP) for Extended-Connectivity Fingerprints (ECFPs) of library compounds from this study, previously screened pharmaceutical drugs (Maier et al., 2018) and a random selection of 250,000 compounds from PubChem.

B) Distribution of similarities between closest matching pollutants from this study and drugs used by Maier et al. 2018. The two compound sets are structurally diverse and distinct, as only very few (6) compounds from the library were used in this study had a similarity >0.75 to a drug from in (Maier et al., 2018). Compounds shared between the two studies ($n=32$, mostly veterinary drugs) were excluded from this analysis.

C) Machine learning workflow to predict species-specific xenobiotic toxicity from molecular descriptors and embedded features. Random forest classifier models were trained and evaluated using 20x cross-validation, using performance metrics appropriate for imbalanced datasets.

D) Performance scores of models trained and tested on various feature and sample sets. Left: Models trained on pesticides only ($n=801$) using different feature sets (x-axis). Middle: Model trained on pesticides and pharmaceutical drugs from (Maier et al., 2018) ($n=1940$). Right: Models trained on pesticides [this study] cannot predict toxicity of pharmaceutical drugs (Maier et al., 2018) and vice versa.

Each dot represents a species (mean of 20x cross-validation splits). Random model performances are overlaid (grey bars/dots) and were obtained by randomly shuffling the labels.

2.4. Com20 rationale

Section “Chemicals alter the composition of synthetic gut bacterial communities”: what was the rationale for selecting a synthetic community containing bacterial strains distinct from the strains tested in pure culture?

We opted to use an established model community which grows reproducibly and stably in mGAM medium (which would most likely not have been the case had we simply combined all species tested in monocultures). Com20 is phylogenetically and functionally diverse, spanning 6 bacterial phyla, 11 families, and 17 genera. Together, these 20 species encoded 246 MetaCyc metabolic pathways, which represents 61.3 % of the 372 pathways present in healthy human gut microbiomes (Müller et al. 2024). We have added an explanation of the rationale to the main text and have expanded strain ID details of the Com20 community members in Supplementary Data 4.

In main text:

“Com20 is an established model community which is phylogenetically and functionally diverse, spanning 6 bacterial phyla, 11 families, and 17 genera. Together, these 20 species encoded 246 MetaCyc metabolic pathways, which represents 61.3 % of the 372 pathways present in healthy human gut microbiomes. Members of the community grow together stably and reproducibly in mGAM (Müller et al., 2023).”

Other points:

-Materials and methods, under “statistical analysis of growth”: what do the authors mean by “control” wells? Why was AUC normalisation for industrial chemicals and mycotoxins done differently – there were no edge effects for the other plates?

Control wells refers to DMSO (vehicle control) wells. We have clarified this in the manuscript.

The screening of industrial chemicals and mycotoxins was conducted first, and this dataset contains noticeable edge effects which we correct for using appropriate row/column-median normalisation. We subsequently optimised the protocol (by wrapping plates during the overnight incubation to become anaerobic) which minimised evaporation and edge effects. Therefore, it is appropriate to process these

two datasets slightly differently; this does not cause any bias since hits and fold-changes are defined plate-wise.

-Figure 1 C: In addition to the total number of compounds tested, could be useful to add the total number of compounds tested from each type to each of the main coloured rectangles.

Thank you for this suggestion, we have adapted the **Figure 1B** accordingly:

-Page 9, line 31: please define “KS test”.

Thank you, we have added this information (“Kolmogorov–Smirnov test”).

Reviewer #3:

Remarks to the Author:

In this manuscript, Lindell et al present a large screen of more than 1000 pollutants and non-pharmaceutical chemicals against 22 members of the human gut microbiome, with a direct readout of growth inhibition. They uncover hundreds of potential interactions, follow up on a few of them to calculate MICs,

and perform a genetic screen to identify resistance mechanisms to these chemicals.

The experiments are executed in a similar manner to what was performed previously by the same authors with pharmaceuticals, and the direct screen results are compared to those from the earlier study. I do have several concerns about the initial rationale of the study design, relevance to the real world, and the overinterpretation of the results.

We would like to thank the reviewer for the detailed comments. The main aim of this study was to create the first large-scale map of pollutant/contaminant-bacteria interactions and to create a dataset that integrates with the prior work on pharmaceutical drugs. Please see the beginning of this letter, "*Summary of the key changes and aim of the study*".

In brief, we agree that our work raises many important questions, including which of these specific interactions are relevant in the context of the human gut. Yet, we hope that the reviewer will also recognise the novelty of our work, the considerable amount of basic knowledge that it represents, and the essential starting point it provides for future studies, both basic and applied. We acknowledge that the present work could be improved in several areas raised by the reviewer and have conducted new analyses and deeper investigations as outlined below.

3.1. Concentrations

I understand that a 20 μ M concentration of the chemicals in the screen was selected as a means of comparison to the previous pharmaceuticals screen. While this concentration may be relevant to pharmaceuticals that are given at high dosages and over extended periods of time, it is unclear if this concentration is at all relevant to pollutants that are typically found in trace amounts in food and water. This is also exaggerated by the fact that the authors present their data in a binary mode (significant/not significant) based on a very limited inhibition cutoff (20%), without clear reflection of the magnitude of inhibition in their hits. Based on the concentrations shown in the few cases of MIC determination, it looks like most of the cases will have limited growth inhibition at micromolar concentrations. The authors should either rationalize the use of such high concentration based on real world exposure data, or repeat the screen at a lower concentration. This statement is really misleading given the presented data from the MIC experiment: "Many compounds show strong inhibitory effects at substantially lower concentrations than were tested in the

main screen. E.g., imazalil sulfate and prochloraz inhibited *Eubacterium rectale*, a prevalent species that produces health-associated metabolite butyrate, at a concentration as low as 2.5 μM . MICs are not even presented in the main text, even though the goal of the study was to determine MICs.

Magnitude of inhibition and data availability

We apologize for our oversight regarding the availability of this data. We have included quantitative effect sizes in **Supplementary Table 3** : AUC, normalised AUC, z-score, p-value and adjusted p-values; and provide the full raw growth data in a Mendeley Data repository (see **Data Availability**). We also include an additional figure to illustrate that most hits show over 50% growth inhibition, with a quarter of hits showing over 90% inhibition (**Figure 1E**).

We also included in the main text:

“Circa 150 chemical-bacteria interactions featured over 90% decrease in growth, uncovering the strong off-purpose anti-gut bacterial activity of 33 compounds (**Figure 1E**).”

In Figure 1E:

Supplementary Data 3. Growth screen results (related to Fig 1C)

Strain ID	Strain	Compound	Compound ID	Median AUC	Median normalised AUC	Median zscore	Median p-value	Median adj-p-value	Significant growth inhibition	
NT5021	Akkermansia muciniphila	(-)-NICOTINE	3_F7		0.373	0.7534	-0.7964	0.42578784	0.98908	FALSE
NT5021	Akkermansia muciniphila	(+)-DICLOFOP	12_C6		0.571	0.81166	-1.6214	0.1049253	0.57016	FALSE
NT5021	Akkermansia muciniphila	(AMINOMETHYL)PHOSPHONIC ACID	4_G3		0.537	1.04251	0.54336	0.51035547	0.93345	FALSE
NT5021	Akkermansia muciniphila	(E)-METOMINOSTROBIN	7_F5		0.716	0.88293	-0.9266	0.3541463	0.9951	FALSE
NT5021	Akkermansia muciniphila	(E)-MEVINPHOS SOLUTION	9_A1		0.849	1.08498	0.97894	0.32761074	0.96316	FALSE
NT5021	Akkermansia muciniphila	(S)-3-ANILINO-5-METHYL-5-PHENYLIMIDAZOLI	7_E4		0.704	0.93122	-0.713	0.47587066	0.9951	FALSE
NT5021	Akkermansia muciniphila	(Z)-METOMINOSTROBIN	7_E5		0.7075	0.87265	-0.9402	0.34709631	0.95258	FALSE
NT5021	Akkermansia muciniphila	(Z)-MEVINPHOS SOLUTION	8_H1		0.8865	1.05473	0.41294	0.67314857	0.95159	FALSE
NT5021	Akkermansia muciniphila	1-NAPHTHOL	4_C9		0.498	1.00252	0.01882	0.30182049	0.99694	FALSE
NT5021	Akkermansia muciniphila	1-NAPHTHYLACETAMIDE	9_G4		0.7865	1.00773	0.10305	0.9179264	0.9951	FALSE
NT5021	Akkermansia muciniphila	1-NAPHTHYLACETIC ACID	7_E11		0.7975	0.98366	-0.1207	0.82102439	0.9951	FALSE

Concentration context:

We have performed literature searches and have added more context on compound concentration in cohort studies, food regulatory limits and food concentration values of key compounds from our screen. To contextualise the relevance of the 20 μM dose, we have re-analysed a recently published dataset by Braun et al, *Science* (2024), which reports concentration of hundreds of xenobiotics in the plasma of pregnant women. For 576 compounds (the compounds left after excluding analytes annotated as 'endogenous' and 'pharmaceuticals', as well as caffeine and the amino acid breakdown product p-tolylsulfate), measured across 592 samples, they report 601 instances where the concentration of the xenobiotic was in the same range as used in our screen ($>10 \mu\text{M}$) [see Table S10]. In 1842 instances it was within the same order of magnitude ($>2 \mu\text{M}$). Several included in our study were observed at concentrations $>2 \mu\text{M}$, e.g. Spinosyn (pesticide, $n=45$ plasma $>2 \mu\text{M}$), Melamine (industrial chemical $n=40$), Mepiquat ($n=39$), Acrylamide ($n=34$), amitrole ($n=33$); and several others which are all included in our compound library.

Plasma concentrations are not directly translatable to colon concentrations but can be indicative. Maier et al *Nature* (2018) estimated that in 161 out of 166 cases, colon concentrations are higher than plasma concentrations (Supplementary Table S1a). In any case, the data from Braun et al shows that xenobiotic concentrations in the human body in the real world can sometimes reach concentrations similar to those tested in our screen. Further, the Braun et al. study was conducted in a developed country with the highest regulatory standards. The pollutant concentrations are likely to be much higher in populations living in developing countries.

Importantly, we also observe that the regulatory limits in food products for some compounds from the screen are above the μM level in Great Britain, such as the antiparasitic closantel (one of the top growth inhibition hits). Additionally, we now cite studies which have reported micromolar range concentrations of closantel in dairy products of animals treated with antiparasitic drug (for example concentrations of 7000 $\mu\text{g}/\text{kg}$ of closantel in dairy products).

We have revised the main text with references as follows:

“To further contextualise this dose, we analysed a recently published study of xenobiotic plasma concentrations (Braun et al., 2024). Out of 592 samples, the majority (305) contained at least one xenobiotic at a concentration of 10 μM or higher. Among the measured pesticides, spinosyn A, melamine, mepiquat, acrylamide, and amitrole, which are all included in our compound library, had the highest number of samples with concentrations above 2 μM in plasma (45, 40, 39, 34 and 33 respectively). Pharmacokinetic studies suggest that colonic concentrations to be in the same range or higher than observed in plasma (Hens et al., 2016; Maier et al., 2018). The relevance of micromolar range doses is further supported by micromolar range regulatory limits in food for certain compounds

such as the antiparasitic closantel that can reach micromolar concentrations in, for example, dairy products (Iezzi et al., 2014; Power et al., 2013; Veterinary Medicines Directorate, 2022).“

Study goal and possible limitations

The main goal of this study was to build a framework and discover general principles of xenobiotic-bacteria interactions (See above “Summary of key changes and aim of the study”). To probe the utility of this framework, we include have included a new section ‘**Machine learning predicts antimicrobial effects**’.

Further, we include in the discussion:

“Towards tackling this problem, our study demonstrates feasibility of computational toxicity predictions that can be used at scale and for safe-by-design product development. This proof-of-concept was possible due to the here-expanded chemical space underscoring the need for standardised chemical-bacteria screens.”

We fully acknowledge that intestinal concentrations will vary based on individual exposure, gut physiology, and compound pharmaco-kinetics/dynamics. It would be impossible to factor these into a high-throughput study like ours; factoring it may also turn out to be a futile exercise as discussed above in “Summary of key changes and aim of the study”: We would first have to measure faecal (or, preferably, colon) concentrations for thousands of chemicals across populations with different exposure levels since no such data is available even for a small subset of these chemicals. Each chemical then must be tested individually against a panel of gut bacteria at different concentrations. And even if such a mega effort is carried out, it is likely that much of this work will be futile since the chemicals used in agriculture and industry constantly change as the regulatory landscape and the market forces rapidly evolve.

Instead, as the first large-scale screen of pollutant-bacteria interactions, we map potential interactions and do not claim that all of the identified interactions will pose a danger to humans in the real world. Instead, we envisage that our study will help the design of future cohort studies.

We have now included a note to this in the discussion:

“Are our results relevant for real world exposure scenarios in humans? Many of the compounds in our study, e.g., melamine and mepiquat, have been found in micro-molar concentrations in human blood (Braun et al., 2024) implying similar or higher concentrations in the gastrointestinal

tract. The concentrations for individuals living in highly polluted areas and with high occupational exposure (e.g., unprotected pesticide spraying, which is unfortunately common in much of the developing world), is likely to be much higher. Yet, currently there is no cohort data available that tracks both chemical exposure and microbiome dynamics. Thus, our study is limited in bridging to *in vivo* impact on gut bacteria. Going forward, we envisage that our pollutant-bacteria screen will encourage collection of exposure data in microbiome studies, similarly to how recording of drug usage data in microbiome cohort studies was accelerated by *in vitro* drug-bacteria screens (Klünemann et al., 2021; Maier et al., 2018). Nevertheless, *in vivo* assessment of the impact of chemicals exposure is likely to be difficult due to complex interactions between the microbiota and diet, medications, and other lifestyle factors. Genetic and physiological insights gained from our study will therefore help in bridging *in vitro* studies with *in vivo* datasets and in designing cohort studies.”

3.2. Phyla analysis

Figure 2a shows the proportions of strains from each phylum that are sensitive to the different chemicals. This analysis is completely biased by the small sample size of the bacterial strains tested here, and the limited sampling in most Phyla. For example, a 9/9 of the Bacteroidetes shows exactly the same result as a 1/1 of the Fusobacteria, which is not a meaningful number to base a whole Phylum results on. Proteobacteria are represented by two strains, both of which are *E. coli* – proportion there means nothing. Verrucomicrobia is also 1, and Actinobacteria is 2....This entire analysis is flawed in my opinion.

We agree and have therefore replaced this figure with a PCA plot (New **Figure 2E**, reproduced below) previously shown in the supplement as this better illustrates the point we were trying to make (i.e. that there is a phylogenetic signal in the sensitivity profiles). The heatmap is now the new **Supplementary Figure 2A** as it illustrates interesting differences between Bacteroidetes and Firmicutes, like the sensitivity of Firmicutes to conazole fungicides.

3.3. Structure of broad spectrum chemicals

An in depth structural analysis of the chemicals that show broad versus specific effects is more useful than the information about their use in EU or US, and may reveal actual features of these molecules that can predict their toxicity (this was partially done with one example in the manuscript, benzoate).

We agree and have performed additional chemoinformatics analyses in new section '**Machine learning predicts antimicrobial effects**', with new **Figure 3C-D** (reproduced below) and **Supporting Figure 3**. We have used machine learning approaches and a recent chemical feature library by IBM – MolFormer (Ross et al., 2022) to identify chemical features enriched in broad-spectrum compounds, and to assess the predictive power of the combined drug-chemical dataset.

Feature analysis is further described in new Supporting Figure 3:

Supplementary Figure 3

A) Coverage of chemical space is increased by our new, complementary xenobiotic compound library. The plot illustrates the fraction of compounds in the random PubChem subset (y-axis, $n=250,000$) which have a similar compound (at the Tanimoto similarity threshold indicated on the x-axis), for the compound library of this study (green line), pharmaceutical drugs screened by Maier et al 2018 (purple line) and the combined dataset. The inset focuses on a Tanimoto threshold of 0.75, where significantly increased coverage of the chemical space is observed ($p < 2.2e-16$, Kolmogorov-Smirnov test). **B)** Heatmap including all features which belonged to the top10 features in at least one species (ranked by mean across 20 cross-validation sets). Feature importance was accessed using the feature_importances_ attribute of the RandomForestClassifier, which relies on decrease in mean impurity to quantify feature importance. Feature based on molecular distance edge descriptors of carbon atoms (MDEC) (S. Liu et al., 1998) stood out as particularly useful for predicting anti-gut-bacterial activity across different species. **C)** Correlation analysis of the features with each other. **D)** Comparison of feature importances between pesticides and pharmaceutical drugs indicates that similar features are important for both compound classes. Spearman $r=0.65$, $p=0$.

3.4. Validation screen overlap

Only 75% replicated between the original screen and the MIC experiment, can the author explain the reason behind the failure of the remaining 25%?

We have investigated the reason for the differences between the original screen and the MIC/validation experiment. We first ruled out issues with the compound stocks. We then further investigated which species and compounds were responsible for inconsistencies. We have clarified in the text and included additional Supporting Data.

1. We tested the original chemical stocks received from the Chemical Biology Facility of EMBL and our own stocks (used for the validation experiment) for contaminants using a broad LC-MS method developed by Agilent which covers ~700 pesticides ('Agilent MassHunter Pesticides PCDL'). This did not detect any cross-contamination of either of the stocks.

Identified compound	Sample	Mass	RT	Width	Height	Area	Score	Base Peak	Ions	Diff (ppm)	Score (Tgt)	Flags (Tgt)	Score (Frag Coelution)
Propiconazole(l)	our_PROPICONAZOLE_r1.d	341.0700	7.201	0.060	1028501	4366252	99.47	342.0774	6	0.50	99.47	multiple IDs;Qualified	97.44
Pyraclostrobin	lib_PYRACLOSTROBIN_r1.d	387.0987	7.222	0.087	2290239	12940562	99.46	388.1062	6	0.38	99.46	Qualified	97.99
Emamectin B1a	lib_EMAMECTIN BENZOATE_r1.d	885.5242	7.619	0.061	1277096	5284540	99.03	886.5318	5	0.40	99.03	Qualified	95.64
Emamectin B1a	our_EMAMECTIN BENZOATE_r1.d	885.5241	7.600	0.030	1242733	2178995	98.85	886.5316	5	0.26	98.85	Qualified	94.37
Prochloraz	lib_PROCHLORAZ_r1.d	375.0311	7.253	0.039	1631557	5348146	98.76	376.0385	7	0.64	98.76	Qualified	93.45
Propiconazole(l)	lib_PROPICONAZOLE_r1.d	341.0698	7.204	0.060	1032368	4303305	98.70	342.0774	6	0.18	98.70	multiple IDs;Qualified	97.21
Prochloraz	our_PROCHLORAZ_r1.d	375.0310	7.249	0.063	1223453	5618018	98.69	376.0385	7	0.46	98.69	Qualified	96.96
Propiconazole(l)	our_PROPICONAZOLE_r1.d	341.0695	6.871	0.078	68537	367695	98.05	121.0509	6	-0.88	98.05	multiple IDs;Qualified	95.09
Imazalil (Enilconazole)	our_IMAZALIL SULFATE_r1.d	296.0486	5.648	0.123	2506423	18362844	97.86	297.0560	6	0.87	97.86	Qualified	98.64
Pyraclostrobin	our_PYRACLOSTROBIN_r1.d	387.0989	7.218	0.088	2197036	12528730	97.23	388.1065	5	0.74	97.23	Qualified	98.74
Imazalil (Enilconazole)	lib_IMAZALIL SULFATE_r1.d	296.0485	5.655	0.117	2641284	18442961	96.47	297.0560	5	0.75	96.47	Qualified	98.39
Propiconazole(l)	lib_PROPICONAZOLE_r1.d	341.0695	6.890	0.087	58320	329700	96.09	121.0509	6	-0.72	96.09	multiple IDs;Qualified	96.69
DNOP / Dioctyl phthalate	blank_9.d	390.2770	9.030	0.051	11984	41095	95.74	121.0509	3	0.05	95.74	Qualified	90.00
DNOP / Dioctyl phthalate	blank_6.d	390.2769	9.036	0.041	12320	41194	82.42	121.0509	2	-0.37	82.42	Qualified	91.16
DNOP / Dioctyl phthalate	lib_PROPICONAZOLE_r1.d	390.2769	9.039	0.063	10558	58562	80.35	121.0509	2	-0.31	80.35	Qualified	94.77

Response Table 1: Analysis of pesticide stocks by LC-MS. DMSO stocks from EMBL (prefix 'lib') or our own validations stocks (prefix 'our') were diluted in water to a final concentration of 20 µM (same as used for screening). 1 ul was injected and analysed using a 15 min reverse phase chromatography method and an Agilent QTOF instrument operated in DIA ('All Ions') mode with 4 different collision energies. The data was analysed using MassHunter Qualitative Analysis using the Agilent Pesticides PCDL. Library matches were filtered to remove those with flags 'Not qualified', 'low score' and 'No library spectrum'. The remaining identifications are shown in the table above. The compound identified by LCMS (first column) matches those expected in the sample (second column). A potential contamination with DNOP was detected although this is of low concern as is it also present in the water blank (i.e. not originating from the DMSO stocks) and the peak area is very low.

2. Targeted analysis of the same data using Skyline software to quantify peak areas confirmed similar peak areas in our stocks versus EMBL library stocks, indicating approximately equal concentrations.

Gut bacteria sometimes exhibit inconsistent growth behaviour across experimental batches. Especially *Lactobacillus gasseri* and *Streptococcus salivarius* behaved differently in the validation screen compared to the main screen. Tightening the AUC or p-value thresholds in the main screen had no effect on the percentage of validated hits.

We have added a note explaining this erratic behaviour of some gut bacteria and have expanded the data of main and validation screen adding **Supporting Data 3** with effect size statistics and including new **Supporting Data 4** with size statistics of validation screen.

In main text:

“The majority of the interactions from the main screen were also observed in this independent experiment (72%, 23 out of 32) (**Supporting Figure 1C+D**), which is the expected range considering the physiological idiosyncrasies of non-model bacteria (Müller et al., 2023).”

New supporting Data 4:

Supporting Data 4. Results of validation screen.

strain.id	strain	compound	concentration	norm_auc	n_sign_rep	p_adj	is_hit_20uN	isHit_anyConc	MainScreen_H
NT5002	Bacteroides	ABAMECTIN	2.5	0.975484072	0	0.90848	FALSE	FALSE	FALSE
NT5002	Bacteroides	ABAMECTIN	5	0.996447533	0	0.951532	FALSE	FALSE	FALSE
NT5002	Bacteroides	ABAMECTIN	10	0.9675203	0	0.907673	FALSE	FALSE	FALSE
NT5002	Bacteroides	ABAMECTIN	20	0.995822923	0	0.943438	FALSE	FALSE	FALSE
NT5002	Bacteroides	ABAMECTIN	40	0.974469082	0	0.709578	FALSE	FALSE	FALSE

3.5. Com20 dose and in vivo

The concentration of the two xenobiotics used to measure effects on the synthetic community was not described – but assuming it is 20 μM , then again it is the same issue as in the point above, this concentration needs to be physiologically relevant, and it is somewhat of an overinterpretation to assume that effects in monoculture are translatable to community effects based on the results of only two xenobiotics tested. The ultimate measure here is of course the exposure of a human-relevant dose in a mammalian setting (mice) and the quantification of microbiome changes in this setting. Such test would take into account real exposure in a mammalian gut, and the concentration/time needed to observe any significant changes.

We agree that such experiments would be interesting. However, these experiments would be more informative of mouse physiology and the toxico-kinetic/dynamics of these compounds, rather than revealing something fundamentally novel about xenobiotic-microbe interactions. The point of our study is to demonstrate the potential impact of these compounds on gut bacteria, to map the xenobiotic-bacteria interaction landscape, and to gain genetic insights into these networks. Further considering the arguments laid out above (“Summary of key changes and aim of the study”), we consider the proposed experiments out of scope.

Regarding Com20 dose, while we illustrate in Figure 2F the changes observed at 20 μM , **Supporting Data 5** contains data of different xenobiotic doses for the community experiment (ranging 0.16-20 μM) where changes in the community composition are seen at lower doses. We reproduce below an example of the data.

3.6. Com20 strains

Were the strains used in the Com20 the same exact strains as the ones in the monoculture screen?

Yes, for those species included in the Com20 and the mono-culture screen, the same strains were used. We apologise that this was not apparent from the strain identifiers provided in the supplement (we provided DSMZ identifiers for the main screen, but internal identifiers for the Com20 screen) and have rectified this by including all strain identifiers in the relevant tables.

3.7. Mechanism BPAF TBBPA

“In the case of therapeutic drugs, the emergent community-scale effects are due to bacterial metabolism of drugs and the impact of drugs on bacterial metabolism. The results from the BPAF and TBBPA suggest that similar mechanisms are applicable in the case of chemical contaminants. “ This is an unsupported statement without any mechanistic insights into metabolism of chemical contaminants by members of the microbiome, or effect of these chemicals on microbial metabolism.

We appreciate this comment and have followed this up with LC-MS² measurements to determine the accumulation and/or degradation of BPAF by the Com20 community (new **Fig 2H**, reproduced below). We find that BPAF is not biotransformed, but it is accumulated to substantial levels (64% reduction in free BPAF). This is congruent with one of our recent studies, demonstrating BPAF accumulation in single-species cultures (Lindell et al. 2025), which we refer to, as well as to the general cross-protective effect of bioaccumulation across a range of pharmaceutical drugs (Garcia-Santamarina et al. 2024).

In main text:

“Community-level effects such as cross-protection have been described in gut bacterial communities exposed to therapeutic drugs (Garcia-Santamarina et al., 2024; Klünemann et al., 2021). We hypothesised that cross-protection against BPAF in Com20 is due to bioaccumulation (Lindell et al., 2025) and used liquid chromatography – tandem mass spectrometry (LC-MS) to measure BPAF concentrations in whole culture, supernatant and pellets of Com20 cultures exposed to 20 µM BPAF. We detected a 64% reduction in BPAF in supernatant compared to whole culture and recovered BPAF in the pellet (**Figure 2H**), indicating that the majority of BPAF in Com20 is sequestered within cells, thereby reducing bioavailability to other community members.“

New Figure 2H:

3.8. Modelling

The authors present the modeling results as a support for the relevance of their findings, yet even in the modeling experiment they appear to simulate an unrealistic dosage scheme for a pollutant. I am not even sure how they calculated the concentrations to be used in their repeated exposures. The description appears to indicate that they selected a regimen that would yield an inhibitory dose, which is circular in nature. Isn't the modeling supposed to test whether a realistic dose can lead to gut exposure levels that result in inhibition?

We acknowledge that the toxicokinetic modelling was not explained in sufficient detail in the context of main text. The goal of the analysis is to compare our results with those toxicity values captured in our screen to toxicology values in mice. The analysis shows that our screen captures toxic effects with higher sensitivity than mice models for comparable doses. We agree this analysis is out of the scope of the main message of the manuscript. We have removed this section from main text and moved the toxicokinetic modelling instead to Supporting Information (**Supplementary Figure 7**).

We include the output of this analysis as a reference point in our discussion:

We include in discussion:

“The prevalent off-purpose activities of chemical pollutants revealed by our screen, together with those of human targeted drugs (Guillen et al., 2024; Maier et al., 2018), challenge the categorical classification used to label antibiotics, human-targeted drugs, industrial chemicals, and pesticides. The antibacterial activity of many xenobiotics is

currently not accounted in assessing the potential adverse impact of these chemicals. The current toxicity studies are typically set up in a way that are incompatible with capturing effects on gut microbiota (**Supplementary Figure 7**). Towards addressing this gap, toxicology assessment on single gut bacteria and synthetic communities could be added to toxicology panels and further expanded to include more complex models like gut-on-chip (Moossavi et al., 2022). Nevertheless, the vast numbers of chemicals in use, and the fast pace at which new molecules are brought to market will make it difficult to implement meaningful experimental assessment for all compounds. Towards tackling this problem, our study demonstrates feasibility of computational toxicity predictions that can be used at scale and for safe-by-design product development. This proof-of-concept was possible due to the here-expanded chemical space underscoring the need for standardised chemical-bacteria screens. “

3.9. Abundance correlation

“We noted a positive correlation of number of compounds affecting a species with its abundance in the human gut” – what is the rationale to perform this correlation? It is heavily confounded by the sampling of the molecules tested and the sampling of the microbes tested. It is also irrelevant: abundant microbes do not have anything specific to them to make them more or less susceptible to the random set of compounds tested in this study. Same with the correlation between microbes affected by pharmaceuticals and non-pharmaceuticals. This statement is misleading: “The correlation in the case of human-targeted drugs indicates that bacteria use a common and non-specific mechanism to tackle non-antibiotic drugs and chemical pollutants.” – bacteria do not evolve to “tackle” specific molecules they have never been exposed to! And certainly do not use common and non-specific mechanisms to interact with hundreds of molecules. I encourage the authors to refrain from making such broad statements that are not based on real biological rationales nor on newly collected data.

We have now explained this analysis better, and improved the figures, however we disagree with the reviewer on the motivation and interpretation of this analysis. The motivation for the correlation analysis was to compare our dataset with sensitivity patterns in (Maier et al., 2018) where a similar correlation between relative abundance of a species in the human gut and sensitivity to human-targeted drug is observed. We note these findings here for an orthogonal set of compounds. Both the number of species and number of compounds tested are large enough to allow drawing such a general, albeit correlative, inference. In the new manuscript version this adds further context to the leveraging both Maier dataset and our dataset for machine learning predictions.

Further, our extensive genetic analyses and observation on cross-resistance to antibiotics mechanistically support the idea.

We have now updated these figures colouring the data points by phylum, which is relevant to contextualise Bacteroidetes as a pollutant-sensitive abundant taxa, which is the reason we focus on this taxon on the genetic screens (following in member species *P. merdae* and *B. thetaiotaomicron*).

New Figure 2A-C:

We have moved specific examples of mechanisms to Discussion, to present it in the context of genetic screen result. We also now highlight the existing literature around this to highlight that our data strengthens the previous observations and broadens their scope considerably to relevant chemicals.

In main text commenting on abundance correlation we conclude instead:

“The correlation between chemicals in our screen and human-targeted drugs supports the previous observations that gut bacteria interact with a variety of chemicals via common and non-specific mechanisms (Guillen et al., 2024; Maier et al., 2018; Nagata et al., 2022).”

Further, we want to clarify that we do not claim that our strains have evolved specific resistance to any of the compounds tested. There are well-known mechanisms that impart broad, non-specific resistance to xenobiotics: In Maier et al 2018, we identify the *E. coli* tolC efflux pump as a major player affecting pan-drug resistance. Similarly, (Sakenova et al., 2024) describe widespread cross-resistance for antibiotics.

Noto et al has shown in *E. coli* with chemical-genetics that even for non-antibiotics with different mode of action similar mechanisms of resistance can be observed which involve transport systems and membrane permeability(Guillen et al., 2024). Finally, a cohort study (Nagata et al., 2022) associated exposure to non-antibiotic drugs to antimicrobial resistance.

3.10. Transposon screen doses

There are several issues with the genetic screen to identify resistance/sensitivity determinants. First, some unrealistic concentrations were used, such as 50 uM and 500 uM (even 20 uM is high). Second, the effects on efflux, membrane integrity, transport are all expected with chemical stress, and appear to be non-specific in most cases. The only unexpected result is the fact that some metabolic pathway mutants appear to mediate a growth advantage under treatment conditions. It is not clear if this result is simply a reflection of these mutants being faster in growth than WT, and therefore appear to be enriched when overall growth inhibition is at play, or if there is anything specific to this interaction. The typical follow-up experiment to confirm these findings is to isolate (or remake) these specific mutants and compare their MIC or growth curves to the WT in the presence of different concentrations of the xenobiotic. Without such confirmation, the results are hard to interpret.

We appreciate the reviewer's comment, which have helped us improve our genetic screen sections. We address in the following sections the observations regarding isolate mutant follow ups, concentrations used, and relevance of efflux hits, and interpretation of TnBarSeq growth effects.

Isolate mutants:

We have conducted extensive follow ups on isolate mutants, including growth effects at different concentrations and MIC changes.

To validate and further characterise pooled genetic screen hits, we have isolated mutant strains across different genomic locations corresponding to the top positive selection hits under TBBPA, closantel (*acrR* mutants), and PFNA (inositol lipid biosynthesis mutant NQ542_07540).

In the case of *acrR*, in the main text:

"To further characterise *acrR* mutant phenotype, we isolated transposon mutant strains with insertions at seven distinct positions within *acrR* coding and promoter regions after growth on closantel or TBBPA. All *Tn::acrR* isolates showed resistance to both chemicals regardless of the compound they were isolated on (**Figure 4C, Supplementary Figure 5C**). When evaluating two *Tn::acrR* strains, we observed an 8-fold increase in MIC for

closantel compared to the wild type strain (WT). Consistent with previous reports linking loss of *acrR* function to ciprofloxacin resistance in diverse bacteria (Zlamal et al., 2021), we observed up to 16-fold increase on ciprofloxacin MIC for *P. merdae* *Tn::acrR* mutants compared to wild type strain (WT) (**Figure 4D, Supplementary Data 8**). Therefore, resistance to closantel and TBBPA can lead to cross-resistance to antibiotics such as ciprofloxacin.”

In main Figure 4:

In the case of inositol lipid biosynthetic cluster, we have validated both resistance to PFNA and hypersensitivity to TBBPA at various concentrations. We also further investigated how these strains interact with PFNA using mass spectrometry.

In the main text:

“To validate the phenotype, we isolated after enriching in PFNA 500 μ M *P. merdae* mutants at two locations of the putative phosphatidylinositol phosphate phosphatase (PIPPh) gene NQ542_07540. These strains showed increased resistance to PFNA and hypersensitivity to TBBPA at a range of concentrations, in agreement with pooled fitness (**Figure 4H, Supplementary Figure 5D,E**). To evaluate if the increased resistance to PFNA was due differences in compound uptake, we tested PFNA bioaccumulation, but no growth-independent differences were seen (**Supplementary Figure 5F**), indicating accumulation-independent function of inositol lipid metabolism in PFNA resistance.”

Supporting Figure 5 includes further characterisation of *P. merdae* isolate strains:

Supplementary Figure 5. Isolate transposon mutant characterisation **A.** Diagnostic PCR genotyping of *acrR* locus transposon mutants (*Tn::NQ542_01170*). Transposon insertions are validated as a larger amplicon size compared to the wild type strain (WT). Mutant insertion location information in Supplementary Data 7. **B.** Genotyping of transposon mutants located at a putative phosphatidylinositol phosphate phosphatase (*Tn::NQ542_07540*) using primers targeting the junction between transposon and insertion locus confirms transposon location and orientation predicted by barcode sequencing. **C.** Growth curves of isolate transposon mutants under xenobiotics. Curves represent strains with independent insertions on *acrR* coding region and promoter regions (purple n=7), *Tn::NQ542_07540* from inositol lipid gene cluster (light blue, n=2) and WT strains for comparison (black, n=6). **D.** Growth curves at 12.5 µM TBBPA and DMSO control show *Tn::NQ542_07540* hypersensitivity to TBBPA. Curves represent two well replicates of two independent insertion strains, and *P. merdae* WT control. **E.** Fitness of isolated *P. merdae* mutants (strain 9 and 10 corresponding to independent insertions at *NQ542_07540*) and WT control across a range of concentrations of PFNA (0–100 µM). nrAUC is normalised to the median AUC of the WT value per concentration, and the median of DMSO control (0 µM) per genotype. Replicates are shown as points and P-adjusted values were obtained comparing mutants to WT at each concentration with Welch's T-test corrected by multiplicity with BH. **F.** PFNA bioaccumulation in isolate *P. merdae* strains of different genotypes grown on a range of PFNA concentrations, as analysed by liquid chromatography coupled to mass spectrometry (LC-MS), see Methods. End OD before compound extraction is indicated in coloured scale. Replicate values are shown as points. P-values were calculated compared to WT using Welch's T-test corrected by multiplicity with BH.

Supplementary Data 7 describes the genomic position and orientation after sequencing of the isolate mutant barcodes, as well as each strain genotyping results:

Supplementary Data 7. *P. merdae* transposon mutant isolated by enriching in selective conditions, with corresponded transposon (Tn) barcode identified by Sanger sequencing and their genomic integration position, transposon orientation and gene orientation. The area under the growth curve perturbation.

Colony_ID	Isolation_Condition	LocI	Gene_ID	Gene_Product	Tn_Barcode	Genome_accession	Tn_genomic_position	Tn_Orientation	Gene_Orientation	Tn_Relative_Position
1	TBBPA_20µM	TetR/AcrR family transcriptional repressor	NQ542_01170	TetR/AcrR family transcr	ACCTCTCTCTGCTGCTCTACTC	CP102286.1	317694	-	-	0.91
2	TBBPA_20µM	TetR/AcrR family transcriptional repressor	NQ542_01170	TetR/AcrR family transcr	CCTGCTCTCATCATGGAGTCGGG	CP102286.1	317840	-	-	0.56
3	TBBPA_20µM	TetR/AcrR family transcriptional repressor	NQ542_01170	TetR/AcrR family transcr	CCTGCTCTCATCATGGAGTCGGG	CP102286.1	317840	-	-	0.56
4	TBBPA_20µM	TetR/AcrR family transcriptional repressor	NQ542_01170	TetR/AcrR family transcr	TTACTCAGTAGGAACATCCACAT	CP102286.1	317979	-	-	0.23
5	Closantel_2µM	TetR/AcrR family transcriptional repressor	NQ542_01170	TetR/AcrR family transcr	ACGTATGTTTGCCTGCTGACG	CP102286.1	318013	-	-	0.15
6	Closantel_2µM	TetR/AcrR family transcriptional repressor	NQ542_01170 promoter region		TTTGAGCGGGGATCATTACGGT	CP102286.1	318097	-	-	0.95
7	Closantel_2µM	TetR/AcrR family transcriptional repressor	NQ542_01170 promoter region		TGTCGTGCGGTCTAGCGCAATG	CP102286.1	318230	-	-	0.63
8	TBBPA_20µM	TetR/AcrR family transcriptional repressor	NQ542_01170 promoter region		ACATAATTTAGGGTTGGTTGAGG	CP102286.1	318344	-	-	0.65
9	PFNA_500µM	Inositol Lipid Operon	NQ542_07540	phosphatidylglyceroph	TGGATTGCACCGTTTCTATCCAGG	CP102286.1	1831690	+	+	0.05
10	PFNA_500µM	Inositol Lipid Operon	NQ542_07540	phosphatidylglyceroph	AACGGGCTTCGGGTGCAGTGTGC	CP102286.1	1831806	+	+	0.30

Efflux related chemical-genetic hits:

We analysed efflux related gene hits in *P. merdae* detail because perturbation of efflux impacts tolerance to both antibiotics and xenobiotics, as observed in genetic screens in *E. coli* (Binsfeld et al., 2025; Guillen et al., 2024). We have now included further text and experiments to clarify the relevance of characterising the transporter “genetic fingerprint” of different pollutants, and how this response can be conserved across bacterial species.

First, we include a new analysis of efflux repression mutants (*Tn::acr*) where resistance to pollutant also selected for antibiotic cross-resistance:

“Consistent with previous reports linking loss of *acrR* function to ciprofloxacin resistance in diverse bacteria (Zlamal et al., 2021), we observed up to 16-fold increase on ciprofloxacin MIC for *P. merdae* *Tn::acrR* mutants compared to wild type strain (WT) (Figure 4D, Supplementary Data 8). Therefore, resistance to closantel and TBBPA can lead to cross-resistance to antibiotics such as ciprofloxacin”.

D) Dose-response curves under closantel and ciprofloxacin of *acrR* mutants and control strains. Data points represent optical density values at 24 hours of culture from technical duplicates across three independent experiments. The two-fold dilution series highest tested concentration of ciprofloxacin 603.6 µM is equivalent to 200 µg/mL.

Supplementary Data 8 contains the replicate results of MIC tests:

Genotype	Compound	Concentration	Unit	Dilution_factor	OD600_blanked_24hrs	OD_normalised_to_DMSO	MIC_dilution_factor	foldchange_MIC_to_WT	Independent_Batch_Replicate	Well
P. merdae WT	Ciprofloxacin	200	µg/ml	1	0.0235	0.045586809	0.03125	1	B	A1
P. merdae WT	Ciprofloxacin	200	µg/ml	1	0.0115	0.022308438	0.03125	1	B	A2
P. merdae WT	Ciprofloxacin	200	µg/ml	1	0.005	0.009242144	0.03125	1	A	A1
P. merdae WT	Ciprofloxacin	200	µg/ml	1	0.007	0.012939002	0.03125	1	A	A2

We have now made clearer in the text which genes have a general response to xenobiotics in *P. merdae*, which were uncharacterised as it is a non-model bacterium.

“The differential enrichment of *acrR* mutants suggests a degree of specificity in the efflux of chemicals and/or that efflux alone is insufficient in tackling the other

chemicals. However, some transporter mutants showed broad pollutant sensitivity, indicating common pollutant tolerance mechanisms in *P. merdae*. For example, the FadL family of hydrophobic compound transporters (NQ542_07320 and NQ542_08455) showed prevalent negative fitness hits across 7 xenobiotics, while RND multidrug efflux locus NQ542_06240-50 was significant hit across 5 chemicals (**Supplementary Figure 5C**)."

Importantly, we have performed additional experiments to explore response in a distant Bacteroidales species (*B. thetaiomicron*). We screened arrayed mutants of 185 putative transporters, observing compound dependent responses, which are linked to previous antibiotic-dependent responses. We also observed that the top hits in *B. thetaiomicron* showed a conserved response in *P. merdae*.

We have added in main text:

"To assess if the involvement of transporters in xenobiotic response is broadly conserved in the order Bacteroidales, we evaluated mutants of *B. thetaiotaomicron*, a member of a distant family to *P. merdae* (Zhang et al., 2024). We tested closantel, TBBPA, PFNA and BPAF on a curated subset of 185 transporter mutants from the *B. thetaiotaomicron* arrayed loss-of-function library (Arjes et al., 2022). Growth was monitored over 24 h, and we evaluated the response of each mutant by calculating the normalised relative AUC (nrAUC), comparing xenobiotic treatment to DMSO control (**Supplementary Figure 6A, Supplementary Data 9**). The strongest growth reduction for closantel was observed at 2 μ M and 10 μ M for the mutant of an RND-efflux pump BT2687 (**Figure 4F, Supplementary Figure 6B**). This gene is part of an *acrR-RND-toIC* locus homologous to *P. merdae* *Tn::acrR* closantel hit, suggesting a conserved efflux response (**Supplementary Figure 6C**). Additional mutants with conditional growth reduction on closantel include the co-localised multidrug ABC transporter genes BT0562 and BT0563. Mutants of BT2687 and BT0562-63 have been previously linked to increased susceptibility to antibiotics and biocides in *B. thetaiotaomicron* (Liu et al., 2021). Exposure to PFNA 250 μ M and 100 μ M inhibited the mutants BT3338 and BT3337, which encode an RND-efflux pump needed for biofilm formation (Lopes et al., 2024), and tolerance to fusidic acid, chlorpromazine, and thioridazine (Liu et al., 2021) (**Figure 4F**). BT3338-BT3337 top match by homology and synteny is the RND loci in *P. merdae* NQ542_09525-35, which was a specific negative fitness hit for PFNA in the pooled screen (**Supplementary Figure 6D**). As PFNA can be bioaccumulated by several Bacteroidales (Lindell et al., 2025), we assessed if the mutant phenotype of BT3338-BT3337 could be caused by increased compound uptake. However, when we tested for PFNA bioaccumulation across concentrations, no differences were observed (**Supplementary Figure 6E,F, Supplementary Data 11**), indicating other mechanisms involving membrane interactions. Overall, the shared responses between *P. merdae* and *B. thetaiotaomicron* support conserved mechanisms of pollutant and antibiotic tolerance across Bacteroidales."

F) Effect of xenobiotics on growth of an arrayed transporter mutants of *B. thetaiotaomicon*. Axis show normalised relative area under the curve per mutant (nrAUC= per mutant average rAUC xenobiotic / average rAUC DMSO), as described in methods. Hits are defined by an absolute effect in growth of >20% and p-value<0.05.

Related new Supplementary Figure, including QC, follow up, and hit loci conservation examples :

Supplementary Figure 6. Isolate transposon mutant characterisation **A.** Diagnostic PCR genotyping of *acrR* locus transposon mutants (*Tn::NQ542_01170*). Transposon insertions are validated as a larger amplicon size compared to the wild type strain (WT). Mutant insertion location information in Supplementary Data 7. **B.** Genotyping of transposon mutants located at a putative phosphatidylinositol phosphate phosphatase (*Tn::NQ542_07540*) using primers targeting the junction between transposon and insertion locus confirms transposon location and orientation predicted by barcode sequencing. **C.** Growth curves of isolate transposon mutants under xenobiotics. Curves represent strains with independent insertions on *acrR* coding region and promoter regions (purple n=7), *Tn::NQ542_07540* from inositol lipid gene cluster (light blue, n=2) and WT strains for comparison (black, n=6). **D.** Growth curves at 12.5 µM TBBPA and DMSO control show *Tn::NQ542_07540* hypersensitivity to TBBPA. Curves represent two well replicates of two independent insertion strains, and *P. merdae* WT control. **E.** Fitness of isolated *P. merdae* mutants (strain 9 and 10 corresponding to independent insertions at *NQ542_07540*) and WT control across a range of concentrations of PFNA (0–100 µM). nrAUC is normalised to the median AUC of the WT value per concentration, and the median of DMSO control (0 µM) per genotype. Replicates are shown as points and P-adjusted values were obtained comparing mutants to WT at each concentration with Welch's T-test corrected by multiplicity with BH. **F.** PFNA bioaccumulation in isolate *P. merdae* strains of different genotypes grown on a range of PFNA

concentrations, as analysed by liquid chromatography coupled to mass spectrometry (LC-MS), see Methods. End OD before compound extraction is indicated in coloured scale. Replicate values are shown as points. Adjusted P-values were calculated compared to WT using Welch's T-test corrected by multiplicity with BH.

Related new Supplementary Data 9:

Supplementary Data 9. Growth analysis under xenobiotics of arrayed transporter mutants of the B. thetaiomicron mutant library (derived from Arjes et al 2022). Gene annotation by TransportAPP database, eggNOG and fitnessbrowser is provided. Growth Screen Results derived from GrowthCurver area under the curve. Normalised rAUC (nrAUC) for each xenobiotic condition, the average rAUC for mutant gene under xenobiotic was divided by the average rAUC in the corresponding DMSO control. **is_hit**: Hits are defined by $p\text{-value} < 0.05$ and normalised_rAUC corresponding to $\geq 20\%$ increase or decrease.

Gene_name	Preferred_name_by_eggNOG	desc_fitnessbrowser	locusid_fitnessbrowser	Description_by_eggNOG	Substrate_TransportAPP	Family_name_TransportAPP	Transporter_Class_TransportAPP	TC_number_TransportAPP	Condition	Control	average_rAUC_conditi
BT0009	pst5	phosphate-binding perit	349537	PBP superfamily domain	phosphate	The ATP-binding Cassette (ABC)	ATP-Dependent	3.A.1	Closantel 2 uM	DMSO 0.4%	0.99182
BT0009	pst5	phosphate-binding perit	349537	PBP superfamily domain	phosphate	The ATP-binding Cassette (ABC)	ATP-Dependent	3.A.1	TBBPA 20 uM	DMSO 0.4%	0.96403
BT0009	pst5	phosphate-binding perit	349537	PBP superfamily domain	phosphate	The ATP-binding Cassette (ABC)	ATP-Dependent	3.A.1	PFNA 250 uM	DMSO 1%	1.05520
BT0009	pst5	phosphate-binding perit	349537	PBP superfamily domain	phosphate	The ATP-binding Cassette (ABC)	ATP-Dependent	3.A.1	BPAF 20 uM	DMSO 0.4%	0.99720
BT0009	pst5	phosphate-binding perit	349537	PBP superfamily domain	phosphate	The ATP-binding Cassette (ABC)	ATP-Dependent	3.A.1	Closantel 10uM	DMSO 1%	1.33650
BT0040	-	hypothetical protein [N	349568	Psort location Cytoplasmic	polysaccharide export	The Multidrug/Oligosaccharidyl	Secondary Transporter	2.A.66	PFNA 250 uM	DMSO 1%	0.95289
BT0040	-	hypothetical protein [N	349568	Psort location Cytoplasmic	polysaccharide export	The Multidrug/Oligosaccharidyl	Secondary Transporter	2.A.66	Closantel 2 uM	DMSO 0.4%	1.01231

Concentrations used:

Regarding the concentrations used for the genetic screen, we have further clarified in the main text:

“TBBPA, BPAF, imazalil sulphate, fluazinam, closantel and emamectin benzoate were tested at concentrations $\leq 20 \mu\text{M}$, while glyphosate, propiconazole, PFOA and PFNA were tested at $\geq 20 \mu\text{M}$ to ensure selective pressure on growth fitness (**Methods**).”

We also clarified in main text that at conditions where there is no growth constrain, gene hits are not resolved, thus the need of fitness screens to be performed at inhibitory concentrations:

“In contrast, PFNA 20 μM , PFOA 20 μM and glyphosate 50 μM did not show significant hits, consistent with the lack of effect in growth at those concentrations (**Supplementary Figure 4B**)”

In isolate mutants follow growth experiments we have tested growth phenotypes at multiple concentrations e.g., *acrR* mutants were evaluated at concentrations as low as 0.1 μM .

Interpretation of TnBarSeq in relation to global growth differences:

We appreciate the reviewer's concern if gene hits could arise from global changes in growth under pollutant (e.g., differences in carrying capacity) rather than specific gene-chemical interactions. We have now carefully accounted for this possibility by performing a post-hoc analysis to detect any major effect of growth bias in TnBarSeq using EEL correction algorithm by (Kim et al., 2024). This method compares the log2fold values from end point perturbation vs end point control, and the log2fold values between time zero (T0) sample to end point control condition, allowing identification of gene hits that could be driven by differential growth. Following this analysis, we observe no major fitness effect that could be attributed to differences in cell doublings alone. This supports the conclusion that our gene hits reflect conditional fitness effects under chemical perturbations.

We have now incorporated in Supporting Data 6 the results of this analysis, and the results of comparing time zero (T0) sample to end point control condition (DMSO or media).

We have now included in the main text:

“We further verified that gene hits were not caused by global differences in cell doublings across conditions (**Supplementary Figure 4A**)(Kim et al., 2024).”

And in Supplementary Figure 5:

Overall, these follow up analysis support the interpretation of TnBarSeq, and highlight the relevance of identified gene hits showing antibiotic cross-resistance and evolutionary conserved gene hits.

3.11. Transposon screen validation

A better experiment than the transposon one is to evolve resistant mutants, by treating the WT with sub MIC concentrations in subsequent rounds and isolating strains that grow undeterred in the presence of such concentrations. This is more of a reflection to a real life setting in the human microbiome. Finally, the authors claim that exposure to pollutants can lead to mutants that lost the functionality of host-relevant metabolic pathways. It is relatively simple to check in bacterial isolates or metagenomic data from the human microbiome if these mutants do indeed exist, where human populations are actually exposed to the studied chemicals for a long time and under the biologically relevant concentrations. Absent this data or similar one presented from new experiments, the current results are indeed confounded by the unrealistically high concentrations and exposures used.

We thank the reviewer for the observation. We agree that experimental evolution is a valuable complementary approach, however we consider it is out of the scope for this

study. We choose transposon library screens as they provide a genome-wide quantitative understanding of gene fitness under xenobiotics encompassing both positive and negative effects. In total, the pooled genetic screen in *P. merdae* mapped >27k gene-chemical interactions in non-essential genes, of which 1979 were defined as gene hits. The resulting ‘genetic fingerprint’ associated to compounds is representative of different modes of action (Noto Guillen 2024). We have now included additional 920 gene-chemical interactions tested in an arrayed subset of *B. theta* transporter mutants, of which 54 were defined significant hits.

Furthermore, recent work has shown that chemical-genetic screens often capture trajectory of evolutionary experiments (Couce et al., 2024; Guillen et al., 2024; Sakenova et al., 2024). Effectively, we want to share with the reviewer that our chemical-genetic screens in *P. merdae* and *B. theta* transporter mutants for PFNA show agreement for the top selection hit under adaptive laboratory evolution for PFNA in *B. uniformis* in previous work (Lindell et al. 2025). This highlights how chemical-genetic screens can lead to evolutionary relevant gene candidates.

Response Figure 5: Conservation between negative fitness hits in PFNA by TnBarSeq in *P. merdae* from this work, with the main selection hit after laboratory evolution in PFNA (missense mutation) in Lindell et al 2025. Similarly, the main hit for PFNA in the arrayed *B. theta* transporter library (this work) aligns with another *P. merdae* locus with strong negative fitness under PFNA. This shows how these experiments are complementary and can arrive to conserved gene candidates. A similar version of the figure including only the results of this work is found in Supplementary Figure 6D.

We appreciate the reviewer's point about the value of identifying candidate genes and mechanistic hints to guide metagenomic analysis. By leveraging the findings of our screen, we could in principle follow up candidate genes in human microbiome cohort data.

However, the lack of linked metadata on specific xenobiotic exposure for the cohort metagenomic datasets limits the possibility of interpretation of such data.

The same way Maier et al (2018) study of non-antibiotic medication effect in gut bacteria inspired cohort studies to incorporate wider medication metadata, we hope that our study encourages future cohort research to incorporate metadata on xenobiotic exposure, or, better, include direct measurements from the same samples.

12. Commensal definition

This may appear trivial but it is an extremely serious issue, especially in a high-profile microbiology journal that should abide to the highest standards: I encourage the authors to study the real definition of “commensals” and “commensalism” before using this terminology. Commensalism, mutualism, parasitism, symbiosis, etc., are all well-bound terms with accurate ecological definitions, and should not be casually used. I don’t believe that we have experimental data to support calling any of the 22 organisms tested in this study “commensals” – obviously not *C. difficile*, but actually neither the remaining 21 strains.

We agree that adhering to ecological definitions is important. We have now replaced most mentions of the word “commensal” with “gut bacteria”, especially for the term “anti-commensal” is now replaced with “anti-gut-bacteria”.

We only include the term commensal to describe the selected bacteria panel, as described in Maier et al 2018. Of note, a) recent studies like have shown that gut bacteria do use the same food as the host (Zeng et al., 2022) b) while *C. difficile* is an opportunistic pathogen, it is quite prevalent in infants wherein it does not cause any disease/symptoms (Ferretti et al., 2023).

References

- Arjes, H. A., Sun, J., Liu, H., Nguyen, T. H., Culver, R. N., Celis, A. I., Walton, S. J., Vasquez, K. S., Yu, F. B., Xue, K. S., Newton, D., Zermeno, R., Weglarz, M., Deutschbauer, A., Huang, K. C., & Shiver, A. L. (2022). Construction and characterization of a genome-scale ordered mutant collection of *Bacteroides thetaiotaomicron*. *BMC Biology*, *20*(1), 1–17.
<https://doi.org/10.1186/S12915-022-01481-2/>
- Binsfeld, C., Olayo-Alarcon, R., Jiménez, L. P., Wartel, M., Stadler, M., Mateus, A., Müller, C., & Brochado, A. R. (2025). Systematic screen uncovers regulator contributions to chemical cues in *Escherichia coli*. *PLOS Biology*, *23*(7), e3003260.
<https://doi.org/10.1371/JOURNAL.PBIO.3003260>

- Braun, G., Herberth, G., Krauss, M., König, M., Wojtysiak, N., Zenclussen, A. C., & Escher, B. I. (2024). Neurotoxic mixture effects of chemicals extracted from blood of pregnant women. *Science*, *386*(6719), 301–309. <https://doi.org/10.1126/SCIENCE.ADQ0336>
- Couce, A., Limdi, A., Magnan, M., Owen, S. V., Herren, C. M., Lenski, R. E., Tenailon, O., & Baym, M. (2024). Changing fitness effects of mutations through long-Term bacterial evolution. *Science*, *383*(6681). <https://doi.org/10.1126/SCIENCE.ADD1417/>
- Ferretti, P., Wirbel, J., Maistrenko, O. M., Rossum, T. Van, Alves, R., Fullam, A., Akanni, W., Schudoma, C., Schwarz, A., Thielemann, R., Thomas, L., Kandels, S., Hercog, R., Telzerow, A., Letunic, I., Kuhn, M., Zeller, G., Schmidt, T. S., & Bork, P. (2023). *C. difficile* may be overdiagnosed in adults and is a prevalent commensal in infants. *ELife*, *12*. <https://doi.org/10.7554/ELIFE.90111.1>
- Garcia-Santamarina, S., Kuhn, M., Devendran, S., Maier, L., Driessen, M., Mateus, A., Mastroilli, E., Brochado, A. R., Savitski, M. M., Patil, K. R., Zimmermann, M., Bork, P., & Typas, A. (2024). Emergence of community behaviors in the gut microbiota upon drug treatment. *Cell*, *187*(22), 6346–6357.e20. <https://doi.org/10.1016/j.cell.2024.08.037>
- Guillen, M. N., Li, C., Rosener, B., & Mitchell, A. (2024). Antibacterial activity of nonantibiotics is orthogonal to standard antibiotics. *Science*, *384*(6691), 93–100. <https://doi.org/10.1126/SCIENCE.ADK7368>
- Hens, B., Brouwers, J., Corsetti, M., & Augustijns, P. (2016). Supersaturation and Precipitation of Posaconazole Upon Entry in the Upper Small Intestine in Humans. *Journal of Pharmaceutical Sciences*, *105*(9), 2677–2684. <https://doi.org/10.1002/JPS.24690>
- Iezzi, S., Lifschitz, A., Sallovitz, J., Nejamkin, P., Lloberas, M., Manazza, J., Lanusse, C., & Imperiale, F. (2014). Closantel plasma and milk disposition in dairy goats: assessment of drug residues in cheese and ricotta. *Journal of Veterinary Pharmacology and Therapeutics*, *37*(6), 589–594. <https://doi.org/10.1111/JVP.12135>
- Kim, L. M., Todor, H., & Gross, C. A. (2024). Correction of a widespread bias in pooled chemical genomics screens improves their interpretability. *Molecular Systems Biology*, *20*(11), 1173–1186. <https://doi.org/10.1038/S44320-024-00069-Y/>
- Klünemann, M., Andrejev, S., Blasche, S., Mateus, A., Phapale, P., Devendran, S., Vappiani, J., Simon, B., Scott, T. A., Kafkia, E., Konstantinidis, D., Zirngibl, K., Mastroilli, E., Banzhaf, M., Mackmull, M. T., Hövelmann, F., Nesme, L., Brochado, A. R., Maier, L., ... Patil, K. R. (2021). Bioaccumulation of therapeutic drugs by human gut bacteria. *Nature*, *597*(7877), 533–538. <https://doi.org/10.1038/s41586-021-03891-8>
- Lindell, A. E., Griebshammer, A., Michaelis, L., Papagiannidis, D., Ochner, H., Kamrad, S., Guan, R., Blasche, S., Ventimiglia, L. N., Ramachandran, B., Ozgur, H., Zelezniak, A., Beristain-Covarrubias, N., Yam-Puc, J. C., Roux, I., Barron, L. P., Richardson, A. K., Martin, M. G., Benes, V., ... Patil, K. R. (2025). Human gut bacteria bioaccumulate per- and polyfluoroalkyl substances. *Nature Microbiology* *2025* *10*:7, *10*(7), 1630–1647. <https://doi.org/10.1038/s41564-025-02032-5>
- Liu, H., Shiver, A. L., Price, M. N., Carlson, H. K., Trotter, V. V., Chen, Y., Escalante, V., Ray, J., Hern, K. E., Petzold, C. J., Turnbaugh, P. J., Huang, K. C., Arkin, A. P., & Deutschbauer, A. M. (2021). Functional genetics of human gut commensal *Bacteroides thetaiotaomicron* reveals metabolic requirements for growth across environments. *Cell Rep*, *34*(9). <https://doi.org/10.1016/j.celrep.2021.108789>

- Lopes, A. A., Vendrell-Fernández, S., Deschamps, J., Georgeault, S., Cokelaer, T., Briandet, R., & Ghigo, J. M. (2024). Bile-induced biofilm formation in *Bacteroides thetaiotaomicron* requires magnesium efflux by an RND pump. *MBio*, *15*(5).
<https://doi.org/10.1128/MBIO.03488-23/>
- Maier, L., Pruteanu, M., Kuhn, M., Zeller, G., Telzerow, A., Anderson, E. E., Brochado, A. R., Fernandez, K. C., Dose, H., Mori, H., Patil, K. R., Bork, P., & Typas, A. (2018). Extensive impact of non-antibiotic drugs on human gut bacteria. *Nature*, *555*(7698), 623–628.
<https://doi.org/10.1038/nature25979>
- Moossavi, S., Arrieta, M. C., Sanati-Nezhad, A., & Bishehsari, F. (2022). Gut-on-chip for ecological and causal human gut microbiome research. *Trends in Microbiology*, *30*(8), 710–721. <https://doi.org/10.1016/J.TIM.2022.01.014>
- Müller, P., de la Cuesta-Zuluaga, J., Kuhn, M., Baghai Arassi, M., Treis, T., Blasche, S., Zimmermann, M., Bork, P., Patil, K. R., Typas, A., Garcia-Santamarina, S., & Maier, L. (2023). High-throughput anaerobic screening for identifying compounds acting against gut bacteria in monocultures or communities. *Nature Protocols* *2023* *19*:3, *19*(3), 668–699.
<https://doi.org/10.1038/s41596-023-00926-4>
- Nagata, N., Nishijima, S., Miyoshi-Akiyama, T., Kojima, Y., Kimura, M., Aoki, R., Ohsugi, M., Ueki, K., Miki, K., Iwata, E., Hayakawa, K., Ohmagari, N., Oka, S., Mizokami, M., Itoi, T., Kawai, T., Uemura, N., & Hattori, M. (2022). Population-level Metagenomics Uncovers Distinct Effects of Multiple Medications on the Human Gut Microbiome. *Gastroenterology*, *163*(4), 1038–1052. <https://doi.org/10.1053/J.GASTRO.2022.06.070>
- Power, C., Sayers, R., O'Brien, B., Clancy, C., Furey, A., Jordan, K., & Danaher, M. (2013). Investigation of the persistence of closantel residues in bovine milk following lactating-cow and dry-cow treatments and its migration into dairy products. *Journal of Agricultural and Food Chemistry*, *61*(36), 8703–8710. <https://doi.org/10.1021/JF4022866/>
- Ross, J., Belgodere, B., Chenthamarakshan, V., Padhi, I., Mroueh, Y., & Das, P. (2022). Large-scale chemical language representations capture molecular structure and properties. *Nature Machine Intelligence* *2022* *4*:12, *4*(12), 1256–1264. <https://doi.org/10.1038/s42256-022-00580-7>
- Sakenova, N., Cacace, E., Orakov, A., Huber, F., Varik, V., Kritikos, G., Michiels, J., Bork, P., Cossart, P., Goemans, C. V., & Typas, A. (2024). Systematic mapping of antibiotic cross-resistance and collateral sensitivity with chemical genetics. *Nature Microbiology* *2024* *10*:1, *10*(1), 202–216. <https://doi.org/10.1038/s41564-024-01857-w>
- Veterinary Medicines Directorate. (2022). *Maximum Residue Limits in Great Britain*.
https://assets.publishing.service.gov.uk/media/66ab59a5ab418ab05559315d/_2097921-v2-MRLs_in_GB.pdf
- Zeng, X., Xing, X., Gupta, M., Keber, F. C., Lopez, J. G., Lee, Y. C. J., Roichman, A., Wang, L., Neinast, M. D., Donia, M. S., Wühr, M., Jang, C., & Rabinowitz, J. D. (2022). Gut bacterial nutrient preferences quantified in vivo. *Cell*, *185*(18), 3441–3456.e19.
<https://doi.org/10.1016/J.CELL.2022.07.020>
- Zhang, Z. J., Cole, C. G., Coyne, M. J., Lin, H., Dylla, N., Smith, R. C., Pappas, T. E., Townson, S. A., Laliwala, N., Waligurski, E., Ramaswamy, R., Woodson, C., Burgo, V., Little, J. C., Moran, D., Rose, A., McMillin, M., McSpadden, E., Sundararajan, A., ... Comstock, L. E. (2024). Comprehensive analyses of a large human gut Bacteroidales culture collection

reveal species- and strain-level diversity and evolution. *Cell Host & Microbe*, 32(10), 1853-1867.e5. <https://doi.org/10.1016/J.CHOM.2024.08.016>

Zlamal, J. E., Leyn, S. A., Iyer, M., Elane, M. L., Wong, N. A., Wamsley, J. W., Vercruyse, M., Garcia-Alcalde, F., & Osterman, A. L. (2021). Shared and unique evolutionary trajectories to ciprofloxacin resistance in gram-negative bacterial pathogens. *MBio*, 12(3). <https://doi.org/10.1128/MBIO.00987-21/>